



# The consolidated European synthesis of CH₄ and N₂O emissions for EU27 and UK: 1990-2020

Ana Maria Roxana Petrescu[1], Chunjing Qiu[2], Matthew J. McGrath[2], Philippe Peylin[2], Glen P. Peters[3], Philippe Ciais[2], Rona L. Thompson[4], Aki Tsuruta[5], Dominik Brunner[6], Matthias Kuhnert[7], Bradley Matthews[8], Paul I. Palmer[9] Oksana Tarasova[10], Pierre Regnier[11], Ronny Lauerwald[12], David Bastviken[13], Lena Höglund-Isaksson[14], Wilfried Winiwarter[14,15], Giuseppe Etiope[16], , Tuula Aalto[5], Gianpaolo Balsamo[17], Vladislav Bastrikov[18], Antoine Berchet[2], Patrick Brockmann[2], Giancarlo Ciotoli[19], Giulia Conchedda[20], Monica Crippa[21], Frank Dentener[21], Christine D. Groot Zwaaftink[4], Diego Guizzardi[21], Dirk Günther[22], Jean-Matthieu Haussaire[6], Sander Houweling[1], Greet Janssens-Maenhout[21], Massaer Kouyate[2], Adrian Leip[21,*], Antti Leppänen[23], Emanuele Lugato[21], Manon Maisonnier[11], Alistair J. Manning[24], Tiina Markkanen[5], Joe McNorton[17], Marilena Muntean[21], Gabriel D. Oreggioni[(ex)21], Prabir K. Patra[25], Lucia Perugini[26], Isabelle Pison[2], Maarit T. Raivonen[23], Marielle Saunois[2], Arjo J. Segers[27], Pete Smith[7], Efisio Solazzo[(ex)21], Hanqin Tian[28], Francesco N. Tubiello[20], Timo Vesala[23,29], Chris Wilson[30,31], Sönke Zaehle[32]

[1]Department of Earth Sciences, Vrije Universiteit Amsterdam, 1081HV, Amsterdam, the Netherlands
[2]Laboratoire des Sciences du Climat et de l'Environnement, 91190 Gif-sur-Yvette, France
[3]CICERO Center for International Climate Research, Oslo, Norway
[4]Norwegian Institute for Air Research (NILU), Kjeller, Norway
[5]Finnish Meteorological Institute, P. O. Box 503, FI-00101 Helsinki, Finland
[6]Empa, Swiss Federal Laboratories for Materials Science and Technology, 8600 Dübendorf, Switzerland
[7]Institute of Biological and Environmental Sciences, University of Aberdeen, 23 St Machar Drive, Aberdeen, AB24 3UU, UK
[8]Umweltbundesamt GmbH, Climate change mitigation & emission inventories, 1090, Vienna, Austria
[9]School of GeoSciences, University of Edinburgh, Edinburgh, UK
[10]Science and Innovation Department, World Meteorological Organization (WMO), Geneva, Switzerland
[11]Biogeochemistry and Modeling of the Earth System, Université Libre de Bruxelles, 1050 Bruxelles, Belgium
[12]Université Paris-Saclay, INRAE, AgroParisTech, UMR ECOSYS, Thiverval-Grignon, France
[13]Department of Thematic Studies – Environmental Change, Linköping University, Sweden
[14]International Institute for Applied Systems Analysis (IIASA), 2361 Laxenburg, Austria
[15]Institute of Environmental Engineering, University of Zielona Góra, Zielona Góra, 65-417, Poland
[16]Istituto Nazionale di Geofisica e Vulcanologia, Sezione Roma 2, via V. Murata 605, Roma, Italy
[17]European Centre for Medium-Range Weather Forecasts (ECMWF), Reading, RG2 9AX, UK
[18]Science Partners, 75010 Paris, France
[19]Consiglio Nazionale delle Ricerche, Istituto di Geologia Ambientale e Geoingegneria, Via Salaria km 29300, 00015 Monterotondo, Rome, Italy
[20]Food and Agriculture Organization of the United Nations, Statistics Division. 00153 Rome, Italy
[21]European Commission, Joint Research Centre, 21027 Ispra (Va), Italy
*now at: European Commission, DG Research and Innovation, 1050 Brussels, Belgium
[22]Umweltbundesamt (UBA), 14193 Berlin, Germany
[23]University of Helsinki, Institute for Atmospheric and Earth System Research/Physics, Faculty of Science, 00560 Helsinki, Finland[24]Hadley Centre, Met Office, Exeter, EX1 3PB, UK
[25]Research Institute for Global Change, JAMSTEC, Yokohama 2360001, Japan
[26]Centro Euro-Mediterraneo sui Cambiamenti Climatici (CMCC), Viterbo, Italy
[27]Department of Climate, Air and Sustainability, TNO, Princetonlaan 6, 3584 CB Utrecht, the Netherlands
[28]International Centre for Climate and Global Change, School of Forestry and Wildlife Sciences, Auburn University, Auburn, AL 36849, USA
[29]Institute for Atmospheric and Earth System Research,/Forest Sciences, Faculty of Agriculture and Forestry, University of Helsinki, Helsinki, Finland
[30]Institute for Climate and Atmospheric Science, University of Leeds, Leeds, UK
[31]National Centre for Earth Observation, University of Leeds, Leeds, UK
[32]Max Planck Institute for Biogeochemistry (MPI-BGC), Jena, Germany

*Correspondence to*: A.M. Roxana Petrescu (a.m.r.petrescu@vu.nl)



**Abstract**

Knowledge of the spatial distribution of the fluxes of greenhouse gases and their temporal variability as well as flux attribution to natural and anthropogenic processes is essential to monitoring the progress in mitigating anthropogenic emissions under the Paris Agreement and to inform its Global Stocktake. This study provides a consolidated synthesis of $CH_4$ and $N_2O$ emissions using bottom-up (BU) and top-down (TD) approaches for the European Union and UK (EU27+UK) and updates earlier syntheses (Petrescu et al., 2020, 2021). The work integrates updated emission inventory data, process-based model results, data-driven sector model results, inverse modelling estimates, and extends the previous period 1990-2017 to 2020. BU and TD products are compared with European National GHG Inventories (NGHGI) reported by Parties under the United Nations Framework Convention on Climate Change (UNFCCC) in 2021. The uncertainties of NGHGIs were evaluated using the standard deviation obtained by varying parameters of inventory calculations, reported by the EU Member States following the guidelines of the Intergovernmental Panel on Climate Change (IPCC) and harmonized by gap-filling procedures. Variation in estimates produced with other methods, such as atmospheric inversion models (TD) or spatially disaggregated inventory datasets (BU), arise from diverse sources including within-model uncertainty related to parameterization as well as structural differences between models. By comparing NGHGIs with other approaches, the activities included are a key source of bias between estimates e.g. anthropogenic and natural fluxes, which, in atmospheric inversions are sensitive to the prior geospatial distribution of emissions. **For $CH_4$ emissions,** over the updated 2015-2019 period, which covers a sufficiently robust number of overlapping estimates, and most importantly the NGHGIs, the anthropogenic BU approaches are directly comparable, accounting for mean emissions of 20.5 Tg $CH_4$ yr$^{-1}$ (EDGAR v6.0, last year 2018) and 18.4 Tg $CH_4$ yr$^{-1}$ (GAINS, 2015), close to the NGHGI estimates of $17.5 \pm 2.1$ Tg $CH_4$ yr$^{-1}$. TD inversions estimates give higher emission estimates, as they also detect natural emissions. Over the same period, high resolution regional TD inversions report a mean emission of 34 Tg $CH_4$ yr$^{-1}$. Coarser-resolution global-scale TD inversions result in emission estimates of 23 Tg $CH_4$ yr$^{-1}$ and 24 Tg $CH_4$ yr$^{-1}$ inferred from GOSAT and surface (SURF) network atmospheric measurements, respectively. The magnitude of natural peatland and mineral soils emissions from the JSBACH-HIMMELI model, natural rivers, lakes and reservoirs emissions, geological sources and biomass burning together could account for the gap between NGHGI and inversions and account for 8 Tg $CH_4$ yr$^{-1}$. **For $N_2O$ emissions**, over the 2015-2019 period, both BU products (EDGAR v6.0 and GAINS) report a mean value of anthropogenic emissions of 0.9 Tg $N_2O$ yr$^{-1}$, close to the NGHGI data ($0.8 \pm 55$ % Tg $N_2O$ yr$^{-1}$). Over the same period, the mean of TD global and regional inversions was 1.4 Tg $N_2O$ yr$^{-1}$ (excluding TOMCAT which reported no data). The TD and BU comparison method defined in this study can be 'operationalized' for future annual updates for the calculation of $CH_4$ and $N_2O$ budgets at the national and EU27+UK scales. Future comparability will be enhanced with further steps involving analysis at finer temporal resolutions and estimation of emissions over intra-annual timescales, of great importance for $CH_4$ and $N_2O$, which may help identify sector contributions to divergence between prior and posterior estimates at the annual/inter-annual scale. Even if currently comparison between $CH_4$ and $N_2O$ inversions estimates and NGHGIs is highly uncertain because of the large spread in the inversion results, TD inversions inferred from atmospheric observations represent the most independent data against which inventory totals can be compared. With anticipated improvements in atmospheric modelling and observations, as well as modelling of natural fluxes, TD



inversions may arguably emerge as the most powerful tool for verifying emissions inventories for CH$_4$, N$_2$O and other
GHGs. The referenced datasets related to figures are visualized at https://doi.org/10.5281/zenodo.6992472 (Petrescu
et al., 2022).

## 1. Introduction

Atmospheric concentrations of greenhouse gases (GHGs) reflect a balance between emissions from sources and
removals by sinks, the former arising from both human activities and natural sources and the latter being found in the
biosphere, oceans and atmospheric oxidation. Increasing levels of GHG in the atmosphere due to human activities
have been the major driver of climate change since the pre-industrial period (pre-1750). In 2020, GHG mole fractions
were record highs, with globally averaged mole fractions reaching 1889±2 parts per billion (ppb) for methane (CH$_4$)

and 333.2±0.1 ppb for nitrous oxide (N$_2$O), representing 262% and 123% of respective pre-industrial levels (WMO,
2021). Since 2004, when CH$_4$ registered a negative dip, the trend in the CH$_4$ concentration in the atmosphere continues
to increase (NOAA, atmospheric data: https://www.esrl.noaa.gov/gmd/ccgg/trends_ch4/, last access May 2022). This
increase was attributed to anthropogenic emissions from agriculture (livestock enteric fermentation and rice
cultivation (12%) and fossil fuel related activities (17%), combined with a contribution from natural tropical wetlands

(Saunois et al., 2020, Thompson et al. 2018, Feng et al, 2022a,b). The increase in atmospheric N$_2$O also continues to
rise with the highest annual increase ever recorded in 2020 (https://gml.noaa.gov/ccgg/trends_n2o/, last access: May
2022). The main sources remain linked to agriculture particularly the application of nitrogen fertilizers and livestock
manure on agricultural land (FAO, 2020, 2015; IPCC, 2019b, Tian et al., 2020).

National GHG emission inventories (NGHGIs) are prepared and reported on an annual basis by Annex I

Parties[1] to the United Nations Framework Convention on Climate Change (UNFCCC). These inventories contain
annual time series of each country's GHG emissions from the 1990 base year[2] until two years before the year of
reporting and were originally set to track progress towards their reduction targets under the Kyoto Protocol (UNFCCC,
1997). Non-Annex I Parties[3] to the UNFCCC also provide emissions estimates in Biennial Update Reports (BURs) as
well as through National Communications (NCs); however, non-Annex I emissions are neither reported for annually

nor use harmonized formats due to the comparatively less-stringent reporting requirements. Annex I NGHGIs are
reported according to the Decision 24/CP.19 of the UNFCCC Conference of the Parties (COP) which states that the
national inventories *shall* be compiled using the methodologies provided in the *IPCC Guidelines for National*

---

[1] Annex I Parties include the industrialized countries that were members of the OECD (Organization for Economic Co-operation and Development)
in 1992 plus countries with economies in transition (the EIT Parties), including the Russian Federation, the Baltic States, and several central and
eastern European states (UNFCCC, https://unfccc.int/parties-observers, last access: February 2022). Under the Paris agreement all countries are
requested to report their emissions.

[2] For most Annex I Parties, the historical base year is 1990. However, parties included in Annex I with an economy in transition during the early
1990s (EIT Parties) were allowed to choose one year up to a few years before 1990 as reference because of a non-representative collapse during the
breakup of the Soviet Union (e.g., Bulgaria, 1988, Hungary, 1985–1987, Poland, 1988, Romania, 1989, and Slovenia, 1986).

[3] Non-Annex I Parties are mostly developing countries. Certain groups of developing countries are recognized by the Convention as being especially
vulnerable to the adverse impacts of climate change, including countries with low-lying coastal areas and those prone to desertification and drought.
Others (such as countries that rely heavily on income from fossil fuel production and commerce) feel more vulnerable to the potential economic
impacts of climate change response measures. The Convention emphasizes activities that promise to answer the special needs and concerns of these
vulnerable countries, such as investment, insurance and technology transfer (UNFCCC, https://unfccc.int/parties-observers, last access: February
2022).



*Greenhouse Gas Inventories* (IPCC, 2006). The 2006 IPCC Guidelines provide methodological guidance for estimating emissions for well-defined sectors using national activity and available emission factors. Decision trees indicate the appropriate level of methodological sophistication (methodological *Tier*) based the absolute contribution of the sector to the national GHG balance (is the source or sink a *Key Category* or not) and the country's national circumstances (availability and resolution of national activity data and emission factors). Generally, Tier 1 methods are based on global or regional default emission factors that can be used with aggregated activity data, while Tier 2 methods rely on country-specific factors and/or activity data at a higher subsector resolution. Tier 3 methods are based on more detailed process-level modelling or even facility-level emission measurements. Annex I Parties are furthermore required to estimate and report uncertainties in emissions (95% confidence interval) following the 2006 IPCC guidelines using, as a minimum requirement, the Gaussian error propagation method (approach 1). Annex I Parties may use Monte-Carlo methods (approach 2) or a hybrid approach and are encouraged to do so.

Annex I NGHGIs should follow principles of transparency, accuracy, consistency, completeness and comparability (TACCC) under the guidance of the UNFCCC (UNFCCC, 2014) and as mentioned above, shall be completed following the 2006 IPCC guidelines (IPCC, 2006). In addition, the IPCC 2019 Refinement (IPCC, 2019a), that may be used to complement the 2006 IPCC guidelines, has updated sectors with additional emission sources and provides guidance on the use of atmospheric data for independent verification of GHG inventories. Complementary to the NGHGIs, research groups and international institutions produce estimates of national GHG emissions, with two kind of approaches: atmospheric inversions (top-down, TD) and GHG inventories based on the same principle as NGHGI but using activity and/or emissions factors from (partially) different sources (bottom-up, BU).

The two approaches (BU and TD) provide useful insights on emissions from two different point of view. First, TD approaches act as an additional quality control tool for BU and NGHGI approaches, and facilitates a deeper understanding of the processes driving changes in different elements of GHG budgets. Second, NGHGIs cover regularly only a subset of countries (Annex I), and it is therefore necessary to construct BU estimates independently for all countries. Furthermore, while additional BU methods do not have prescribed standards like the IPCC Guidelines, independent BU methods can draw on different input data, or can provide estimates at higher-sectoral resolution, and therefore add complementary information to help quality control NGHGIs and help inform climate mitigation policy processes. Additionally, BU estimates are needed as input for TD estimates. As there is no formal guideline to estimate uncertainties in TD or BU approaches, uncertainties are usually assessed from the spread of different estimates within the same approach, though some groups or institutions report uncertainties for their individual estimates using a variety of methods, for instance, by performing sensitivity tests (Monte Carlo approach) on input data parameters. However, this can be logistically and computationally difficult when dealing with complex process-based models.

Despite the important insights gained from complementary BU and TD emission estimates, it should be noted that comparisons with the official reported is not always straightforward. BU estimates often share common methodology and input data, and through harmonization, structural differences between BU estimates and NGHGIs can be bridged. However, the use of common input data, albeit to varying extents, restricts the independence between the datasets and, from a verification perspective, may limit the conclusions drawn from the comparisons. On the other



hand, TD estimates are constrained by independent atmospheric observations and can serve as an additional, almost-fully independent quality control for NGHGIs. Nonetheless, structural differences between NGHGIs (what sources and sinks are included, and where and when emissions/removals occur) and the actual fluxes of GHGs to the atmosphere must be factored in to the comparison of estimates. While NGHGIs go through a central QA/QC review process, the IPCC procedures do not incorporate mandatory large-scale observation-derived verification.

Nevertheless, the individual countries may use atmospheric data and inverse modelling within their data quality control, quality assurance and verification processes, with expanded and updated guidance provided in chapter 6 of the 2019 Refinement of IPCC 2006 Guidelines (IPCC, 2019). So far, only a few countries (e.g. Switzerland, UK, New Zealand and Australia) have used atmospheric observations to constrain national emissions and documented these verification activities in their national inventory reports (Bergamaschi et al., 2018).

A key priority in the current policy process is to facilitate the 5 yearly Global Stocktakes (GSTs) of the Paris agreement, the first of which is in 2023, and to assess collective progress towards achieving the near- and long-term objectives, considering mitigation, adaptation and means of implementation. The GSTs are expected to create political momentum for enhancing commitments in Nationally Determined Contributions (NDCs) under the Paris Agreement. Though the modalities of the GSTs implementation are not clear, the key component of this process will be the NGHGI

reporting by countries under the Enhanced transparency framework of the Paris Agreement. Under the framework, emissions reporting will move away from the differential Annex I and non-Annex I reporting requirements and become more harmonized across Parties. Non-Annex I parties will be required to follow the 2006 IPCC guidelines and provide regular (biennial) national GHG inventory reports to the UNFCCC, alongside developed countries, that will continue to submit their inventories on an annual basis. Some developing countries will face challenges to

construct and subsequently update their NGHGIs and meet the more-stringent reporting requirements.

     The work presented in this paper covers dozens of distinct datasets and models, in addition to the individual country submissions to the UNFCCC of the EU Member States and the UK. As Annex I Parties, the NGHGIs of the EU Member States and the UK are consistent with the general guidance laid out in IPCC (2006) yet still differ in specific approaches, models, and parameters, in addition to differences underlying activity datasets. A comprehensive

investigation of detailed differences between all datasets is beyond the scope of this paper, though systematic analyses have been previously made for specific sectors (e.g. agriculture Petrescu et al., 2020) and by the Global Carbon Project $CH_4$ and $N_2O$ syntheses (Saunois et al., 2020 and Tian et al., 2020). The focus of this paper is on updates of the information from Petrescu et al., 2021a discussing whenever needed the changes in terms of emissions and trends. The data from Petrescu et al., 2021a is labeled as v2019, while the latest results are labeled v2020 and v2021

respectively. Except for one on $N_2O$, the global inversions did not provide an update for v2021, and, therefore, the earlier results are incorporated into this synthesis.

     As this is the most comprehensive comparison of NGHGIs and research datasets (including both TD and BU approaches) for the European continent to date, the focus of the paper is on improvement of estimates in the most recent version in comparison with the previous one, changes in the uncertainty estimates and identification of the

knowledge gaps and added value of the updated data sets for policy making. Official anthropogenic NGHGI emissions were compared with research datasets, including necessary harmonization of the latter on total emissions to ensure



consistency. Differences and inconsistencies between emissions were analyzed, and recommendations were made towards future evaluation of NGHGI data. While NGHGI include uncertainty estimates, individual spatially disaggregated research datasets of emissions often lack quantification of uncertainty. Here, the median[4] and minimum/maximum (min/max) range of different research products of the same type were used in this work to get a first estimate of uncertainty.

**2. $CH_4$ and $N_2O$ data sources and estimation approaches**

The $CH_4$ and $N_2O$ emissions in the EU27+UK from inversions and anthropogenic emissions inventories from various BU approaches covering specific sectors were analyzed. The data (Table 2) span the period from 1990 to 2020, with some of the data only available for shorter time periods. The estimates are available both from peer-reviewed literature and from unpublished research results from the VERIFY project (Table 1 and Appendix A) and in this work they are compared with NGHGIs reported in 2021 (time series for 1990-2019). Data sources are summarized in Table 2 with the detailed description of all products provided in Appendix A1-A3.

For both $CH_4$ and $N_2O$ BU approaches, inventories of anthropogenic emissions covering all sectors (EDGAR v6.0 and GAINS) and models and inventories limited to agriculture (CAPRI, FAOSTAT, DayCent, ECOSSE) were used. For $CH_4$ biogeochemical models of natural peatland emissions (JSBACH-HIMMELI), and lakes and reservoirs emissions (Lauerwald et al., 2019; Maisonnier et al., in prep.), as well as updated data for inland waters (rivers, lakes and reservoirs; Lauerwald et al., in prep.) and updated data for total geological emissions (Etiope et al., 2019) were used. Emissions from gas hydrates and termites are not included as they are close to zero in the EU27+UK (Saunois et al., 2020). Anthropogenic NGHGI $CH_4$ emissions from the LULUCF sector are very small for EU27+UK (3 % in 2019 including biomass burning) (section 2.2).

TD approaches include both regional and global inversions, the latter having a coarser spatial resolution. These estimates are described in section 2.3.

For $N_2O$ emissions, the same global BU inventories as for $CH_4$, natural emissions from inland waters (rivers, lakes and reservoirs) (RECCAP2 Lauerwald et al., in prep) were used, which did not change with respect to Petrescu et al., 2021a. In this study, about 66 % of the $N_2O$ emitted by Europe's natural rivers are considered anthropogenic indirect emissions, caused by leaching and run-off of N-fertilizers from the agriculture sector. One important update is the inclusion of estimates of natural $N_2O$ emissions from soils simulated with the O-CN model (Zaehle et al., 2011). These emissions are derived from model simulations in which land-use and atmospheric $CO_2$ remain constant, but climate varies through to 2020. These estimates are considered to be closer to what background natural $N_2O$ emissions would be present day, so they were used for subtraction from outputs of inversions (as it has a reasonable representation of the inter-annual variability (IAV)). The TD $N_2O$ inversions include one regional inversion FLEXINVERT and three global inversions (Friedlingstein et al., 2019; Tian et al., 2020, Patra et al., 2022).Agricultural sector emissions of $N_2O$ were presented in detail by Petrescu et al., 2020. In this current study,

---

[4] The reason for using median instead of mean for the ensembles is because there is a large spread between global inversions and we don't want to be biased by outliers/extremes.





CAPRI and ECOSSE models and FAO provided updated emissions , with the latter additionally covering non-$CO_2$ emissions from biomass burning as a contribution to LULUCF. Fossil fuel related emissions and industrial emissions were obtained from GAINS (see Appendix A1). Table A2 in Appendix A presents the methodological differences of the current study with respect to Petrescu et al., 2020 and Petrescu et al., 2021a.


*Table 1: Sectors included in this study and data sources providing estimates for these sectors.*

| Anthropogenic (BU)[5] $CH_4$ and $N_2O$ | Natural (BU)[6] $CH_4$ | Natural** (BU) $N_2O$ | TD ($CH_4$ and $N_2O$) |
|---|---|---|---|
| 1. Energy **(NGHGI, GAINS, EDGAR v6.0)** | | | No sectoral split – total emissions **FLExKF (CAMSv19r); TM5-4DVAR; FLEXINVERT; CTE-$CH_4$ InGOS inversions GCP-$CH_4$ 2019 anthropogenic partition from inversions GCP-$CH_4$ 2019 Natural partition from inversions G$N_2O$B 2019 CHIMERE InTEM NAME (only for UK)** |
| 2. Industrial Products and Products in Use (IPPU) **(NGHGI, GAINS, EDGAR v6.0)** | | | |
| 3. Agriculture* **(NGHGI, CAPRI, GAINS, EDGAR v6.0, FAOSTAT, ECOSSE and DayCent (only for $N_2O$)** | | | |
| 4. LULUCF total emissions **(NGHGIs Fig. 1,2,4,6, B1a for $CH_4$ and Fig. 10, 11, 14 and B1b for $N_2O$)** | | | |
| 5. Waste **(NGHGI, GAINS, EDGAR v6.0)** | | | |
| | Peatlands, mineral soils, inland waters (lakes, rivers and reservoirs) and geological fluxes **(JSBACH-HIMMELI, inland water RECCAP2 estimate, Rosentreter_et_al, Etiope et al., 2019) with updated activity (this study), biomass burning GFEDv4.1** | Inland water (lakes, rivers and reservoirs) fluxes **(inland water RECCAP2 estimate), biomass burning GFEDv4.1, pre-industrial natural soil emissions (O-CN)** | |

\* Anthropogenic (managed) agricultural soils can also have a level of natural emissions.

\*\*Natural soils (unmanaged) can have both natural and anthropogenic emissions.

The units used in this paper are metric tonne (t) [1kt = $10^9$ g; 1Mt = $10^{12}$g] of $CH_4$ and $N_2O$. The referenced data used for the figures' replicability purposes are available for download at https://doi.org/10.5281/zenodo.6992472

---

[5] For consistency with the NGHGI, here we refer to the five reporting sectors as defined by the UNFCCC and the Paris Agreement decision (18/CMP.1),the IPCC Guidelines (IPCC, 2006), and their Refinement (IPCC, 2019a), with the only exception that the latest IPCC Refinement groups together Agriculture and LULUCF sectors in one sector (Agriculture, Forestry and Other land Use - AFOLU).

[6] The term natural refers here to unmanaged natural $CH_4$ emissions (peatlands, mineral soils, geological, inland waters and biomass burning) not reported under the UNFCCC LULUCF sector.



(Petrescu et al., 2022). Upon request, the codes necessary to plot the figures in the same style and layout can be provided. The focus is on EU27+UK emissions. In the VERIFY project, an additional web tool was developed which allows for the selection and display of all plots shown in this paper (as well as the companion paper on $CO_2$), not only

for the EU Member States and UK but for a total of 79 countries and groups of countries in Europe (Table A1, Appendix A). The data, located on the VERIFY project website: http://webportals.ipsl.jussieu.fr/VERIFY/FactSheets/, is free and can be accessed upon registration.

### 2.1. CH₄ and N₂O anthropogenic emissions from NGHGI

Anthropogenic $CH_4$ emissions from the four UNFCCC sectors (excluding LULUCF) were grouped together. Anthropogenic $CH_4$ emissions in 2019 account for 17.1 Tg $CH_4$ yr$^{-1}$ and represent 10.5 % of the total EU27+UK emissions (in $CO_2e$ , GWP 100 years, IPCC AR4[7]). $CH_4$ emissions are predominantly related to agriculture (9.2 Tg $CH_4$ yr$^{-1}$ ± 0.8 Tg $CH_4$ yr$^{-1}$) or 53.8 % in 2019 (52.5 % in 2018) of the total EU27+UK $CH_4$ emissions. Anthropogenic NGHGI $CH_4$ emissions from the LULUCF sector are very small for EU27+UK e.g. 0.5 Tg $CH_4$ yr$^{-1}$ or 3 % in 2019,

including emissions from biomass burning.

Regarding $CH_4$ emissions from wetlands, following the recommendations of the 2013 IPCC Wetlands supplement (IPCC, 2014) only emissions from managed wetlands are reported by Parties. According to NGHGI data between 2008 and 2018, managed wetlands in the EU27+UK for which emissions were reported under LULUCF (CRF table 4(II) accessible for each EU27+UK country[8]) represent one fourth of the total wetland area in EU27+UK

(G. Grassi, EC-JRC, pers. comm.) and their emissions summed up in 2019 to 0.1 Tg $CH_4$ yr$^{-1}$.

Anthropogenic $N_2O$ emissions (excluding LULUCF) in 2019 account for 0.8 Tg $N_2O$ yr$^{-1}$ and represent 6.2 % of the total EU27+UK emissions in $CO_2eq$. $N_2O$ emissions are predominantly related to agriculture (0.6 Tg $N_2O$ yr$^{-1}$ or 73.0 % in 2019 (73.5 % in 2018) of the total EU27+UK (including LULUCF+BB) $N_2O$ emissions) but are also found in the other sectors (Tian et al. 2020). In addition, $N_2O$ has natural sources, which are defined as the pre-

industrial background emissions before the use of synthetic N-fertilizers and intensive agriculture, and derive from natural processes in soils but also in lakes, rivers and reservoirs (Lauerwald et al., in prep, Maavara et al., 2019; Lauerwald et al., 2019; Tian et al., 2020).

### 2.2. CH₄ and N₂O anthropogenic and natural emissions from other bottom-up estimates


Data from five global data sets and models of $CH_4$ and $N_2O$ anthropogenic emissions inventories were used, namely: CAPRI, DayCent, ECOSSE, FAOSTAT, GAINS and EDGAR v6.0 (Table 3). These estimates are not completely independent from NGHGIs (see Figure 4 in Petrescu et al., 2020) as they integrate their own sectorial modelling with the UNFCCC data (e.g., common activity data and IPCC emission factors) when no other source of

information is available. The $CH_4$ biomass and biofuel burning emissions are included in NGHGI under the UNFCCC

---

[7] IPCC AR4 GWP 100 values are still used by the Member States in their NGHGI reporting to the UNFCCC.
[8] https://unfccc.int/process-and-meetings/transparency-and-reporting/reporting-and-review-under-the-convention/greenhouse-gas-inventories-annex-i-parties/national-inventory-submissions-2019



LULUCF sector, although they are identified as a separate category by the Global Carbon Project $CH_4$ budget synthesis (Saunois et al., 2020). For both $CH_4$ and $N_2O$, CAPRI (Britz and Witzke, 2014; Weiss and Leip, 2012) and FAOSTAT (FAO, 2021) report only agricultural emissions. DayCent and ECOSSE report only emissions for agriculture $N_2O$. Out of all BU inventories, only CAPRI reported new uncertainties for 2014, 2016 and 2018, while

values for EDGARv6.0 were the same (Solazzo et al., 2021) as those reported in Petrescu et al., 2021a.

In this study, natural $CH_4$ emissions are included under the category "peatlands" and "other natural emissions", the latter including geological emissions, biomass burning emissions and two estimates of inland waters (rivers, lakes and reservoirs). One inland water estimate comes from "process-based models" and is based on the Rosentreter et al., (2021) with ranges from Bastviken et al., (2011) and Stanley et al., (2016) and the second represent

an upscaled estimate for inland waters from the RECCAP2 project (Lauerwald et al., in prep).

For peatlands and mineral soils, the JSBACH-HIMMELI framework was used. Additionally, the ensemble of thirteen monthly gridded estimates of peatland emissions based on different land surface models as calculated for Saunois et al. (2020) were used as described in Appendix B2. Geological emissions were initially based on the global gridded emissions from Etiope et al. 2019 and previously estimated to be 1.3 Tg $CH_4$ yr$^{-1}$ (Petrescu et al., 2021a). For

this study these emissions were recalculated, using more detailed input data related to the activity, i.e., a more precise estimate of the continental oil-gas field area (which determines the potential area of microseepage) and offshore seepage area (Appendix A2) and now account for 3.3 Tg $CH_4$ yr$^{-1}$ (0.9 Tg $CH_4$ yr$^{-1}$ from offshore marine seepage and 2.4 Tg $CH_4$ yr$^{-1}$ onshore). This rescaled geological source represent the second largest natural component accounting for 42 % of the total EU27+UK natural $CH_4$ emissions. The upscaled inland waters, (rivers, lakes, and

reservoirs, based on Lauerwald et al., in prep) are the largest component of natural emissions (3.3 Tg $CH_4$ yr$^{-1}$ and ranging from 2.7 Tg $CH_4$ yr$^{-1}$ to 4.3 Tg $CH_4$ yr$^{-1}$) and account for 44 %. The remaining 14 % emissions are attributed to peatlands, mineral soils and biomass burning. Overall, in EU27+UK the natural emissions thus accounted for 8 Tg $CH_4$ yr$^{-1}$. Finally, It should be noted that to a small extent the $CH_4$ natural emissions from waters are also due to an anthropogenic component, namely eutrophication following N-fertilizer leaching to inland waters.

The $N_2O$ anthropogenic emissions from inventory datasets belong predominantly to agriculture and are associated to two main categories: 1) direct emissions from the agricultural sector where synthetic fertilizers and manure were applied, and from manure management, and 2) indirect emissions on non-agricultural land and water receiving anthropogenic N through atmospheric N deposition, leaching and run-off (also from agricultural land). Additional anthropogenic emissions result from industrial processes, in particular, adipic and nitric acid production,

which are declining owing to the implementation of emission abatement technologies. Other $N_2O$ emissions come from the wastewater treatment activities and fossil fuel combustion.

In this study, "natural" $N_2O$ fluxes refer to emissions from inland waters (lakes, rivers and reservoirs, Maavara et al., 2019; Lauerwald et al., 2019, Lauerwald et al., in prep,) which include also lakes with dams. The other component is the natural $N_2O$ emissions from soils simulated with the O-CN model (Zaehle et al., 2011). Regarding

the inland water emissions, more than half of the emissions (56 % globally, Tian et al., 2020, and 66 % for Europe this study) are due to enhanced N inputs from fertilizers, manure, sewage and, to a smaller extent, atmospheric N



deposition. However, emissions from natural soils in this study are considered as "anthropogenic" because, according to the country specific National Inventory Reports (NIRs), all land in EU27+UK is considered to be managed

For both $CH_4$ and $N_2O$ the natural biomass burning emissions from GFEDv4.1 (van der Werf et al., 2017)
are included in Figs. 1, 4b, 5b, 9 and 13, while for $CH_4$ only, biomass burning emissions from the GCP 2020 (Saunois et al., 2020) are included in Fig 6.

### 2.3. $CH_4$ and $N_2O$ emission data from inversions

Atmospheric inversions optimize prior estimates of emissions and sinks through modeling frameworks that
utilizes atmospheric observations as a constraint on fluxes. Emission estimates from inversions depend on the data set of atmospheric measurements and the choice of the atmospheric model, as well as on other inputs (e.g., prior emissions and their uncertainties). Inversion results were taken from original publications without evaluation of their performance through specific metrics (e.g., fit to independent cross validation atmospheric measurements (Bergamaschi et al., 2013, 2018; Patra et al., 2016)). Some of the inversions allow for explicit attribution to different
sectors, while others optimize all fluxes in each grid cell and then attribute emissions to sectors using prior grid-cell fractions (see details in Saunois et al. 2020 for global inversions).

For $CH_4$, the same set of nine regional inversions and 22 global inversions as listed in Table 3 and presented in Petrescu et al., 2021a was used. While many different inversion have been used, it should be stressed that the variants are not completely independent of one another. Table B4, Appendix B in Petrescu et al., 2021a illustrates this
by documenting to what extent the transport models, priors and atmospheric measurement data vary between the inversion datasets". The subset of InGOS inversions (Bergamaschi et al., 2018a) belongs to a project where all models used the same atmospheric data over Europe covering the period 2006-2012. The global inversions from Saunois et al. 2020 were not updated for this work and cover a period until 2017.

The regional inversions generally use both higher-resolution prior data and higher-resolution transport models, and
e.g. TM5-JRC runs simultaneously over the global domain at coarse resolution and over the European domain at higher resolution, with atmospheric $CH_4$ concentration boundary conditions taken from global fields. For $CH_4$, 11 global inversions use GOSAT for the period 2010-2017, eight global inversions use surface stations (SURF) from 2000 to 2017, two global models use SURF since from 2010-2017 and one SURF from 2003-2017 (see "Appendix 4 Table" in Saunois et al. 2020). All regional inversions use observations from SURF stations as a base of their emission
calculation.

*Table 2: Data sources for $CH_4$ and $N_2O$ emissions used in this study:*

| Name | $CH_4$ | $N_2O$ | Contact / lab | References | Status compared to Petrescu et al., 2021a |
|---|---|---|---|---|---|
| **$CH_4$ and $N_2O$ Bottom-up anthropogenic** | | | | | |
| UNFCCC NGHGI (2021) CRFs | $CH_4$ emissions 1990-2019 | $N_2O$ emissions 1990-2019 | MS inventory agencies | UNFCCC CRFs https://unfccc.int/process-and-meetings/transparency- | updated |



| | | | Yearly uncertainties from UBA Vienna | and-reporting/reporting-and-review-under-the-convention/greenhouse-gas-inventories-annex-i-parties/national-inventory-submissions-2019 | |
|---|---|---|---|---|---|
| EDGAR v6.0 | CH$_4$ sectoral emissions 1990-2018 | N$_2$O sectoral emissions 1990-2018 | EC-JRC | Crippa et al., 2019a Crippa et al., 2019 EU REPORT Janssens-Maenhout et al., 2019 Solazzo et al., 2020 (in review ACP) | Updated |
| CAPRI | CH$_4$ agricultural emissions 1990-2014 and 2016, 2018 | N$_2$O agricultural emissions 1990-2014 and 2016, 2018 | EC-JRC | Britz and Witzke, 2014 Weiss and Leip, 2012 | Updated |
| GAINS | CH$_4$ sectoral emissions 1990-2015 | N$_2$O sectoral emissions 1990-2015 (every five years) | IIASA | Höglund-Isaksson, L. 2017 Höglund-Isaksson, L. et al., 2020 Winiwarter et al., 2018 | Not updated |
| FAOSTAT | CH$_4$ agriculture and land use emissions 1990-2019 | N$_2$O agricultural emissions 1990-2019 | FAO | Tubiello et al. 2013 FAO, 2015, 2020 Tubiello, 2019 | Updated |
| ECOSSE | | Direct N$_2$O emissions from agricultural soils 2000-2020 | UNIABDN | Bradbury et al., 1993 Coleman., 1996 Jenkinson., 1977, 1987 Smith et al., 1996, 2010a,b | Updated |
| DayCent | | N$_2$O emissions from direct agricultural soils avg. 2015-2019 | EC-JRC | Orgiazzi et al., 2018 Lugato et al., 2018, 2017 Quemada et al., 2020 | Updated |
| **CH$_4$ and N$_2$O natural** | | | | | |



| JSBACH-HIMMELI | $CH_4$ emissions from peatlands and mineral soils<br><br>2005-2020 | | FMI | Raivonen et al., 2017<br><br>Susiluoto et al., 2018 | Updated |
|---|---|---|---|---|---|
| Non-wetland inland waters | One average value for $CH_4$ fluxes from rivers, lakes and reservoirs with uncertainty<br><br>2010-2019<br><br>One median upscaled value from RECCAP2 analysis<br><br>1990-2019 | One median $N_2O$ value for emissions from lakes, rivers, reservoirs from the RECCAP2 analysis<br><br>1990-2019 | ULB | Maisonnier et al., in prep., after Maavara et al., 2017, 2019 and Lauerwald et al., 2019<br><br>Bastviken et al., 2011<br><br>Stanley et al. 2016<br><br>Rosentreter et al., 2021<br><br>Lauerwald et al., in prep. | Updated |
| Geological emissions, (onshore and offshore) | Global grid geological $CH_4$ emission model (2019) | | Istituto Nazionale di Geofisica e Vulcanologia (INGV) | Etiope et al., 2019 and this work (updated activity data) | updated |
| GFED4.1 | Biomass burning emissions<br><br>2000-2020 | Biomass burning emissions<br><br>2000-2020 | VU Amsterdam | van der Werf et al., 2017 | new |
| O-CN | | Background natural $N_2O$ emissions from soils (model simulations in which land-use and atm. $CO_2$ remain constant, but climate varies through to 2020) | MPI-BGC | Zaehle et al., 2011<br><br>Zaehle & Friend, 2010 | new |
| **$CH_4$ and $N_2O$ inversions**<br><br>**Regional inversions over Europe (high transport model resolution)** | | | | | |
| FLExKF-CAMSv19r | Total $CH_4$ emissions from inversions with uncertainty<br><br>2005-2019 | | EMPA | Brunner et al., 2012<br><br>Brunner et al., 2017<br><br>Background concentrations from CAMSv19r (Arjo Segers) | Updated |





| TM5-4DVAR | CH$_4$ emissions from inversions, split into total, anthropogenic and natural 2005-2018 | | EC-JRC | Bergamaschi et al., 2018a | Not updated |
|---|---|---|---|---|---|
| FLEXINVERT | CH$_4$ total emissions from inversions 2005-2018 | N$_2$O total emissions, 2005-2019 | NILU | Thompson and Stohl, 2014 | Updated for N$_2$O |
| CTE-CH$_4$ | Total CH$_4$ emissions from inversions for Europe with uncertainty 2005-2018 | | FMI | Brühl et al., 2014 Houweling et al., 2014 Giglio et al., 2013 Ito et al., 2012 Janssens-Maenhout et al., 2013 Krol et al., 2005 Peters et al., 2005 Saunois et al., 2020 Stocker et al., 2014 Tsuruta et al., 2017 | Not updated |
| InGOS inversions | Total CH$_4$ emissions from inversions 2006-2012 | | EC-JRC and InGOS project partners | Bergamaschi et al., 2018a TM5-4DVAR: Meirink et al., 2008; Bergamaschi et al. 2010; 2015 TM5-CTE: Tsuruta et al., 2017 LMDZ-4DVAR: Hourdin and Armengaud, 1999; Hourdin et al., 2006 TM3-STILT: Trusilova et al., 2010, Gerbig et al., 2003; Lin et al., 2003; Heimann and Koerner, 2003 NAME: Manning et al. 2011; Bergamaschi et al., 2015 CHIMERE: Berchet et al. 2015a; 2015b; Menut et al., 2013; Bousquet et al., 2011\ COMET: Eisma et al., 1995; Vermeulen et al., 1999; Vermeulen et al., 2006 | Not updated |
| **Global inversions from the Global Carbon Project CH$_4$ and N$_2$O budgets (Saunois et al. 2020, Tian et al., 2020)** | | | | | |



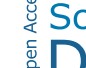

| GCP-CH$_4$ 2019 anthropogenic partition from inversions | 22 models for CH$_4$ inversions, both *SURF and GOSAT* 2000-2017 | | LSCE and GCP-CH$_4$ contributors | Saunois et al., 2020 and model specific references in Appendix B, Table B4 | Not updated |
|---|---|---|---|---|---|
| GCP-CH$_4$ 2019 Natural partition from inversions | 22 models with optimized wetland CH$_4$ emissions 2000-2017 | | LSCE | Saunois et al., 2020 and model specific references in Appendix B, Table B3 | Not updated |
| GN$_2$OB 2019 | | Inverse N$_2$O emissions - 3 Inversions PYVAR (CAMS-N$_2$O) TOMCAT MIROC4-ACTM 1998-2016 | GN$_2$OB 2019 and contributors | Thompson et al., 2019 Tian et al., 2020 | Not updated |

For N$_2$O, one regional inversion (FLEXINVERT) for the 2005-2019 period and three global inversions for the period 1998-2016 from Tian et al., (2020) and Thompson et al. (2019) were used as listed in Table 3. These estimates were not updated for this paper. These inversions are not completely independent from each other since most of them use the same input information (Appendix B3). The regional inversion uses a higher resolution atmospheric transport model for Europe, with atmospheric N$_2$O concentration boundary conditions taken from global model fields. As all inversions produced total rather than anthropogenic emissions, emissions from soils (O-CN) and inland waters (lakes, rivers and reservoirs) estimated by Lauerwald et al., in prep were subtracted from the total emissions. Note that inland water emissions include anthropogenic emissions from N-fertilizer leaching accounting for 66 % of the inland water emissions in EU27+UK. In 2019, emissions from inland waters represented 1.4 % of the total UNFCCC NGHGI (2021) N$_2$O emissions.

The largest share of N$_2$O emissions comes from agricultural soils (direct and indirect emissions from the applications of fertilizers, whether synthetic or manure) contributing in 2019 79 % of the total N$_2$O emissions (excluding LULUCF) in EU27+UK. In Petrescu et al., 2021a, "Table B1c, Appendix B1" presented the allocation of emissions by activity type covering all agricultural activities and natural emissions, following the IPCC (2006) sector classification scheme. Each data product has its own particular way of grouping emissions, and does not necessarily cover all emissions activities. The main inconsistencies between process-based models and inventories are observed regarding activity allocation in the two models, ECOSSE and DayCent. ECOSSE only estimates direct N$_2$O emissions, and does not estimate downstream emissions of N$_2$O, for example indirect emissions from nitrate leached into water courses, which also contributes to an underestimation of total N$_2$O emissions. Field burning emissions are also not included by most of the data sources.



## 3. Results and discussion

### 3.1. Comparing CH$_4$ emission estimates from different approaches

#### 3.1.1. *Estimates of European and regional total CH$_4$ fluxes*

Total CH$_4$ fluxes from EU27+UK and five main regions in Europe: North, West, Central, East (non-EU) and South are presented in the paper. The countries included in these regions, which include countries outside the
EU27+UK bloc, are all Annex I Parties to UNFCCC and are listed in Appendix A, table A. Figure 1 shows the total CH$_4$ fluxes from the NGHGIs for base year 1990, as well as five-year mean values for the 2011-2015 and 2015-2019 periods. We use the five-year periods as an exercise for what could be achieved in 2023, the year of the first GST, when for most parties to the Convention the reported inventories will include 2021. Given that the GST is only repeated every five years, a five-year average is clearly of interest.

The total NGHGI estimates include emissions from all sectors (excluding LULUCF) and are plotted and compared to fluxes from global datasets, BU models and inversions. There is a good agreement noted in absolute total values between inventories, as well as between regional and global inversion ensembles, but uncertainties (min/max ranges) are large. This match can be explained by interdependencies in input data (AD and EFs) for the BU estimates (Petrescu et al., 2020) and similar prior information used by inversions (Petrescu et al., 2021a). In Figure 1,
hatched transparent bars represent the 2011-2015 mean while colour-filled bars represent the new updated 2015-2019 mean values. For GAINS and some inversions that do not have annual estimates for all five years, only the average of available years is calculated (e.g., 2015 for GAINS).

For all study regions, 2019 CH$_4$ emissions decreased by 24 % (Southern Europe) to 57 % (Eastern Europe), with respect to NGHGI 1990 values; and for EU27+UK emissions decreased by 39 %. This is encouraging in the
context of meeting EU total GHG commitments under the Paris Agreement (55% decrease in 2030 compared to 1990 levels and reaching carbon neutrality by 2050). This reduction will need to be achieved by strong reductions in top emitter sectors (e.g. Agriculture) and compensated by sinks in the LULUCF sector. It also shows that not only at EU27+UK level, but also at regional European level, the emissions from BU (anthropogenic and natural) and TD estimates agree in magnitude with reported NGHGI data despite the high uncertainty associated with the TD estimates.
This uncertainty is represented here by the variability in the model ensembles and denotes the range (min and max) of estimates within each model ensemble. The comparison of TD to anthropogenic estimates (Fig. 1), suggests that the total CH$_4$ flux is dominated by natural emissions (i.e., Northern Europe) although comparison with EDGAR v6.0 would indicate that anthropogenic emissions are dominant (e.g. Northern, Central and Western Europe).



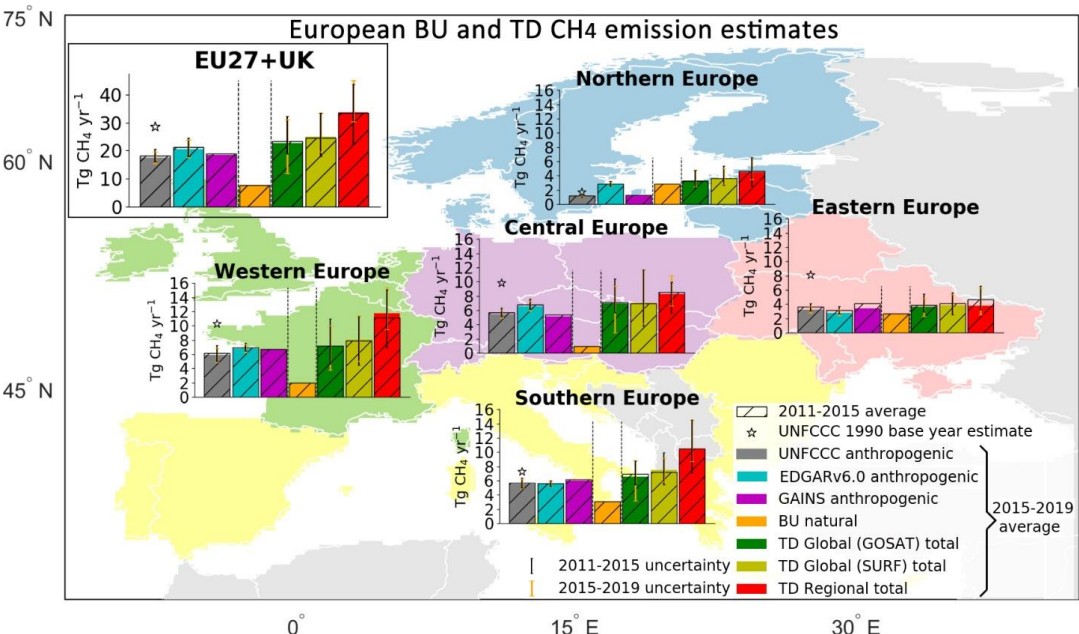

*Figure 1: Five-year means (2011-2015 and 2015-2019) in total CH₄ emission estimates (excluding LULUCF) for*
*EU27+UK and five European regions (North, West, Central, South and East non-EU). Eastern European region does*
*not include European Russia. Northern Europe includes Norway. Central Europe includes Switzerland. The data comes*
*from UNFCCC NGHGI (2021) submissions (grey), which are plotted with respective base year 1990 (black star)*
*estimates, two inventories (GAINS and EDGAR v6.0), natural unmanaged emissions (sum of peatland, geological,*
*inland waters (Reccap2) and GFEDv4.1 biomass burning emissions) and three inversion estimates: one regional*
*European inversion (excluding InGOS unavailable for 2013-2015) and GOSAT and SURF ensemble estimates from*
*global inverse models. The relative error on the UNFCCC value represents the NGHGI (2021) reported uncertainties*
*computed with the error propagation method (95% confidence interval) and gap-filled to provide respective*
*estimates for each year. Uncertainty for EDGAR v6.0 was calculated for 2015 based on the 95 % confidence interval*
*of a lognormal distribution (Solazzo et al., 2021).*

The EDGAR v6.0 updated estimates for Northern Europe remain two-times higher than NGHGI and GAINS
ones. The EDGAR approach is to use a globally harmonised methods and sources of data, which means that country-
specific detail is often replaced with global averages. In some countries and for some sectors or gases, these
assumptions lead to huge differences. For example, fugitive emissions of methane in the oil and gas sector are
estimated based on the level of production of oil and gas. In the case of Norway this ignores the substantial effects of
regulation on reducing such fugitive emissions. Instead, EDGAR's methane emissions estimates for Norway follow
the pattern of its total production of oil and gas (Olhoff et al., 2022). For Eastern Europe we note that all estimates
decreased compared to the previous five year mean and the BU anthropogenic estimates remain similar in magnitude
to the TD estimates of total CH₄ emissions. One possible explanation is that for TD estimates (i.e. using atmospheric





inversions) the fluxes are better constrained by a larger number of observations. Where there are fewer or no observations, like in Eastern Europe, the fluxes in the inversion will stay close to the prior estimates, since there is little or no information to adjust them.

In line with Bergamaschi et al., 2018a the potentially significant contribution from natural unmanaged sources (peatlands, mineral soils, geological and inland waters (RECCAP2)), which for EU27+UK accounted in 2019 for 8 Tg $CH_4$ yr$^{-1}$ (Figure 1) can be highlighted. Taking into account these natural unmanaged $CH_4$ emissions, and adding it to the range of the BU anthropogenic estimates (22 Tg $CH_4$ yr$^{-1}$ (NGHGI) – 26 Tg $CH_4$ yr$^{-1}$ (EDGARv6.0)) improves agreement with the TD estimates. BU estimates become consistent with the lower range of the regional total TD estimates (32 Tg $CH_4$ yr$^{-1}$ (TM5_JRC)– 41 Tg $CH_4$ yr$^{-1}$ (FLEXINVERT)) and show even better agreement in

absolute values with the global median SURF (24 Tg $CH_4$ yr$^{-1}$) and GOSAT (23 Tg $CH_4$ yr$^{-1}$) inversions. The broad consistency between the TD and BU estimates could be interpreted in two ways: 1) BU and TD regional estimates are similar given the large uncertainties and spread in TD results, or 2) regional TD higher estimates potentially indicate shortcomings of BU inventories, the latter interpretation being more consistent with the general atmospheric developments (WMO, 2021).

Is it notable to highlight that the regional TD total is considerably higher for all regions and EU27+UK total and by considering this estimate the best to date total estimate for the whole Europe, including all sources and sinks, this would infer a missing of 20 to 30 % of $CH_4$ emissions from the other BU approaches.

### 3.1.2. NGHGI sectoral emissions and decadal changes

According to the UNFCCC (2021) NGHGI estimates, in 2019 the EU27+UK emitted GHGs totaling 3.7 Gt $CO_2$e (including LULUCF), of this total, $CH_4$ emissions accounted for 11.8 % (0.4 Gt $CO_2$e or 17.5 Tg $CH_4$ yr$^{-1}$± 2.2 Tg $CH_4$ yr$^{-1}$ ) (Appendix, B2, Figure B2a) with France, UK and Germany together contributing 37 % of total $CH_4$ emissions.

The data in Figure 2 shows anthropogenic $CH_4$ emissions and their change from one decade to the next, from

UNFCCC NGHGI (2021), with the split between the different sectors. In 2019, NGHGI report $CH_4$ from agricultural activities to be 52.4 % (± 8.7 %) of the total EU27+UK $CH_4$ emissions, followed by emissions from waste, 27.5 % (± 22.5 %). The large share of agriculture in total anthropogenic $CH_4$ emissions also holds at global level (IPCC Special Report on Climate Change and Land (SRCCL), 2019). Between the 1990s and the 2000s, the net 17.6 % reduction originates largely from the energy and waste sectors, with only negligible contributions to emission trends and levels

from IPPU (metal and chemical industry) and LULUCF. Between the 2000s and 2010-2019, a further reduction by 16.5 % is observed with the waste sector as the largest contributor to this reduction. The two largest sectors contributing to total EU27+UK emission are agriculture and waste, but energy and waste are showing the higher reductions over the last decade.

The reduction observed in the waste sector coincide with the adoption of the first EU methane strategy

published in 1996 (COM(96) 557, 1996). EU legislation addressing emissions in the waste sector may have been successful to trigger the largest reductions. Directive 1999/31/ EC on the landfill (also referred to as the Landfill Directive) required the Member States to separate waste, minimizing the amount of biodegradable waste disposed



untreated in landfills and to install landfill gas recovery at all new sites. Based on the 1999 Directive, the new

2018/1999 EU Regulation on the Governance of the Energy Union requires the European Commission to propose a

strategic plan for methane, which will become an integral part of the EU's long-term strategy. In the waste sector, the

key proposal included the adoption of EU legislation requiring the installation of methane recovery and use systems

at new and existing landfills. Other suggested actions included measures aimed at the minimization, separate collection

and material recovery of organic waste (Olczak and Piebalgs, 2019).

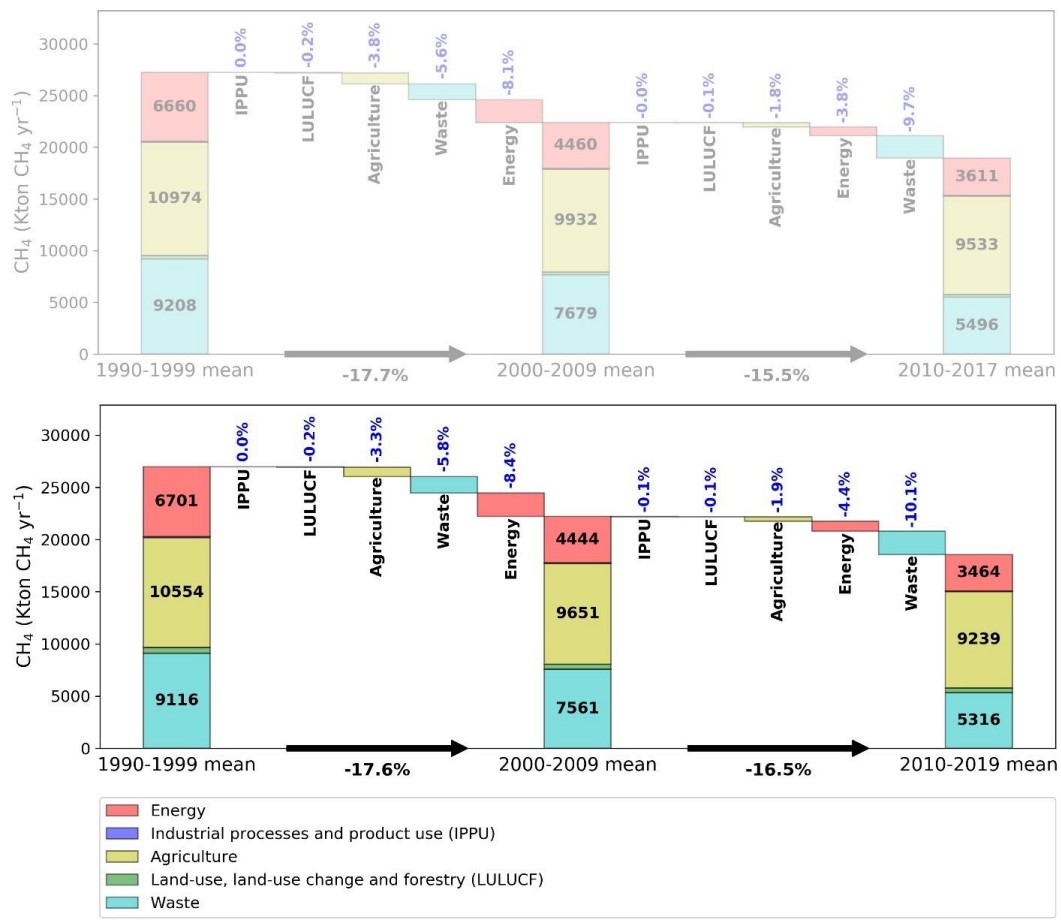


*Figure 2: The contribution of changes (%) in CH₄ anthropogenic emissions in the five sectors to the overall change*
*in decadal mean for the EU27+UK, as reported to UNFCCC. The top plot shows the previous NGHGI data from*
*Petrescu et al., 2021a and bottom plot illustrates data from UNFCCC NGHGI (2021). The three stacked columns*
*represent the average CH₄ emissions from each sector during three periods (1990-1999, 2000-2009 and 2010-2019)*

*and percentages represent the contribution of each sector to the total reduction percentages (black arrows) between*
*periods.*

### 3.1.3. NGHGI estimates compared with bottom-up inventories

The data in Figure 3 presents the total anthropogenic $CH_4$ emissions from four BU inventories and UNFCCC

NGHGI (2021) submissions excluding emissions from LULUCF, which was identified to a non-significant contributor (Figure 2). According to NGHGI, in 2019 anthropogenic $CH_4$ emissions from the four sectors (Table 1, excluding LULUCF) amounted to 17.1 Tg $CH_4$ yr$^{-1}$, representing 10.5 % of the total EU27+UK GHG emissions in $CO_2$eq.. Figure 3a shows EDGARv6.0 and GAINS trends being consistent with the ones of NGHGI (excluding LULUCF), although while GAINS and NGHGI agree in terms of emissions levels. EDGARv6.0estimates are consistently higher

estimates (~19 %) than NGHGI. In contrast to the previous version, EDGAR v4.3.2, which was found by Petrescu et al. 2020 to be consistent with NGHGI (2018) data, EDGAR v6.0 reports higher estimates then EDGARv5.0 (~8% higher) and falls outside the 9.6 % UNFCCC uncertainty range. Over the 1990-2019 period, the trends in emissions agree well between the two BU data sets and NGHGI, showing linear trend reductions of 40 % for EDGAR v6.0 and 36 % for GAINS and NGHGIs. The average yearly reduction trend was 2 % yr$^{-1}$ for all three data sources .


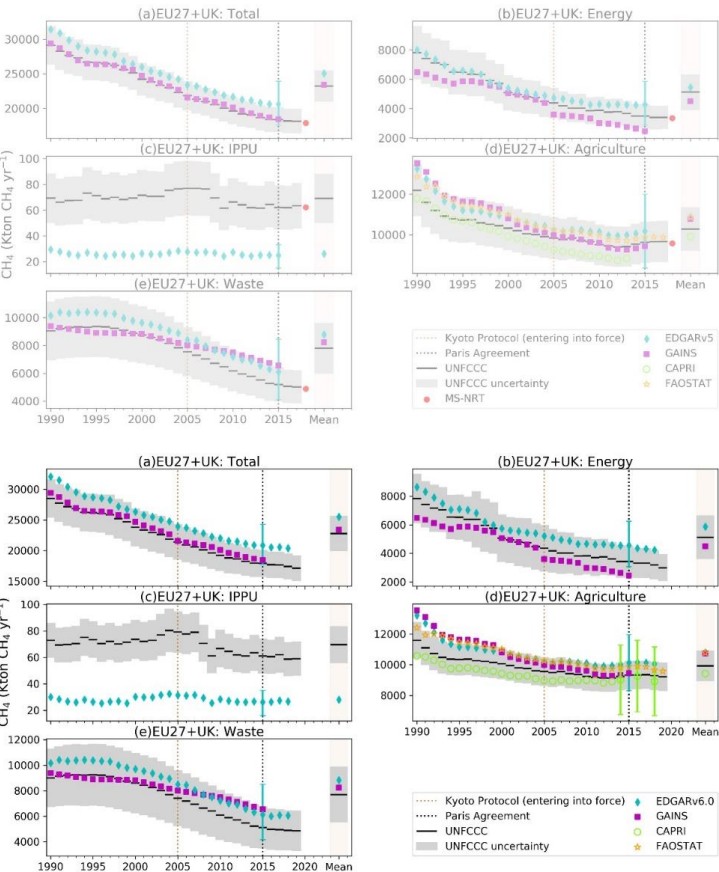

*Figure 3: Total annual anthropogenic $CH_4$ emissions (excluding LULUCF) for the EU27+UK over time. The top plot presents previous data synthesized in Petrescu et al., 2021a while bottom plot data synthesized by the current study:*





Sectoral time series of anthropogenic $CH_4$ emissions (excluding LULUCF) and their means are shown in Figures 3b,c,d and e. For the energy sector (Figure 3b), both EDGAR v6.0 and GAINS agree in trends with the NGHGI thanks to updated methodology that derives emission factors and accounts for country-specific information about associated petroleum gas generation and recovery, venting and flaring (Höglund-Isaksson, 2017). After 2005, GAINS

reports consistently lower emissions than UNFCCC due to a phase-down of hard coal production in Czech Republic, Germany, Poland and the UK, a decline in oil production in particular in the UK, and declining emission factors reflecting reduced leakage from gas distribution networks as old town gas networks are replaced. A difference in tiers is also one reason for the differences (Petrescu et al., 2020).

The consistently higher estimates (+6 % compared to the UNFCCC mean) of EDGAR v6.0 might be due to

the use of default emission factors for oil and gas production based on data from the US (Janssens-Maenhout et al. (2019). There are several other reasons that could be the cause for the differences, including the use of Tier 1 emission factors for coal mines, assumptions for material in the pipelines (in the case of gas transport) and the activity data). Also EDGAR v6.0, similar to the previous estimates from EDGARv5.0 uses the gas pipeline length as a proxy for the activity data however this may not be appropriate for the case of the official data, which could consider the total

amount of gas being transported or both methods according to the countries. Using pipeline length may overestimate the emissions because the pipeline is not always at 100% capacity thus a larger amount of methane is assumed to be leaked (Rutherford et al., 2021). For coal mining, emissions are a function of the different types of processes being modelled.

The IPPU sector (Figure 3c), which has only a small share of the total emissions, is not included in GAINS,

while EDGAR v6.0 estimates are less than half of the emissions reported by NGHGI 2021 in this sector. The discrepancy for this sector has negligible impact on discrepancy for the total $CH_4$ emission. However, we identified that the low bias of EDGAR v6.0 could be explained by fewer activities included in EDGAR v6.0 (e.g. missing solvent, electronics and other manufacturing goods) accounting for 5.5 % of the total IPPU emissions in 2015 reported to UNFCCC. The reason for the remaining difference could be explained by the allocation of emissions from auto-



producers[9] in EDGAR v6.0 to the Energy sector (following the 1996 IPCC guidelines), while in NGHGI they are reported under the IPPU sector (following the 2006 IPCC guidelines).

As CAPRI and FAOSTAT report only emissions from agriculture, they are included only in Fig. 3d. The data (EDGAR v6.0, GAINS, CAPRI and FAOSTAT) shows good agreement, with CAPRI at the lower range of emissions (Petrescu et al., 2020) and on average 3% lower than that of  NGHGI, and EDGAR v6.0 at the upper range. The reason

for EDGAR v6.0 having the highest estimate (contrary to Petrescu et al., 2020 where NGHGI were the highest and EDGAR v4.3.2 was the second highest) is likely due to the activity data updates in EDGAR v6.0 based on FAOSTAT values, compared to EDGAR v4.3.2. When looking at the time series mean, EDGAR v6.0, GAINS and FAOSTAT show 5 % higher emissions than that of NGHGI. The three BU estimates and NGHGI estimates show similar mean values likely due to the use of similar activity data and emission factors (EFs) (i.e. Figure 4 in Petrescu et al., 2020).

The updates submitted by CAPRI, for the years 2014, 2016 and 2018 match the NGHGI emission estimates and have uncertainties of 21 %. Compared to the previous version of CAPRI used in Petrescu et al., 2021a, the new runs report lower $CH_4$ emissions. Compared to previous results, in the last version some changes have been implemented in the last version (e.g. introduction of slope and altitude limits based on LUCAS[10], improved distribution of grazing livestock etc.). The main activity triggering the differences was the emissions from enteric fermentation. Statistical

information on most agricultural data required for the estimation of $CH_4$ and $N_2O$ emissions are not available at high spatial (regional) and temporal (annual since 1990) resolution. Therefore, the CAPRI model features a module that provides generic data at regional level (CAPREG) and additionally a module that also estimates feed distribution and GHG emissions at the required resolution for VERIFY (CAPINV). As indicated in an internal VERIFY report (Leip et al., 2019), the results of the CAPINV module were scrutinized and shortcomings were identified. These concern

mainly the distribution of feed, which is one of the most important parameter for $CH_4$ emissions from enteric fermentation, and manure excretion and subsequent GHG emissions. Other updates included addition of some regional input data (sources: FAOSTAT and EUROSTAT).

For the waste sector (Figure 3e) EDGAR v6.0 shows consistently higher estimates compared to the NGHGI data, while GAINS has higher emissions than the NGHGI after 2000 (mean 1990-2015 value 6% higher than NGHGI

emissions). The two inventories, EDGAR v6.0 in its 2020 update for landfills, and GAINS used an approach based on the decomposition of waste into different biodegradable streams, with the aim of applying the methodology described in the 2019 Refinement of the 2006 IPCC guidelines and the IPCC waste model (IPCC, 2019) using the First-Order-Decay (FOD) method. The main differences between the two datasets come from i) sources for total waste generated per person, ii) assumption for the fraction composted and iii) the oxidation. The two inventories may have

used different strategies to complete the waste database when inconsistencies were observed in the EUROSTAT database or in the waste emissions trends in NGHGI.

---

[9] auto-producers of electricity and heat: cogeneration by industries and companies for housing management (central heating and other services) (Olivier et al., 2017 PBL report)
[10] https://ec.europa.eu/eurostat/web/lucas

### 3.1.4. NGHGI estimates compared to atmospheric inversions

#### European estimates from regional inversions


Figure 4 compares TD regional estimates, NGHGI anthropogenic data for CH$_4$ emissions and natural BU emissions. Figure 4a presents TD estimates of total emissions (anthropogenic and natural) from Petrescu et al., 2021a while Fig. 4b shows the current study with updated total TD estimates. Figs. 4c and 4d show estimates of anthropogenic emissions (Petrescu et al., 2021a and current study) calculated by subtracting the total natural emissions

from the total TD emissions.

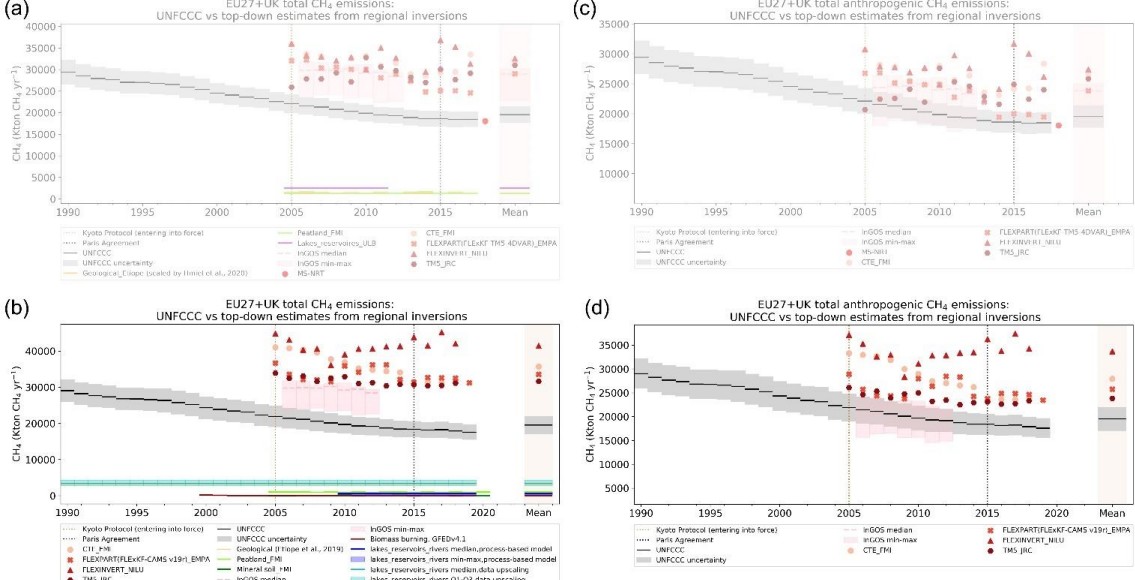

*Figure 4: a) and b) Comparison of total CH$_4$ emissions for EU27+UK from four top-down regional inversions with UNFCCC NGHGI (grey) data and two estimates for inland waters (lakes_rivers_reservoirsprocess-based models,*

*blue and upscaled emissions, cyan), peatlands and mineral soils(from JSBACH-HIMMELI, green), geological emissions (yellow) and biomass burning, (from GFEDv4.1, brown) as following: a) shows previous data from Petrescu et al., 2021a and b)current study; c)and d) comparison of anthropogenic CH$_4$ emissions from four top-down regional inversions with UNFCCC NGHGI (grey) data as following: c) previous data from Petrescu et al., 2021a and d) current study. Anthropogenic emissions from these inversions are obtained by removing natural emissions and biomass*

*burning from total TD CH$_4$ emissions shown in Figure 4a,b. UNFCCC NGHGI (2021) reported uncertainties computed with the error propagation method (95% confidence interval) were calculated for each year of the time series and represents the gap-filled harmonized Member States reported uncertainty for all sectors (including LULUCF). The time series mean was computed for the common period 2005-2018 between datasets (excluding InGOS).*




The TD estimates of European $CH_4$ emissions in Figure 4b use four European regional models for 2005-2018 period and an ensemble of five different inverse models (InGOS, Bergamaschi et al., 2015) for 2006-2012. For the 2005-2018 period (excluding InGOS), the four regional inversions give a total $CH_4$ emissions mean of 36 (32-42) Tg $CH_4$ yr$^{-1}$ compared to anthropogenic total of 20 Tg $CH_4$ yr$^{-1}$ in NGHGI (Fig. 4b). The large positive difference

between TD and NGHGI suggests a potentially significant contribution from BU natural sources (peatlands, geological sources, inland waters and biomass burning), which for the same period are estimated at 8 Tg $CH_4$ yr$^{-1}$. However, it needs to be emphasized that natural wetland emission estimates have large uncertainties and show large variability in the spatial (seasonal) distribution of $CH_4$ emissions but for Europe their inter-annual variability is not very strong (mean of 14 years from JSBACH-HIMMELI peatland emissions is 1.0 Tg $CH_4$ yr$^{-1}$). Overall, they do

represent an important source, and could dominate the budget assessments in some regions such as Northern Europe (Figure 1). That TD and NGHGIs diverged in terms of both emissions levels and trends is certainly significant and potentially has implications for bottom-up and NGHGI estimates of CH4 emissions, if the discrepancies cannot be explained by natural fluxes alone.

The geological emissions were recalculated based on the global grid model of Etiope et al (2019), using more

precise "activity" data for EU27+UK (details in Appendix A2): the emission resulted to be 3.3 Tg $CH_4$ yr$^{-1}$, i.e. 42 % of the total natural $CH_4$ emissions in EU27+UK. Geological emissions are an important component of the EU27+UK emissions budget, but their temporal variability is unknown (Etiope and Schwietzke, 2019) and so their impact on climate warming cannot be predicted.

The other natural sources of $CH_4$ contribute as following: natural emissions from inland waters (based on

Lauerwald et al., in prep, see Appendix A2) contribute 3.4 Tg $CH_4$ yr$^{-1}$, or 43 % of the total natural $CH_4$ emissions; peatlands and mineral soils (Raivonen et al. 2017 and Susiluoto et al. 2018) account for 1.0 Tg $CH_4$ yr$^{-1}$, i.e. 13.4 % of the total natural $CH_4$ emissions while biomass burning contributes only 0.6 % to the total $CH_4$ natural emissions. Similar to peatlands, inland water emissions also remain highly uncertain. The compilation of emission estimates lead to a total flux that is 3.3 Tg $CH_4$ yr$^{-1}$ (min 2.7 Tg $CH_4$ yr$^{-1}$ and max 4.3 Tg $CH_4$ yr$^{-1}$) and about five times larger than

the process-based model estimates for lakes+reservoirs and the spatially resolved flux for rivers (0.6 Tg $CH_4$ yr$^{-1}$ with min 0.2 and max 0.8 Tg $CH_4$ yr$^{-1}$) and about 25 % larger than the previous budget in Petrescu et al., 2021a (2.5 Tg $CH_4$ yr$^{-1}$), which ignored the contribution of rivers and relied on one observation-based estimate (extrapolation from late-summer data reported in Rinta et al. 2017) and four semi-empirical model assessments (Petrescu et al., 2021a). Interestingly, the new process-based estimate for natural lake+reservoirs $CH_4$ emissions matches well the data-driven

assessment by Rinta et al. (2017) for the late summer season, with a relative difference smaller than 5 %. The first approach synthesizes 15 average annual $CH_4$ emissions fluxes for Europe that were rescaled to a consistent set of inland water surface area (Lauerwald et al., in prep.) and corrected for the effect of seasonal ice cover.

Model results however also reveal a strong seasonal variability in $CH_4$ emissions, with much lower fluxes during winter. This finding partly explains why the spatio-temporally resolved model for rivers results lead to

significantly lower estimates than observation-based methods that do not capture well the temporal variability in lake $CH_4$ emissions.



According to the IPCC 2006 guidelines (IPCC, 2006) $CH_4$ emissions from wetlands are reported by the Member States to the NGHGI under the LULUCF sector and considered anthropogenic, if the wetlands in question are considered managed land. They are included in the total LULUCF values (Figure 1, 2, 4 and 6) and in 2019 reported $CH_4$ emissions from wetlands accounted for 0.1 Tg $CH_4$ yr$^{-1}$.

To quantify the anthropogenic $CH_4$ component in the European TD estimates, the BU peatland emissions from the regional JSBACH-HIMMELI model and those from geological, inland water sources and biomass burning were subtracted from the total TD emissions (Fig. 4d). It remains however uncertain to perform these corrections due to the prior inventory data allocation of emissions to different sectors (e.g. anthropogenic or natural) used in inversions, which can induce uncertainty of up to 100 % if for example an inventory allocates all emissions to natural emissions and the correction is made by subtracting the natural emissions. All regional inversion anthropogenic estimates are higher compared to the UNFCCC NGHGI (2021), mean of 28 Tg $CH_4$ yr$^{-1}$ from inversions compared to 20 Tg $CH_4$ yr$^{-1}$ from the NGHGIs.. Regarding trends, TD are stable except for CTE showing a linear decreasing trend up to 2015 followed by an increase over the next three years, while NGHGIs and BU trends are declining. From this attempt we find that not many of the inversions showed the clear decline reported by the NGHGIs. As NGHGI emissions are dominated by anthropogenic fluxes and decline by almost 30% compared to 1990, a similar decline was expected in the corrected anthropogenic inversions. Further investigation into how well the NGHGIs reflect reality or how well the TD estimates capture the trends is clearly needed. Currently, in the UK NIR (https://unfccc.int/documents/273439) the national inversion system produced similar recent UK $CH_4$ emission levels, but did not validate the large declining trend since 1990 that is estimated by the UK inventory.

***Spatial distribution of $CH_4$ emissions from regional inversions***

A novelty in this study is represented by the new top-down estimates of $CH_4$ fluxes were also calculated in this reporting period using the Community Inversion Framework (CIF) (Berchet et al. 2021). For $CH_4$ (Figure 5), inversions using three atmospheric transport models (or model variants) were performed with the CIF, there were: i) the regional non-hydrostatic Eulerian model, CHIMERE (Fortems-Cheiney et al., 2021) used by LSCE, ii) the Lagrangian particle dispersion model, FLEXPART used by EMPA (from hereon, FLEXPART-EMPA), and iii) FLEXPART used by NILU (from hereon, FLEXPART-NILU).

The spatial distribution of $CH_4$ fluxes are similar for the three inversions with higher emissions in the Netherlands and Belgium, western France and southern UK. However, FLEXPART-NILU inversions show some spurious areas of very low fluxes in Italy, Switzerland and southern France, which are presumably owing to the positive bias in the prior modelling mixing ratios at mountain sites, which will be corrected in future simulations. The patterns of differences, however, are quite different between the CHIMERE and the two FLEXPART inversions. All inversions find positive increments (posterior high than prior) over northern Netherlands, but FLEXPART-EMPA finds negative increments over southern Netherlands and both FLEXPART inversions find negative increments over northern Italy, which is not the case in CHIMERE (Figure 5, top). The total mean emissions for EU27+UK over 2006-2017 (Figure 5) were 26, 22 and 24 Tg $CH_4$ yr$^{-1}$, for CHIMERE, FLEXPART-NILU and FLEXPART-EMPA, respectively. FLEXPART-EMPA is the same model as used in the comparison shown in Figure 4



(FLEXPART(FLExKF-CAMSv19r)) but in those inversions the total mean emissions for EU27+UK were higher at 33 Tg yr⁻¹. This difference is likely owing to the different dataset used for determining the background mixing ratios and farther analysis is ongoing.

*Figure 5: Posterior CH₄ fluxes averaged over 2006-2017 (g CH₄ m⁻² yr⁻¹) from three regional inversions, CHIMERE*

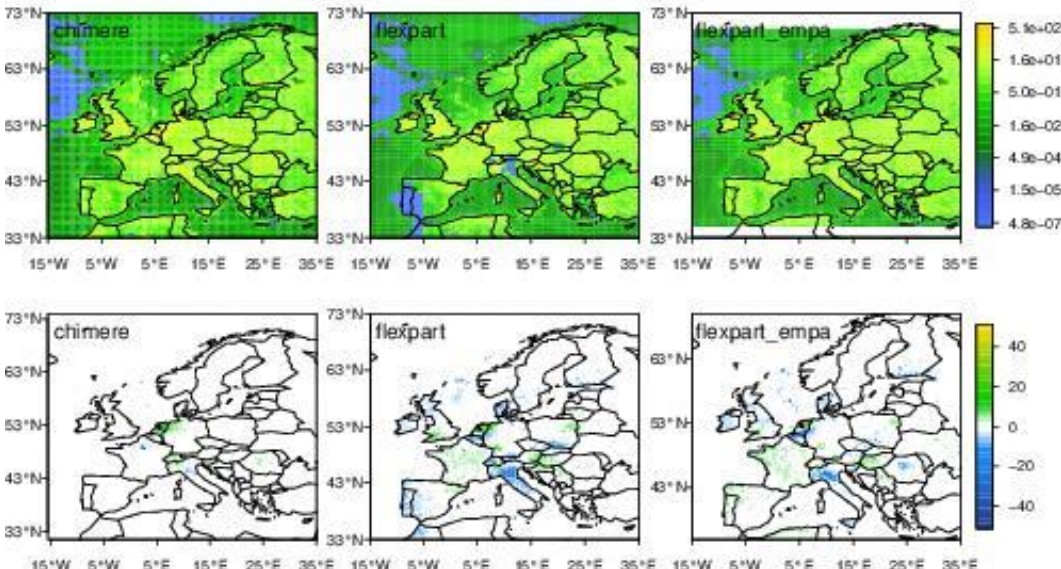

*(LSCE), FELXPART (NILU) and FLEXPART (EMPA) shown with a log base 2 color scale (top) and the flux*
*increments (g CH₄ m⁻² yr⁻¹) shown on a linear color scale (bottom).*

### *European estimates from global inversions*

Figures 6 compares TD global estimates, with NGHGI data and provides information about the wetland emissions from global wetland inversions (Saunois et al., 2020). Figure 6a presents TD estimates of total emissions
(anthropogenic and natural) from Petrescu et al., 2021a while Fig. 6b shows the current study with updated total TD estimates. Figs. 6c and 6d show estimates of anthropogenic emissions (Petrescu et al., 2021a and current study) calculated by subtracting the total natural emissions from the global total TD emissions.

The global inversion models were split according to the type of observations used, 11 of them using satellites (GOSAT) and 11 using surface stations (SURF). Each of these 22 global inversions provided as well wetlands
emissions used by the Global Methane Budget (Saunois et al., 2020) and are post-processed with prior ratios estimates for wetlands CH₄ emissions (Appendix B2, Table B2.4).

For the common period between datasets (2010-2016), the two ensembles of regional and global models give a total CH₄ emission mean (Figure 6a) of 23 Tg CH₄ yr⁻¹ (GOSAT) and 24 Tg CH₄ yr⁻¹ (SURF) for the EU27+UK compared to 19 Tg CH₄ yr⁻¹ ± 2.3 Tg CH₄ yr⁻¹ of NGHGI (Figure 6a). The mean of the natural wetland emissions from
the global inversions is 1.3 Tg CH₄ yr⁻¹ and partly explains the positive difference between total emissions from inversions and NGHGI anthropogenic emissions.



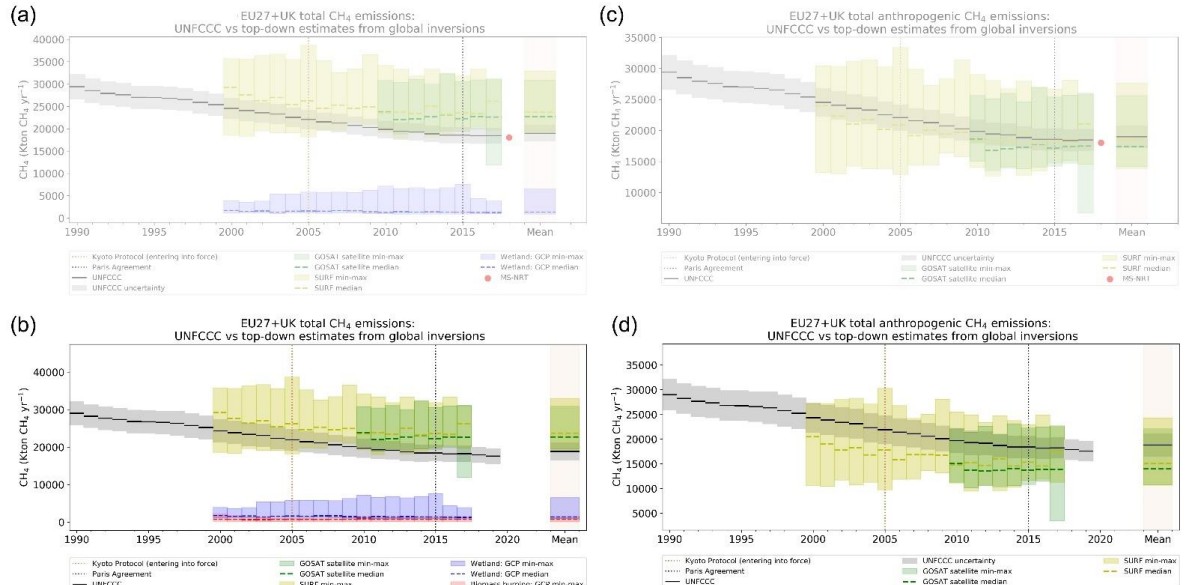

*Figure 6: a) and b) Total CH$_4$ emissions from TD global ensembles based on surface stations data (SURF) (yellow)*
*and satellite concentration observations (GOSAT) (green) from 22 global models compared with UNFCCC NGHGI*
*(grey) data (including LULUCF) as following: a) represents previous data from Petrescu et al., 2021a and b) the*
*current study; c) and d) Anthropogenic CH$_4$ emissions from top-down global inversions based on surface stations*
*(SURF) (yellow) and on satellite concentration observations (GOSAT) (green) from different estimates as following:*
*c) previous data from Petrescu et al., 2021a and d) the current study. Anthropogenic emissions from these inversions*
*were obtained by removing the sum of the natural emissions (global wetland GCP emissions (blue), the inland waters*
*and geological fluxes as shown in figure 4a) from the total estimates. The biomass burning emissions included in each*
*inversion results was removed as well. UNFCCC NGHGI (2021) Member States reported uncertainty computed with*
*the error propagation method (95% confidence interval) was gap-filled and provided for every year for all sectors*
*(including LULUCF). The time series mean was computed for the common period 2010-2016. Two out of 11 SURF*
*products (GELCA-SURF_NIES, TOMCAT-SURF_UOL) were not available for 2016.*

To quantify the European TD anthropogenic CH$_4$ component, the GCP inversions wetlands emissions and
those from geological, inland water sources and biomass burning emissions (reported by the global inversions) were
subtracted from the total CH$_4$ emissions (Fig. 6d).

For the 2010-2016 common period, the two ensembles of global models give an anthropogenic CH$_4$ emission
median (Figure 6b) of 13 Tg CH$_4$ yr$^{-1}$ with min and max values of 10 and 21 Tg CH$_4$ yr$^{-1}$) (GOSAT) and 14 Tg CH$_4$
yr$^{-1}$ with min and max values of 9 and 22 Tg CH$_4$ yr$^{-1}$) (SURF) compared to 19 ± 2.3 Tg CH$_4$ yr$^{-1}$ for NGHGI. The





TD ensemble that produced the closest anthropogenic estimate (Figure 6d) to the UNFCCC NGHGI (2021) is SURF, with the median of SURF inversions falling just below the uncertainty range of the NGHGI.

Between 2010-2016, total TD $CH_4$ emissions (Figure 6b) from the SURF and GOSAT ensemble decreased by 0.5% and 4.6%, respectively. For anthropogenic $CH_4$ emissions (Figure 6d), the SURF and GOSAT ensembles show a decrease of 1.1 % and 6.3%, respectively, compared to the 7.7 % decrease for the NGHGI.

### 3.1.5 $CH_4$ uncertainty reduction maps

Bergamaschi et al (2010) used TM5 4DVAR to analyze the sensitivity of the modelling system to observations, for further interpretation of the derived emissions, in particular in the context of verification of BU inventories. For this purpose, Bergamaschi et al., 2010 calculated uncertainty reduction maps, as a measure of the
sensitivity of the observational network used for the reference inversion. This reduction in uncertainty is calculated as the ratio between a posterior and a prior uncertainty with the formula (1-Δpost/Δprior), where Δpost represents the posterior uncertainties and Δprior the prior uncertainties of the inversion system. The same methodology was applied to two VERIFY regional inversions systems, CTE-$CH_4$ and FLExKF (Brunner et al., 2022).

The first inversion system, FLExKF, calculated the uncertainty reduction maps for $CH_4$ for the year 2018
with two different sets of observation stations (Figure 7). Maps of uncertainty reduction can be really informative and the results below (Figure 7) present the uncertainty reductions for two different sets of stations, which show the value of only considering ICOS sites (left figure) and when adding also other stations in the U.K. and Switzerland (right figure). However, the larger the prior uncertainties, the stronger potential for uncertainty reduction is, therefore given that the prior uncertainty varies, the uncertainty reduction is not a direct indication of the information provided by
observations.

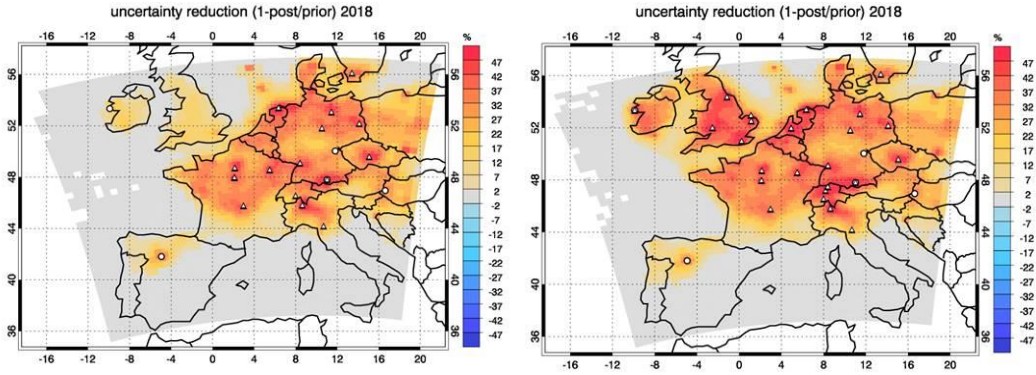

*Figure 7: FLExKF uncertainty reduction maps computed as (1-Δpost/Δprior) for the same year, 2018, but with two different sets of observation stations (white dots).*

The second inversion system, CTE-$CH_4$ (Tsuruta et al., 2017) calculated the uncertainty reduction maps from
surface inversions (SURF) for 2006 and 2018, as those used in Thompson et al., (2022), referred to here as



VERIFY_S4 ("inclusive" inversion) (Figure 8). The system included two sets of inversions with different observation sets assimilated. However, the degrees of freedom in the state of the system was low, and therefore, the uncertainty estimates may not differ much between the two. The data from CTE-CH$_4$ includes uncertainties (standard deviations) and fluxes for 2006 and 2018. The differences in the simulations are observation sets and underlying prior covariance structure. "VERIFY_S4" has the most observation sites assimilated. From the two panels of the Fig.8, higher uncertainty reductions are seen in 2018 compared to 2006 because in 2018 more measurements were available. The largest uncertainty reductions are observed in Central Europe (the Netherlands, Germany and Switzerland).

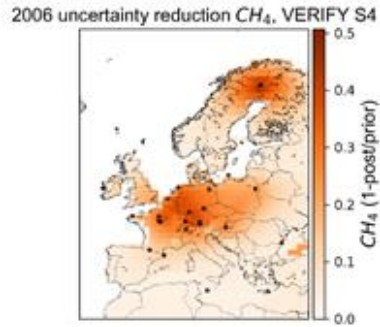
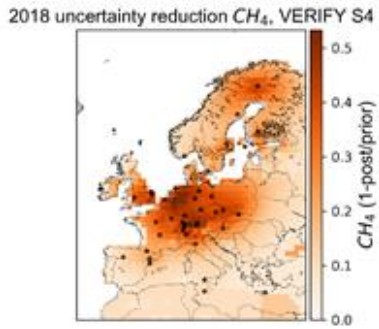

*Figure 8: VERIFY_S4 inversion run, uncertainty reduction maps computed as (1 − Δpost/Δprior) for 2006 (left) and 2018 (right) with different sets of observation stations.*

The differences between the two years are mostly due to changes in the amount of observational data, although additional observation stations in certain locations may produce only a limited reduction in uncertainty. This can occur if: i) uncertainty assigned to the observations (i.e. how much weight/trust we put on it) are comparatively high, ii) prior emissions and/or their uncertainties around the sites are simply very small, and therefore the inversion does not change fluxes much; and/or iii) the location is not very sensitive to emissions in the surrounding area (e.g. mountain sites) due to the atmospheric transport to the observation site. Generally, sites that contribute to a larger uncertainty reduction should be included in the inversions and located closer to emission sources and/or sink areas.

CTE-CH$_4$ was also used to estimate fluxes utilizing prior information from GOSAT data, for 2010 and 2017. Figure 9 presents the associated uncertainty reduction maps. Because of the different inversion system set-up (e.g. resolution, spatial correlation) compared to previous results, where prior data was coming from observation networks, it is difficult to conclude on the effects satellites have on posterior emissions from the two years. However, it is interesting to note how satellite data assimilation infers changes on a regional scale. Unlike surface stations, satellite data have more power to constrain northern European emissions than central European emissions.



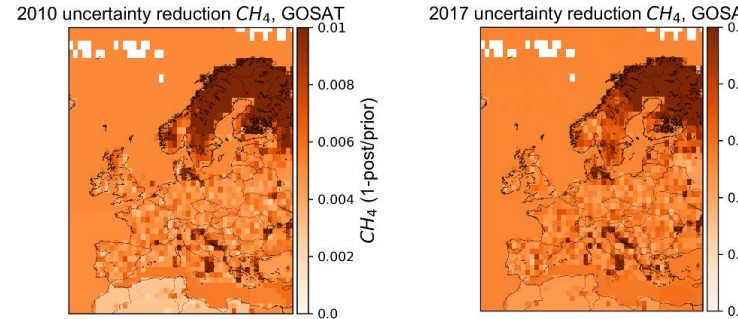

*F*igure 9: CTE-CH$_4$ GOSAT inversion run, uncertainty reduction maps computed as (1-Δpost/Δprior) for 2010 (left)
and 2017 (right).

## 3.2 Comparing N$_2$O emission estimates from different approaches

### 3.2.1. Estimates of European and regional total N$_2$O fluxes

Total N$_2$O fluxes from EU27+UK and five main regions in Europe are presented in a similar fashion as the
CH$_4$. Figure 10 summarizes the total N$_2$O fluxes from NGHGI 2021 (excluding LULUCF) for the base year 1990 as
well as mean annual emissions for the 2011-2015 and 2015-2019 five-year periods.

The total UNFCCC estimates that include emissions from all sectors are compared with the fluxes from
global datasets, BU models and TD inversions. Relative to 1990, N$_2$O emissions in 2019 decreased by a minimum of
26 % (Eastern Europe) up to a maximum of 46 % (Western Europe) and by 39 % for EU27+UK. At European level,
the emissions from BU estimates (anthropogenic NGHGI plus the sum of all natural, 991 kton N$_2$O) and TD total
(including natural) regional estimate (1443 kton N$_2$O) averaged over 2015-2019, roughly agree within the uncertainty
reported by UNFCCC (±59%). The TD uncertainty is represented as the variability in the model ensembles and
denotes the range between the minimum and maximum estimates within each model ensemble. There is significant
uncertainty in Northern Europe, where the TD average estimates indicate sources yet the ensemble ranges from a net
sink to a net source (Figure 10). The current observation network is sparse, which currently limits the capability of
inverse models to quantify N$_2$O emissions at country or regional scale.

For all other regions, the BU anthropogenic emissions agree in absolute values with the NGHGI given
uncertainties, though consistently higher estimates are produced by TD regional and global models. The difference is
still too high to be attributed to the sum of the natural emission, which ranges for all five regions in 2019 between a
minimum of 13 kton N$_2$O yr$^{-1}$ (Northern Europe) to a maximum of 113 kton N$_2$O yr$^{-1}$ (Southern Europe), while the
EU27+UK total natural emission is estimated at 178 kton N$_2$O yr$^{-1}$.

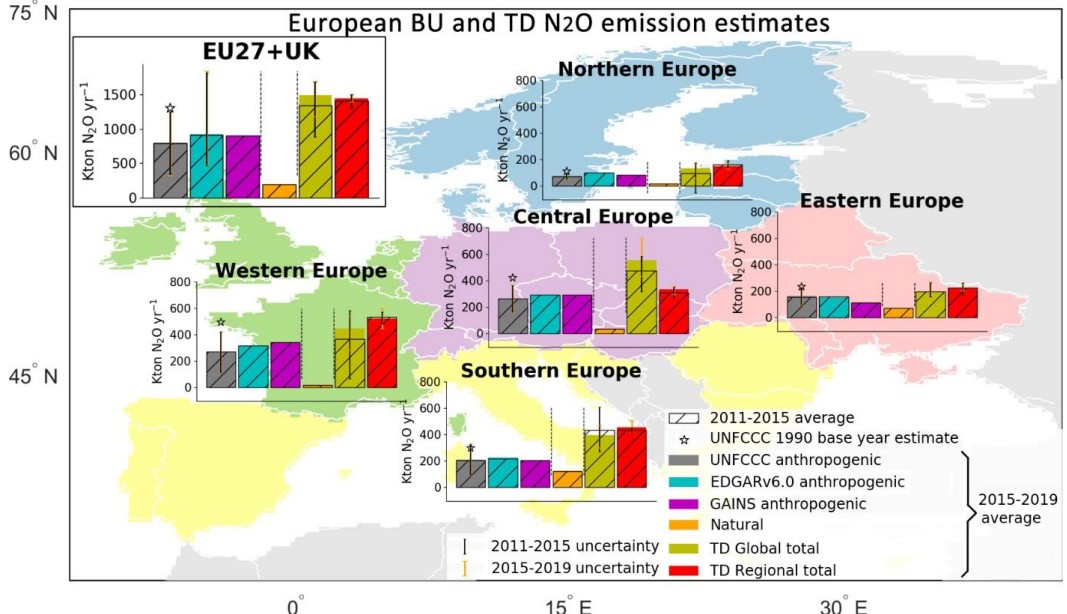

*Figure 10: Five-yearly means (2011-2015 and 2015-2019) in total $N_2O$ emission estimates (excluding LULUCF) for*
*EU27+UK and five European regions (Northern, Western Central, Southern and Eastern non-EU). The Eastern*
*European region does not include European Russia, Northern Europe includes Norway and Central Europe includes*
*Switzerland. The data are from the UNFCCC NGHGI (2021) submissions (grey), which are plotted with respective*
*base year 1990 (black star) estimates, two inventories (GAINS and EDGAR v6.0), natural unmanaged emissions*
*(lakes_rivers_reservoirs emissions from RECCAP2 and natural $N_2O$ from O-CN) and two inversion total estimates*
*(one regional European inversions (FLEXINVERT) and average of three global inverse models from $GN_2OB$, Tian et*
*al. 2020). The relative error on the UNFCCC value represents the NGHGI (2021) reported uncertainties computed*
*with the error propagation method (95% confidence interval) and gap-filled to provide respective estimates for each*
*year. (see Appendix A); For Easter Europe non-EU the uncertainty value of 42.3 % was calculated from the NIRs.*
*Northern Europe Tier 1 uncertainty for Norway was not available.*

### 3.2.2. NGHGI sectoral emissions and decadal changes

According to the UNFCCC NGHGI (2021) estimates for 2019, the EU27+UK emitted GHGs totaling 3.7 Gt
$CO_2e$ (including LULUCF, using a GWP 100, IPCC AR4) (Appendix B1, Figure B1b), of which $N_2O$ emissions
accounted for ~7 % (254 Mt $CO_2e$ or 854 kton $N_2O$ yr$^{-1}$) (Figure 11). France, Germany and UK together contributed
40 % of total $N_2O$ emissions (338 kton $N_2O$ yr$^{-1}$). For 2019, NGHGI reported anthropogenic emissions from the
EU27+UK for the four activity sectors (excluding LULUCF) (Table 1), to be 793 kton $N_2O$ yr$^{-1}$. Agricultural $N_2O$
emissions accounted for 79 % (± 72.5 %) of total EU27+UK emissions in 2019, followed by emissions from the
energy sector with 12 % (± 30 %).



Figure 11 shows anthropogenic $N_2O$ emissions from UNFCCC NGHGI (2021) and their changes from one
800 decade to the next, with the respective contributions from different sectors also illustrated.

 Between the 1990s and the 2000s, the net reduction of 17.9 % originates mainly from IPPU (-13.2 %), with
a smaller contribution from agriculture (-4.4 %). For the period between the 2000s and 2010-2019, the net reduction
of 15.4 % was again mainly attributed to the IPPU sector (14.1 %), despite very small increases from the LULUCF
(0.2 %) and waste sectors (0.2 %).

805 By 2019, emissions from the IPPU sector were only 36 kton $N_2O$ yr$^{-1}$, a 91 % decrease compared to 1990.
Although the IPPU sector contributes in only 4% to 2019 total $N_2O$ emissions, it is the sector associated with the
largest emissions reduction. IPPU sector emissions are mainly linked to the production of nitric acid (e.g. used in
fertilizer production) and adipic acid (e.g. used in nylon production). In the late 1990's and early 2000's the five
European adipic acid plants were equipped with efficient abatement technology, cutting emissions by 95-99 %, largely
810 through voluntary agreements of the companies. Much of the remaining IPPU emissions, from nitric acid plants, were
cut in a similar manner around 2010, a development that has been connected with the introduction of the European
Emission Trading System that made it economically attractive for companies to apply emission abatement
technologies (catalytic reduction of $N_2O$ in the flue gas) to reduce their emissions.



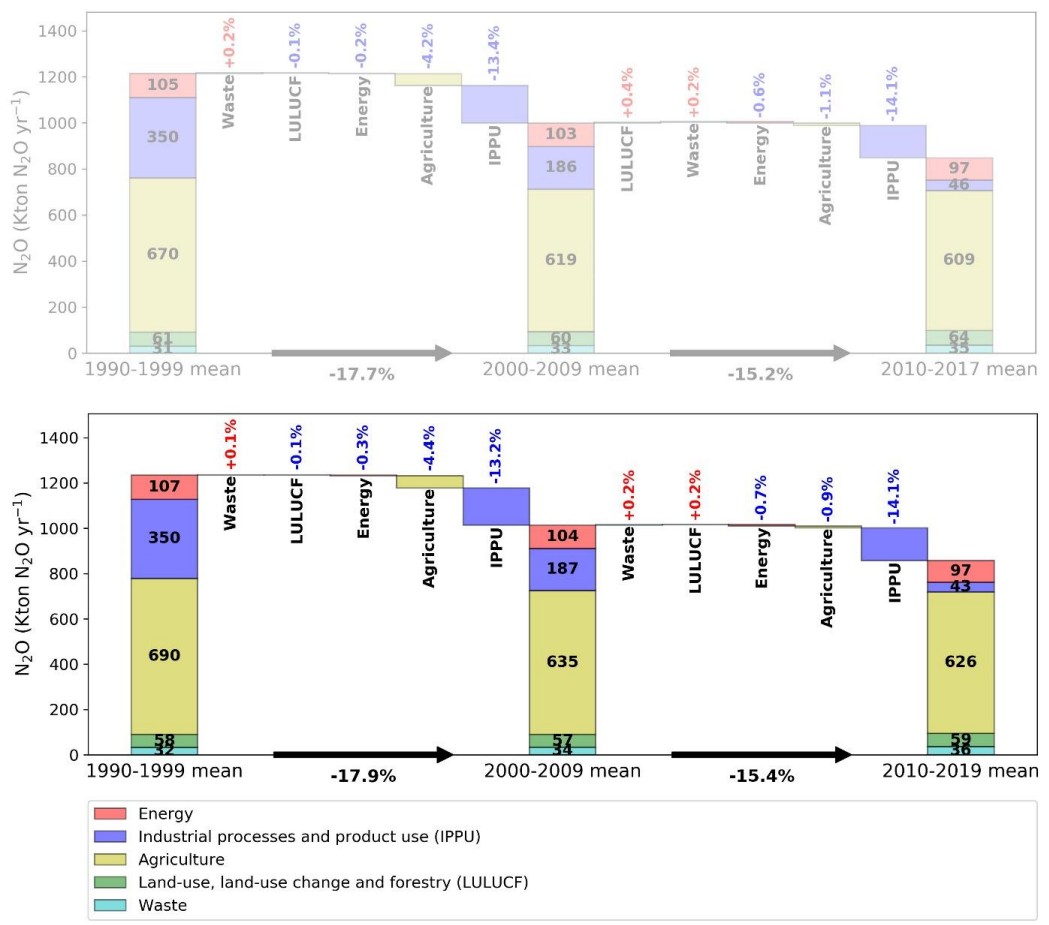

*Figure 11: The contribution of changes (%) in N₂O anthropogenic emissions in the five sectors to the overall change in decadal mean for the EU27+UK as reported to UNFCCC. The top plot shows the previous NGHGI data in Petrescu et al., 2021a and bottom plot depicts data from UNFCCC NGHGI (2021).The three stacked columns represent the average N₂O emissions from each sector during three periods (1990-1999, 2000-2009 and 2010-2019) and percentages represent the contribution of each sector to the total reduction percentages between periods.*

### 3.2.3. NGHGI estimates compared with bottom-up inventories

Figure 12 compares the six bottom-up inventories with UNFCCC NGHGI (2021) data, and shows that all of them are around the NGHGI estimates (Figure 12a), noting that GAINS only provides emissions every five years. The BU estimates show good agreement with one another and with the NGHGI estimates until 2005. After 2005 the slightly increasing trend is influenced by the IPPU (Figure 12c) and Waste (Figure 12e) sectors, with estimates of both EDGAR v6.0 and GAINS for total anthropogenic N₂O emissions in the year 2018 being 9 % and 13 %higher than the respective UNFCCC NGHGI (2021) estimates. Except for agriculture (Figure 12d), where four of the five models/inventories



show good match in absolute mean values with the NGHGI and over 1990-2018 and have similar linear trends of - 0.18, -0.17, -0.15 and -0.11 % yr$^{-1}$ in NGHGI, EDGAR v6.0, GAINS and FAOSTAT respectively, for the other sectors the trends differ. The match in agriculture trends reflects that the sources rely on the same basic activity data from FAOSTAT and follow the IPCC EF Tier 1 or 2 approach (Petrescu et al., 2020). However, the high reported uncertainty range from the NGHGIs contradicts the match of the BU estimate absolute values and represents an

important research question to be further investigated. In contrast, ECOSSE shows lower estimates because it does not use the FAO fertilizer application rate data base, but instead calculates ideal fertilizer application rates from the nitrogen demand of the crops. ECOSSE uses fertilizer data derived by Mueller et al. (2012) and simulates only for winter wheat. It is very likely that the assumed fertilizer application rates are lower than those used in FAO for the country specific average, which could explain the lower estimates. This means that it may severely under-estimate the

applied fertilizer amounts for some areas (e.g. Netherlands, Denmark or North-West Germany), and the results are more indicative of emissions under idealized fertilizer application rates. Additionally, as mentioned above, the model simulates only the direct emissions.

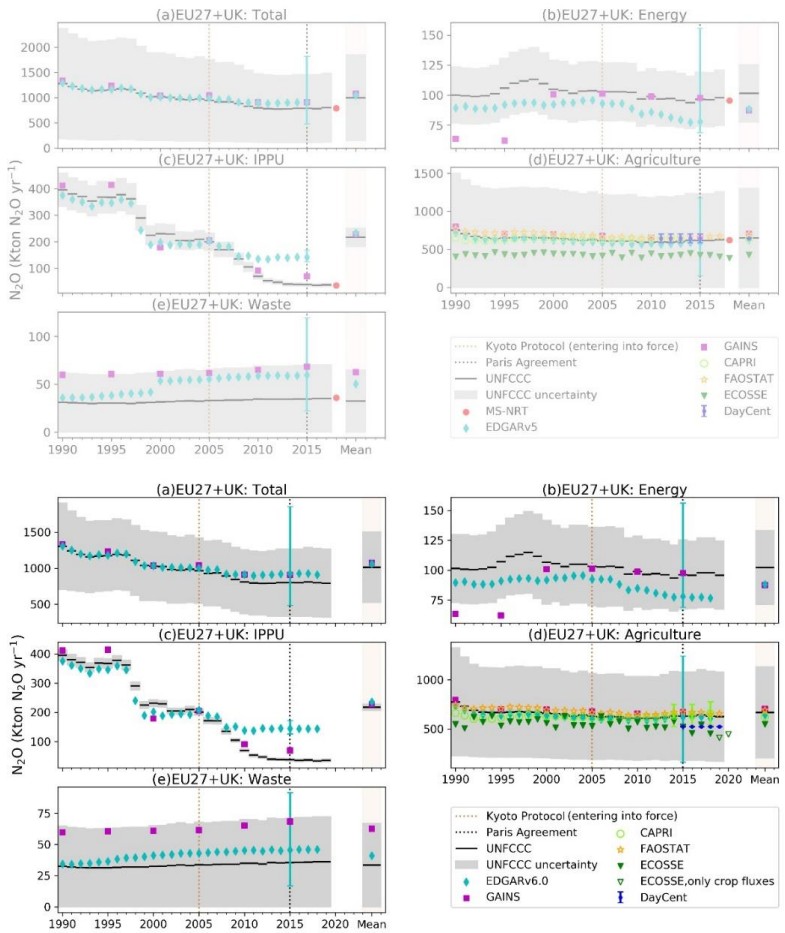


*Figure 12: a) Total annual anthropogenic $N_2O$ emissions (excluding LULUCF) for the EU27+UK over time. The top plot presents previous data synthesized in Petrescu et al., 2021a while the bottom plot depicts data synthesized by the current study: a) EU27+UK and total sectoral emissions from: b) Energy, c) IPPU, d) Agriculture, e) Waste from*

*UNFCCC NGHGI (2021) submissions compared to global BU inventories for agriculture (CAPRI, ECOSSE, FAOSTAT, DayCent) and all sectors excl. LULUCF (EDGAR v6.0, GAINS). CAPRI reports one value for Belgium and Luxembourg. The relative error on the UNFCCC value represents the UNFCCC NGHGI (2021) Member States reported uncertainties computed with the error propagation method (95% confidence interval) that were gap-filled and provided for every year. The uncertainty for EDGARv6.0 is the same as that of v5.0 and calculated for 2015 as min/max values for the total and each sector (Solazzo et al., 2021) and represents the 95 % confidence interval of a*

*lognormal distribution. CAPRI reports uncertainties for the last three years as following: 2014 and 2016 (17.6 %) and 2018 (17.8 %). The mean column represents the common overlapping period 1990-2018 between datasets Last years of the time series of the respective datasets are 2019 (UNFCCC and FAOSTAT), 2018 (EDGAR v6.0, CAPRI),*





*2015 (GAINS every five years), 2015-2019 (DayCent) and 2020 ECOSSE, with last two years reporting only crop emissions.*


In the NGHGI (2021) submissions, for 2019, the EU27+UK Tier 1 total uncertainty for the waste sector (based on the IPCC chapter 3 error propagation method described in detail by Petrescu et al., 2020) and the gap-filling method described in Appendix A, was 360 %. The sectoral activity responsible for this high uncertainty is the wastewater treatment and discharge (462%) and this remains one of the most uncertain sources of $N_2O$ having the highest emissions in the waste sector. Emissions are known to vary markedly in space and time even within a single wastewater treatment plant (Gruber et al., 2020), a fact that only recently has been properly accounted for in the inventory guidelines (IPCC, 2019a). However, the total emissions from the waste sector account for only 4.2 % of the total EU27+UK 2019 $N_2O$ emissions (excl. LULUCF).

### 3.3.4. NGHGI estimates compared to atmospheric inversions


Figure 13 compares inversion estimates of total $N_2O$ emissions, including natural, from regional (FLEXINVERT) and global (three models) $N_2O$ inversions with the UNFCCC NGHGI (2021) estimates. The min-max range of all inversions is within the 2-sigma uncertainty of NGHGI, with the median of global inversions being on average 42 % or 0.4 Tg $N_2O$ $yr^{-1}$ higher than that of NGHGI. Over the period 2005-2019, the regional FLEXINVERT is almost double that of UNFCCC NGHGI (2021). From the three global inversions, MIROC4-ACTM shows consistently higher estimates until 2019, when it registers a drop in the estimated emission level (similar to FLEXINVERT). Similar reduction of emissions are seen for 2003 and 2005. In all these years, Europe registered record breaking heatwaves. One plausible explanation for the low $N_2O$ is that high temperature accompanied with lesser soil moisture reduces $N_2O$ emission, as seen in the tropics (Patra et al., 2022).


The other two global inverse models, TOMCAT and CAMS-$N_2O$ register high estimates as well as very high variability. Regarding trends, FLEXINVERT shows a similar decreasing trend of 18 % over 2005-2019, compared to 16 % for UNFCCC NGHGI (2021). The global CAMS-$N_2O$ inversion agrees the best in its absolute mean value (1.0 Tg $N_2O$ $yr^{-1}$) with the NGHGI estimate (0.9 Tg $N_2O$ $yr^{-1}$) but not in its trend. In this updated synthesis, natural pre-industrial soil emissions of $N_2O$ from the O-CN model, were included, but these are not considered in NGHGI reporting, and therefore cannot explain the gap between inventories and TD estimates. In addition, the emission factors used in NGHGI reporting are regarded to be very uncertain (up to 300% for direct agricultural emissions) which, based on the comparison with TD estimates, could imply that inventories underestimate $N_2O$ emissions. The uncertainty reported by the NGHGIs in 2019 was 59% compared to 86% in 2017 (Petrescu et al., 2021a).

Regarding the natural $N_2O$ emissions, the median natural flux from inland waters, is very low (12.7 kton $N_2O$ $yr^{-1}$) and part of the inland water natural estimate is considered anthropogenic in Europe and is due to the leaching of N-fertilizers from agriculture. The anthropogenic share accounts for 66 % of the total inland waters emissions (Petrescu et al., 2021a). In the current study more natural $N_2O$ estimates have been added. The soil natural background emissions are estimated at 177 kton $N_2O$ $yr^{-1}$ averaged over 2005-2014 (the common overlapping period of all data


sources), while the biomass burning emissions account for only 1.6 kton $N_2O$ yr$^{-1}$for the same period. In the lower
plot, the NGHGIs uncertainty was recalculated to 56 % compared to 86% in Petrescu et al., (2021a) (upper plot).

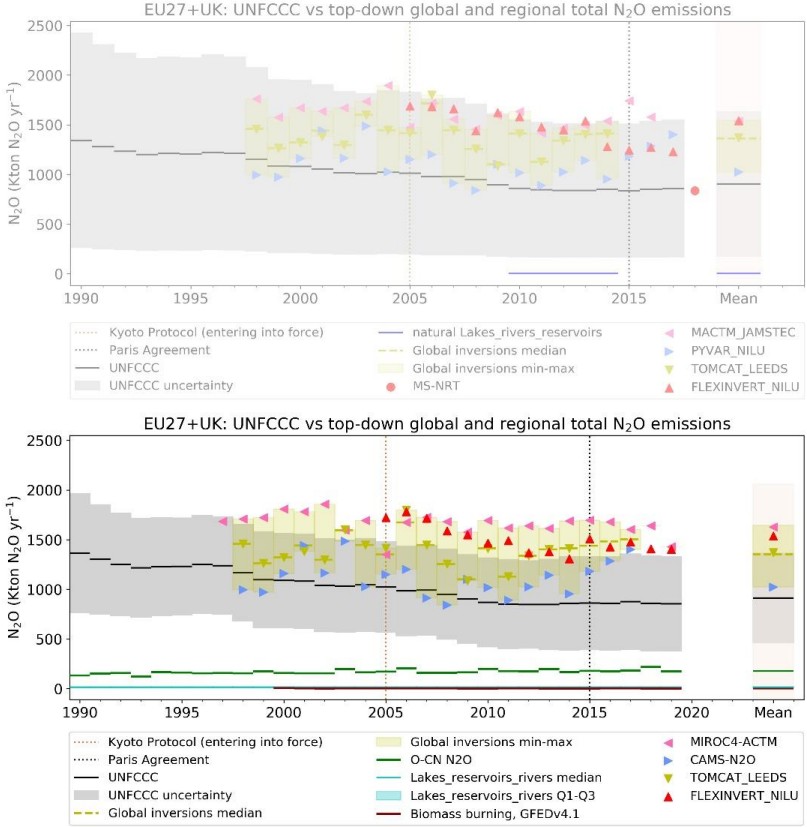

*Figure 13: Comparison of total $N_2O$ emissions for EU27+UK from one top-down regional inversion (FLEXINVERT)
and three inversions (TOMCAT, CAMS-$N_2O$ and MIROC4-ACTM) with UNFCCC NGHGI (grey) data and natural
$N_2O$ emissions (lakes_ reservoirs_ rivers from RECCAP2, natural pre-industrial soil emissions from the O-CN model*
*and biomass burning from GFEDv4.1) as following: the top plot shows the previous synthesized data in Petrescu et
al., 2021a and the bottom plot depicts data synthesized by the current study. The relative error on the UNFCCC value
represents the UNFCCC NGHGI (2021) Member States reported uncertainties computed with the error propagation
method (95% confidence interval) that were gap-filled and provided for every year (including LULUCF). Last years
of the time series of the respective datasets are 2014 (TOMCAT), 2017 CAMS-$N_2O$, 2019 (UNFCCC, FLEXINVERT*
*and MIROC4-ACTM). The mean column represents the common overlapping period (2005-2014) between datasets
MACTM-JAMSTEC is the same with MIROC4-ACTM and PYVAR_NILU with CAMS-$N_2O$.*

*Spatial distribution of N₂O emissions from regional inversions*

New top-down estimates of $N_2O$ fluxes were produced using the CIF (Berchet et al. 2021). For $N_2O$, inversions were performed by LSCE using three CHIMERE and by NILU using FLEXPART(v10.4). For the CIF inversions, the prior fluxes and observations of $N_2O$ were the same as those used in the Figure 13, and the background mixing ratios were
calculated using the CAMSv19r products for $N_2O$ (based on the global TM5-4DVAR assimilation run (Bergamaschi et al. 2018a, Rödenbeck et al. (2009))).

For $N_2O$, the FLEXPART inversion resulted in slightly larger fluxes over Europe compared to the CHIMERE inversion, especially over the Netherlands, Belgium, northern France and England. In these regions, FLEXPART also results in larger fluxes compared to the prior. FLEXPART estimated smaller fluxes compared to the prior and to
CHIMERE in northeastern Germany. CHIMERE, on the other hand remained close to the prior estimates. The total mean emission for EU27+UK for the period 2005-2018 was 1538 kton $N_2O$ and 1680 kton $N_2O$ yr$^{-1}$ for CHIMERE and FLEXPART, respectively, compared to the estimate of 1513 kton $N_2O$ yr$^{-1}$ from FLEXPART (Figure 13). Both inversions also found a decreasing trend over 2005-2018 with decreases of 157 kton $N_2O$ yr$^{-1}$ and 298 kton $N_2O$ yr$^{-1}$ per year for CHIMERE and FLEXPART, respectively, which was not seen in the prior estimates.


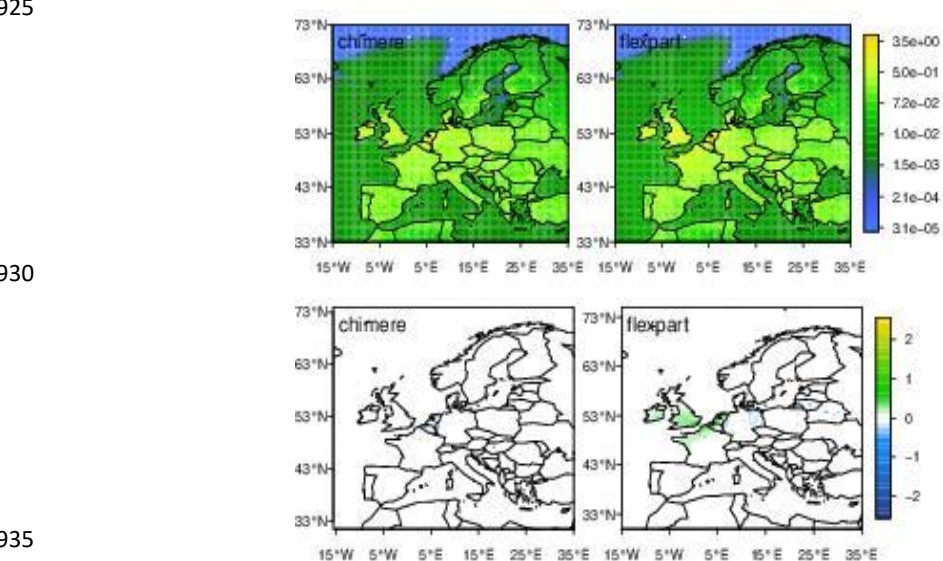



*Figure 14. Posterior N₂O fluxes averaged over 2005-2018 (g N₂O m⁻² yr⁻¹) from two regional inversions CHIMERE (LSCE) and FLEXPART (NILU) shown with a log base 2 color scale (top) and the flux increments (g N₂O m⁻² yr⁻¹) shown on a linear color scale.*

Similar to $CH_4$, the differences might be owing to the different dataset used for determining the background mixing ratios. Farther analysis is ongoing.





### 4. Data availability

Data files reported in this work which were used for calculations and figures are available for public download
at https://doi.org/10.5281/zenodo.6992472 (Petrescu et al., 2022). The data are reachable with one click (without the
need for entering login and password), with a second click to download the data, consistent with the two click access
principle for data published in ESSD (Carlson and Oda, 2018). The data and the DOI number are subject to future
updates and only refers to this version of the paper. The raw gridded data, according to the VERIFY consortium
governing document, will be made publicly available 12 months after its publication in ESSD.

### 5. Summary and concluding remarks

This study is an update of the first comprehensive synthesis of European $CH_4$ and $N_2O$ emission estimates
(Petrescu et al., (2021a), that compares total and sectoral European $CH_4$ and $N_2O$ from BU (anthropogenic and natural)
with TD estimates to assess their use for quality control and verification of UNFCCC NGHGI reporting. Using the
most recent data, differences between TD and BU estimates were compared and comparisons were made with the
previous synthesis, Petrescu et al., (2021a). Identification of source specific uncertainty is key in understanding these
differences and can lead to a reduction of the overall uncertainty in GHG inventories. Furthermore, the results have
been synthesized in a way that would be compatible with the methodological framework of the first 2023 Global
Stocktake of the Paris Agreement. Five-year means of $CH_4$ (Figure 1) and $N_2O$ (Figure 9) emissions for the periods
2011-2015 and 2015-2019 from the different BU and TD datasets have been calculated for the EU27+UK bloc, as
well as for five regions in Europe. These estimates are then compared with respective NGHGI emissions for the same
periods and the 1990 base year.

Inconsistencies between $CH_4$ BU estimates and NGHGI data at EU27+UK level (Figure 3), are mainly caused
by different methodologies in calculating emissions as highlighted in Petrescu et al. (2020, 2021a). Both BU
inventories and the NGHGI use similar activity data and, to varying extents, the default EFs reported in the IPCC 2006
guidelines meaning that the estimates are predestinated to agree rather well. Thus, the spread in all BU estimates may
not be indicative of the uncertainty. For global consistency purposes, EDGAR v6.0 and FAOSTAT mostly uses Tier
1 approaches in calculating emissions (and uncertainties), a fact which triggers differences with other data sources
using Tier 2 or 3 methods (GAINS for all sectors and CAPRI for agriculture). Within the UNFCCC reporting process,
the agriculture sector was the highest contributor to the $CH_4$ emissions, followed by energy and waste (Figure 1).

A reason for the small inconsistencies between datasets is the allocation of emissions to different sectors and
that some data sources use updated methods and emission factors from different versions of the IPCC guidelines (e.g.
1996 versus 2006 or 2019 Refinements to the 2006 (i.e. EDGAR v6.0)).

For $N_2O$ anthropogenic emissions, all BU data sources show good agreement with the UNFCCC NGHGI
(2021) data in both trends and means (Figure 12), agriculture remaining the largest emitter (e.g. soil emissions due to
fertilizer additions), within the reported uncertainties. As with $CH_4$, the different BU estimates share some common
elements such as activity data and emissions factors, and the agreement between estimates may not be relevant to the
underlying uncertainties.



An important improvement compared to Petrescu et al., 2021a, was the harmonization of UNFCCC Member States uncertainty estimates, which were gap-filled and calculated for each year of the time series. VERIFY interaction with the EU inventory team has helped improve the uncertainty estimations on the EU GHG emissions reported under UNFCCC. For both $CH_4$ and $N_2O$ the uncertainties reported by NGHGI are large and underline the need for further improvement in the inventories of these two GHGs.

Regarding the TD estimates, this analysis shows that comparison between $CH_4$ inversions estimates and NGHGIs is highly uncertain because of the large spread in the inversion results. Nevertheless, in contrast to BU methods, TD inversions inferred from atmospheric observations represent the most independent data against which $CH_4$ inventory totals can be compared. With anticipated improvements in atmospheric modelling and observations, as well as modelling of natural fluxes, TD inversions may arguably emerge as the most powerful tool for verifying emissions inventories for $CH_4$ and other GHGs.

As TD inversions do not fully distinguish between all emission sectors used by NGHGI and report either total emissions or a coarse sectorial partitioning, their comparison to NGHGI is only possible for total emissions. It is also necessary to make an adjustment for natural emissions, which are included in TD inversions but not reported by the NGHGIs. A future improvement for the natural $CH_4$ emissions are the consistent time series of measurements to make clear statements about how wetland and lake fluxes change over time. For lakes there is virtually no long-term monitoring, while for wetlands variability (e.g. area) is a key uncertainty but Fronzek et al., (2018) have shown that model ensembles work well in simulating highly uncertain variables. In general regional inversions show less spread than the global inversions as they used recent updates of transport models and simulate atmospheric transport at higher resolutions.

The global models use fewer observations for Europe compared to the included regional inversions, and thus are expected to have larger uncertainties for the European fluxes. In addition, the global models are at coarser resolution, and thus likely have larger model representation errors compared to the regional ones, which may contribute to further systematic uncertainty for the European fluxes. Currently, for Europe the regional TD total is considerably higher than the global estimates (Figure 1). If the regional TD estimate for whole EU27+UK including all sources and sinks is considered to be the best total estimate in place, and if the natural fluxes are assumed to have been accurately subtracted from the optimized net flux, then NGHGI and BU approaches may be underestimating total EU27+UK $CH_4$ emissions by approximately 20-30%. For $N_2O$, the TD estimates fall within the large range of the NGHGI uncertainties, and, in fact, the spread in the regional ensembles is much smaller than the inventory uncertainty range (Figure 9). Compared to Petrescu et al., 2021a, the natural emissions consisting of pre-industrial natural soil emissions and biomass burning emissions were included, however, for $N_2O$ natural emissions do not explain the 415 kton $N_2O$ difference between NGHGI (and BU) estimates and the average TD estimate. More research is thus needed to identify the source of discrepancies.

A key challenge for the inversion $CH_4$ community remains the separation of emissions in specific source sectors, as derived total emissions may also include natural emissions (or removals). In the case of $N_2O$ this won't be possible since the anthropogenic emissions due to agriculture are caused by a perturbation to microbial processes (i.e., nitrification and denitrification) and cannot be cleanly separated from "natural" emissions (defined as the level of



emission in the pre-industrial period, i.e., before perturbation by anthropogenic N-inputs). However, some TD quantification of industrial emissions should be possible in high-resolution inversions. In any case, TD inversions for estimating $N_2O$ emissions should mainly focus on the trends. Furthermore, the accuracy of derived emissions and the spatial scales at which emissions can be estimated depend on the quality and density of measurements and the quality

of the atmospheric models (Bergamaschi et al., 2018b). Significant further developments of the global observations system and the top-down methods would be required to support the implementation of the Paris Agreement.

The exercise of presenting uncertainty reduction maps illustrated the effect on uncertainty reductions with removing/adding ground-based observation stations. This is one of the elements informing policy makers on the need for further investing into a denser and efficient surface observation network used by inverse systems to calibrate their

estimates and better inform climate policy with respect to emissions verification. This might serve to monitor and build more accurate budget of country estimates as well as provide data for inferring subnational (e.g. city-scale) emissions.

This synthesis makes use and brings together state-of-the-art BU and TD estimates from different sources and compares these data with the official NGHGI estimates reported to UNFCCC. The exercise underlines the

uncertainties in the emissions of these important GHGs and illustrates the importance of regional consistent analyses and synthesis of available estimates for informing climate policy. Specifically, the approach demonstrated here could form the basis of the Multilateral Facilitative Consideration of Progress under the enhanced transparency framework of the Paris Agreement. The implementation of the Paris Agreement requires accurate quantification of GHG emissions in order to track the progress of all parties with their "Nationally Determined Contributions" and to assess

collective progress towards achieving the purpose of this Agreement and its long-term goals (GST). As this will be mainly achieved and built upon BU methodologies developed by the IPCC, we need to take into consideration the potential to quantify GHG emissions by using "top-down" methods ("inverse modelling") (Bergamaschi et al., 2018b). One advantage of the inverse estimate is that it provides total emission estimates inferred from atmospheric GHG measurements. Therefore, the capability to quantify anthropogenic emissions depends on the magnitude of natural

sources and sinks and the capability to quantify them and subtract them from the TD estimates.

As stated in the introduction, the main aim was to explore and discuss the issues causing differences between NGHGI, BU and TD approaches. Such an exercise can help to improve the different respective approaches and furthermore can inform the development of formal verification systems. Some differences in BU and NGHGI estimates were observed and were traced back to factors such as the variations activity data, emission factors and

sectoral allocation of emissions ($CH_4$). Nevertheless, BU and NGHGI estimates generally converged at the total emission level for the EU27+UK bloc and the five European regions. The overall agreement is generally due to similar sources of input activity data and emission factors (albeit with some aforementioned variations) and is not indicative of the true uncertainties in the respective $CH_4$ and $N_2O$ inventories. Indeed, NGHGI report $CH_4$ and $N_2O$ emissions with large uncertainties and, furthermore, NGHGI estimates generally diverged from the respective TD fluxes. Despite

the significant spread in the inversion estimates (due to e.g. use of different transport models and/or observation datasets, while priors might be the same (Table B2.4)), TD estimates were generally higher than NGHGI, even when accounting for the (albeit uncertain) natural fluxes.





The analysis done here generally compared estimates in terms of long-term trends and averages over five or more years, and thus provides a working example of how such syntheses could inform future Global Stocktakes under the Paris Agreement. A further step could involve analysis at finer temporal resolutions. While NGHGIs are reported at annual scales, analyzing emissions over intra-annual timescales, of great importance for $CH_4$ (wetland emission estimates have large uncertainties and show large variability in space and between seasons) and $N_2O$ (agricultural fertilizer application), may help to identify sector contributions to divergence between prior and posterior estimates at the annual/inter-annual scale. To do this, however requires expanded in-situ monitoring so that such dynamics can be better represented in the temporally-resolved prior estimates that feed into the top-down inversions.

## 6. Appendices

### Appendix A: Data sources, methodology and uncertainty descriptions

The country specific plots are found at: http://webportals.ipsl.jussieu.fr/VERIFY/FactSheets/ v1.27

**VERIFY project**

VERIFY's primary aim was to develop scientifically robust methods to assess the accuracy and potential biases in national inventories reported by the parties through an independent pre-operational framework. The main concept is to provide observation-based estimates of anthropogenic and natural GHG emissions and sinks as well as associated uncertainties. The proposed approach is based on the integration of atmospheric measurements, improved emission inventories, ecosystem data, and satellite observations, and on an understanding of processes controlling GHG fluxes (ecosystem models, GHG emission models).

Two complementary approaches relying on observational data-streams were combined in VERIFY to quantify GHG fluxes:

1) atmospheric GHG concentrations from satellites and ground-based networks (top-down atmospheric inversion models) and

2) bottom-up activity data (e.g. fuel use and emission factors) and ecosystem measurements (bottom-up models).

For $CH_4$ and $N_2O$, agricultural emissions were separated from fossil fuel and industrial emissions. Finally, trends in the budget of the three GHGs were analyzed in the context of NDC targets.

The objectives of VERIFY were:

**Objective 1**. Integrate the efforts between the research community, national inventory compilers, operational centres in Europe, and international organizations towards the definition of future international standards for the verification of GHG emissions and sinks based on independent observation.

**Objective 2**. Enhance the current observation and modelling ability to accurately and transparently quantify the sinks and sources of GHGs in the land-use sector for the tracking of land-based mitigation activities.

**Objective 3.** Develop new research approaches to monitor anthropogenic GHG emissions in support of the EU commitment to reduce its GHG emissions by 40% by 2030 compared to the year 1990.



**Objective 4.** Produce periodic scientific syntheses of observation-based GHG balance of EU countries and practical policy-oriented assessments of GHG emission trends, and apply these methodologies to other countries.
For more information on project team and products/results check https://verify.lsce.ipsl.fr/.

Table A1: *Country grouping used for reconciliation purposes between BU and TD estimates. The countries and*
1095 *groups of countries in italic are not directly used by this study but their figures and data is available on the VERIFY project web portal at:* http://webportals.ipsl.jussieu.fr/VERIFY/FactSheets/.

| Country name – geographical Europe | BU-ISO3 | Aggregation from TD-ISO3 |
|---|---|---|
| Luxembourg | LUX | |
| Belgium | BEL | BENELUX |
| Netherlands | NLD | BNL |
| Bulgaria | BGR | BGR |
| Switzerland | CHE | |
| *Lichtenstein* | *LIE* | *CHL* |
| Czech Republic | CZE | Former Czechoslovakia |
| Slovakia | SVK | CSK |
| Austria | AUT | AUT |
| Slovenia | SVN | North Adriatic countries |
| Croatia | HRV | NAC |
| Romania | ROU | ROU |
| Hungary | HUN | HUN |
| Estonia | EST | |
| Lithuania | LTU | Baltic countries |
| Latvia | LVA | BLT |
| Norway | NOR | NOR |
| Denmark | DNK | |
| Sweden | SWE | |
| Finland | FIN | DSF |
| Iceland | ISL | ISL |
| Malta | MLT | MLT |
| Cyprus | CYP | CYP |
| France (Corsica incl.) | FRA | FRA |
| *Monaco* | *MCO* | |
| *Andorra* | *AND* | |



| | | |
|---|---|---|
| Italy (Sardinia, Vatican incl.) | ITA | ITA |
| *San Marino* | *SMR* | |
| United Kingdom (Great Britain + N Ireland) | GBR | UK |
| *Isle of Man* | *IMN* | |
| Iceland | | |
| Ireland | IRL | IRL |
| Germany | DEU | DEU |
| Spain | ESP | IBERIA |
| Portugal | PRT | IBE |
| Greece | GRC | GRC |
| *Russia (European part)* | *RUS European* | |
| *Georgia* | *GEO* | *RUS European+GEO* |
| *Russian Federation* | *RUS* | *RUS* |
| Poland | POL | POL |
| *Turkey* | *TUR* | *TUR* |
| EU27+UK (Austria, Belgium, Bulgaria, Cyprus, Czech Republic, Germany, Denmark, Spain, Estonia, Finland, France, Greece, Croatia, Hungary, Ireland, Italy, Lithuania, Latvia, Luxembourg, Malta, Netherlands, Poland, Portugal, Romania, Slovakia, Slovenia, Sweden, United Kingdom) | AUT, BEL, BGR, CYP, CZE, DEU, DNK, ESP, EST, FIN, FRA, GRC, HRV, HUN, IRL. ITA, LTU, LVA, LUX, MLT, NDL, POL, PRT, ROU, SVN, SVK, SWE, GBR | E28 |
| Western Europe (Belgium, France, United Kingdom, Ireland, Luxembourg, Netherlands) | BEL, FRA, UK, IRL, LUX, NDL | WEE |
| Central Europe (Austria, Switzerland, Czech Republic, Germany, Hungary, Poland, Slovakia) | AUT, CHE, CZE, DEU, HUN, POL, SVK | CEE |
| Northern Europe (Denmark, Estonia, Finland, Lithuania, Latvia, Norway, Sweden) | DNK, EST, FIN, LTU, LVA, NOR, SWE | NOE |
| *South-Western Europe (Spain, Italy, Malta, Portugal)* | *ESP, ITA, MLT, PRT* | *SWN* |
| *South-Eastern Europe (all) (Albania, Bulgaria, Bosnia and Herzegovina, Cyprus, Georgia, Greece, Croatia, Macedonia, the* | *ALB, BGR, BIH, CYP, GEO, GRC, HRV, MKD, MNE,* | *SEE* |





| | | |
|---|---|---|
| *former Yugoslav, Montenegro, Romania, Serbia, Slovenia, Turkey)* | *ROU, SRB, SVN, TUR* | |
| *South-Eastern Europe (non-EU) (Albania, Bosnia and Herzegovina, Macedonia, the former Yugoslav, Georgia, Turkey, Montenegro, Serbia)* | *ALB, BIH, MKD, MNE, SRB, GEO, TUR* | *SEA* |
| *South-Eastern Europe (EU) (Bulgaria, Cyprus, Greece, Croatia, Romania, Slovenia)* | *BGR, CYP, GRC, HRV, ROU, SVN* | *SEZ* |
| *Southern Europe (all) (SOE) (Albania, Bulgaria, Bosnia and Herzegovina, Cyprus, Georgia, Greece, Croatia, Macedonia, the former Yugoslav, Montenegro, Romania, Serbia, Slovenia, Turkey, Italy, Malta, Portugal, Spain)* | *ALB, BGR, BIH, CYP, GEO, GRC, HRV, MKD, MNE, ROU, SRB, SVN, TUR, ITA, MLT, PRT, ESP* | *SOE* |
| *Southern Europe (non-EU) (SOY) Albania, Bosnia and Herzegovina, Georgia, Macedonia, the former Yugoslav, Montenegro, Serbia, Turkey)* | *ALB, BIH, GEO, MKD, MNE, SRB, TUR,* | *SOY* |
| Southern Europe (EU) (SOZ) (Bulgaria, Cyprus, Greece, Croatia, Romania, Slovenia, Italy, Malta, Portugal, Spain) | BGR, CYP, GRC, HRV, ROU, SVN, ITA, MLT, PRT, ESP | SOZ |
| Eastern Europe (non-EU) (Belarus, Moldova, Republic of, Russian Federation, Ukraine) | BLR, MDA, RUS, UKR | EAE |
| *EU-15 (Austria, Belgium, Germany, Denmark, Spain, Finland, France, United Kingdom, Greece, Ireland, Italy, Luxembourg, Netherlands, Portugal, Sweden)* | *AUT, BEL, DEU, DNK, ESP, FIN, FRA, GBR, GRC, IRL, ITA, LUX, NDL, PRT, SWE* | *E15* |
| *EU-27 (Austria, Belgium, Bulgaria, Cyprus, Czech Republic, Germany, Denmark, Spain, Estonia, Finland, France, Greece, Croatia, Hungary, Ireland, Italy, Lithuania, Latvia, Luxembourg, Malta, Netherlands, Poland, Portugal, Romania, Slovakia, Slovenia, Sweden)* | *AUT, BEL, BGR, CYP, CZE, DEU, DNK, ESP, EST, FIN, FRA, GRC, HRV, HUN, IRL. ITA, LTU, LVA, LUX, MLT, NDL, POL, PRT, ROU, SVN, SVK, SWE* | *E27* |
| *All Europe (Aaland Islands, Albania, Andorra, Austria, Belgium, Bulgaria, Bosnia and Herzegovina, Belarus, Switzerland, Cyprus, Czech Republic, Germany, Denmark, Spain, Estonia, Finland, France, Faroe Islands, United Kingdom, Guernsey, Greece, Croatia, Hungary, Isle of Man, Ireland, Iceland, Italy,* | *ALA, ALB, AND, AUT, BEL, BGR, BIH, BLR, CHE, CYP, CZE, DEU, DNK, ESP, EST, FIN, FRA, FRO, GBR, GGY, GRC,* | *EUR* |





| *Jersey, Liechtenstein, Lithuania, Luxembourg, Latvia, Moldova, Republic of, Macedonia, the former Yugoslav, Malta, Montenegro, Netherlands, Norway, Poland, Portugal, Romania, Russian Federation, Svalbard and Jan Mayen, San Marino, Serbia, Slovakia, Slovenia, Sweden, Turkey, Ukraine)* | *HRV, HUN, IMN, IRL, ISL, ITA, JEY, LIE, LTU, LUX, LVA, MDA, MKD, MLT, MNE, NDL, NOR, POL, PRT, ROU, RUS, SJM, SMR, SRB, SVK, SVN, SWE, TUR, UKR* | |

*countries highlighted in *italic* are not discussed in the current 2019 synthesis mostly because unavailability of NGHGI data (non-Annex I countries[11]) but are present on the web-portal: http://webportals.ipsl.jussieu.fr/VERIFY/FactSheets/. Results of Annex I countries (NOR, CHE, ISL) and non-EU EAE countries/groups are represented in Figures 1 and 9.

*Table A2: Main methodological changes (**in bold**) of current study with respect to Petrescu et al., 2020 and 2021a; n/a cells mean that there is no data available.*

| Publication year | Gas | Bottom-up anthropogenic CH₄ / N₂O emissions | | | Bottom-up natural CH₄ / N₂O emissions | Top-down CH₄ / N₂O emissions | | Uncertainty and other changes |
|---|---|---|---|---|---|---|---|---|
| | | Inventories | Global databases | Emission models | Emission models | Regional models | Global models | |
| Petrescu et al., 2020 | CH₄ | National emissions from UNFCCC (2018) 1990-2016<br><br>*AFOLU sector (Agriculture and LULUCF) EU28 data for four years (1990, 2005, 2010 and 2016)* | EGDAR v4.3.2 1990-2012<br><br>EDGAR FT2017 1990-2016<br><br>FAOSTAT 1990-2016<br><br>*Agriculture sector EU28 data for four years (1990, 2005, 2010 and last reported year)* | CAPRI 1990-2013<br><br>GAINS 1990-2015<br><br>*Agriculture sector EU28 data for four years (1990, 2005, 2010 and last reported year)* | Natural (wetlands) CH₄ emissions model ensemble GCP (2018) Poulter et al. (2017)<br><br>Time series 1990-2017 | n/a | n/a | UNFCCC (2018) uncertainty estimates for 2016 (error propagation 95% interval method)<br><br>EDGAR v.4.3.2. reports only for 2012 |

---

[11]Non-Annex I countries are mostly developing countries. The reporting to UNFCCC is implemented through national communications (NCs) and biennial update reports (BURs): https://unfccc.int/national-reports-from-non-annex-i-parties.

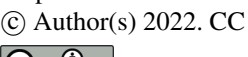



| | | | | | | | | |
|---|---|---|---|---|---|---|---|---|
| | N$_2$O | National emissions from UNFCCC (2018) 1990-2016  *Agriculture sector* EU28 data for four years (1990, 2005, 2010 and 2016) | EGDAR v4.3.2 1990-2012  EDGAR FT2017 1990-2016  FAOSTAT 1990-2016  *Agriculture sector* EU28 data for four years (1990, 2005, 2010 and last reported year) | CAPRI 1990-2013  GAINS 1990-2015  *Agriculture sector* EU28 data for four years (1990, 2005, 2010 and last reported year) | n/a | n/a | n/a | UNFCCC (2018) uncertainty estimates for 2016  EDGAR v.4.3.2. reports only for 2012 |
| Petrescu et al., 2021a | CH$_4$ | National emissions from UNFCCC (2019) 1990-2017  All UNFCCC sectors  EU27+UK time series  and 2018 MS-NRT estimate (EEA, 2019)  Regional EU27+UK totals (incl. NOR, CHE, UKR, MLD and BLR) | EGDAR v 5.0 1990-2015  FAOSTAT (only agriculture) 1990-2017  Anthropogenic EU27+UK time series (excl. LULUCF)  Regional EU27+UK totals (incl. NOR, CHE, UKR, MLD and BLR)  Excl. LULUCF | CAPRI 1990-2013  GAINS 1990-2015  Agriculture sector EU27+UK  Times series | Non-wetland inland waters  Average 2005-2011  Geological fluxes  Total pre-industrial era  JSBACH-HIMMELI  2005-2017 | Total CH$_4$ column  Time series 2005-2017:  FLEXPART - FLExKF  TM5-4DVAR  FLEXINVERT_NILU  CTE-FMI  InTEM-NAME Only for UK  InGOS inversions 2006-2012 | Anthropogenic and natural partitions  GCP-GCB 2019  2000-2017 | UNFCCC (2018) uncertainty estimates for 2016 (error propagation 95% interval method)  EDGAR v.4.3.2. reports only for 2015  For model ensembles reported as variability in extremes (min/max) |
| | N$_2$O | National emissions from UNFCCC (2019) 1990-2017  *All UNFCCC sectors* | EGDAR v 5.0 1990-2015 (excl. LULUCF)  FAOSTAT (only agriculture) 1990-2017 | *Agriculture* CAPRI 1990-2013  ECOSSE 1990-2018 | N$_2$O missions from lakes, rivers, reservoirs  Average 2010-2014 | Total N$_2$O column  Time series  FLEXINVERT_NILU  2005-2017 | Total N$_2$O column  Time series  GCP - GN$_2$OB 2019  CAMS-N$_2$O  TOMCAT | UNFCCC (2018) uncertainty estimates for 2016 (error propagation 95% interval method)  EDGAR v.4.3.2. reports only for 2015 |



| | | | | | | | MIROC4-ACTM<br><br>1998-2016 | For model ensembles reported as variability in extremes (min/max) |
|---|---|---|---|---|---|---|---|---|
| | | *EU27+UK* time series<br><br>and 2018 MS-NRT estimate (EEA, 2019)<br><br>*Regional EU27+UK totals* (incl. NOR, CHE, UKR, MLD and BLR) | *Anthropogenic* EU27+UK time series<br><br>Regional EU27+UK totals (incl. NOR, CHE, UKR, MLD and BLR)<br><br>Excl. LULUCF | | | | | |
| **Current study** | CH₄ | National emissions from UNFCCC (**2021**)<br><br>1990-**2019**<br><br>All UNFCCC sectors<br><br>EU27+UK time series<br><br>Regional EU27+UK totals (incl. NOR, CHE, UKR, MLD and BLR) excl. LULUCF<br><br>Two means:<br><br>2011-2015<br><br>**2015-2019** | **EGDAR v6.0**<br><br>1990-**2018**<br><br>FAOSTAT (only agriculture)<br><br>1990-**2020**<br><br>Anthropogenic EU27+UK time series<br><br>(excl. LULUCF)<br><br>Regional EU27+UK totals (incl. NOR, CHE, UKR, MLD and BLR)<br><br>excl. LULUCF | CAPRI<br><br>1990-**2014 and 2016 and 2018**<br><br>GAINS<br><br>1990-2015<br><br>Agriculture sector EU27+UK<br><br>Times series | **One median value from process-based models for non-wetland inland waters (lakes, rivers, reservoirs)**<br><br>**2010-2019**<br><br>**One median value from upscaled results in RECCAP2**<br><br>**1990-2019**<br><br>Geological fluxes<br><br>Total pre-industrial era **updates for EU27+UK**<br><br>JSBACH-HIMMELI **peatlands and mineral soils**<br><br>2005-**2020**<br><br>**Biomass burning GFEDv4.1 emissions** | Total CH₄ column<br><br>Time series 2005-2018:<br><br>FLEXPART - FLExKF<br><br>TM5-4DVAR<br><br>CTE-FMI<br><br>FLEXINVERT_NILU<br><br>1990-**2019**<br><br>InGOS inversions<br><br>2006-2012 **not included anymore in the mean**<br><br>**CHIMERE**<br><br>InTEM-NAME (only for UK plots on the VERIFY website) | Anthropogenic and natural partitions<br><br>GCP-GCB<br><br>2000-2017 | UNFCCC (**2021**) uncertainty estimates for **2019** (error propagation 95% interval method) **and for every year provided by UBA Vienna**<br><br>EDGAR reports only for 2015 values from v6.0<br><br>For model ensembles reported as variability in extremes (min/max)<br><br>**CAPRI uncertainties for 2014, 2016 and 2018** |
| | N₂O | National emissions from UNFCCC (**2021**)<br><br>1990-**2019** | **EGDAR v 6.0**<br><br>1990-**2018** | CAPRI<br><br>1990-**2014 and 2016 and 2018** | **One N₂O median value for emissions from lakes, rivers,** | Total N₂O column<br><br>Time series | Total N₂O column<br><br>Time series | UNFCCC (**2021**) uncertainty estimates for **2019** (error propagation 95% interval |





| | | FAOSTAT **(v2021)** (only agriculture) **1990-2020** | GAINS 1990-2015 | reservoirs **(RECCAP2)** Average **1990-2019** | FLEXINVERT_NILU 2005-**2019** | CAMS-N₂O 1998-2017 | method) **and for every year provided by UBA Vienna** |
|---|---|---|---|---|---|---|---|
| | All UNFCCC sectors EU27+UK time series Regional EU27+UK totals (incl. NOR, CHE, UKR, MLD and BLR) **excl. LULUCF** Two means: 2011-2015 **2015-2019** | Anthropogenic EU27+UK time series (excl. LULUCF) Regional EU27+UK totals (incl. NOR, CHE, UKR, MLD and BLR) excl. LULUCF | Agriculture sector EU27+UK Times series ECOSSE 1990-**2020** | **Natural N₂O pre-industrial emissions from O-CN model 1990-2020** **Biomass burning GFEDv4.1 emissions 2000-2019** | | TOMCAT 1998-2014 **MIROC4-ACTM 1997-2019 FELXINVERT 2005-2019 CHIMERE** | EDGAR reports only for 2015 values from v6.0 For model ensembles reported as variability in extremes (min/max) **CAPRI uncertainties for 2014, 2016 and 2018** |

## A1: Anthropogenic CH₄ emissions (sectors Energy, IPPU, Agriculture, LULUCF and Waste)

*Bottom-up CH₄ emissions estimates*

*UNFCCC NGHGI (2021)*

Under the UNFCCC and its Kyoto Protocol national greenhouse gas (GHG) inventories are the most important source of information to track progress and assess climate protection measures by countries. In order to build mutual trust in the reliability of GHG emission information provided, national GHG inventories are subject to standardized reporting requirements, which have been continuously developed by the Conference of the Parties (COP)[12]. The calculation methods for the estimation of greenhouse gases in the respective sectors is determined by the methods provided by the 2006 IPCC Guidelines for National Greenhouse Gas Inventories (IPCC, 2006). These Guidelines provide detailed methodological descriptions to estimate emissions and removals, as well as provide recommendations to collect the activity data needed. As a general overall requirement, the UNFCCC reporting guidelines stipulate that reporting under the Convention and the Kyoto Protocol follow the five key principles of transparency, accuracy, completeness, consistency and comparability (TACCC).

The reporting under UNFCCC should meet the TACCC principles. The three main GHGs are reported in time series from 1990 up to two years before the due date of the reporting. The reporting is strictly source category based and is done under the Common Reporting Format tables (CRF), downloadable from the UNFCCC official submission portal: https://unfccc.int/ghg-inventories-annex-i-parties/2021.

---

[12] The last revision has been made by COP 19 in 2013 (UNFCCC, 2013)



The UNFCCC NGHGI anthropogenic $CH_4$ and $N_2O$ emissions analyzed in this study include estimates from 4 key sectors for the EU27+UK: 1 Energy, 2 Industrial processes and product use (IPPU), 3 Agriculture and 5 Waste.

The methodological tiers a country applies depends on the source contribution to the national total (*Key Category* or not), on the national circumstances and the individual conditions of the land, which explains the variability of uncertainties among the sector itself as well as among EU countries. The LULUCF $CH_4$ and $N_2O$ emissions are very small but are included in some figures (see Table 1).

**Gap-filling harmonization procedure for NGHGI uncertainties**

The presented uncertainties in the emission levels of the individual countries and the EU27+UK bloc were calculated by using the methods and data used to compile the official GHG emission uncertainties that are reported by the EU under the UNFCCC (NIRs, 2022). The EU uncertainty analysis reported in the bloc's National Inventory Report (NIR) is based on country-level, Approach 1 uncertainty estimates (IPPC, 2006, Vol. 1, Chap. 3) that are reported by EU Member States, Iceland and United Kingdom under Article 7(1)(p) of Regulation (EU) 525/2013.

These country-level uncertainty estimates are typically reported at beginning of a submission cycle and are not always revised with updated CRF submissions later in the submission cycle. Furthermore, the compiled uncertainties of some countries are incomplete (e.g. uncertainties not estimated for LULUCF and/or indirect $CO_2$ emissions, certain subsector emissions are confidential) and the sector and gas resolution at which uncertainties are provided varies between the countries. The EU inventory team therefore implements a procedure to harmonise and gap-fill these

uncertainty estimates. A processing routine reads the individual country uncertainty files that are pre-formatted manually to assign consistent sector and gas labels to the respective estimates of emissions/removals and uncertainties. The uncertainty values are then aggregated to a common sector resolution, at which the emissions and removals reported in the uncertainty tables of the countries are then replaced with the respective values from the final CRF tables of the countries. Due to the issue of incompleteness mentioned above, the country-level data are then screened

to identify residual GHG emissions and removals for which no uncertainty estimates have been provided. Where sectors are partially complete, the residual net emission is quantified in $CO_2$ equivalents and incorporated. An uncertainty is then estimated, by calculating the overall sector uncertainty of the sources and sinks that were included in that country's reported uncertainties estimates and assigning this percentage average to the residual net emission. In cases where for certain sectors no uncertainties have been provided at all (e.g. indirect $CO_2$ emissions, LULUCF),

an average (median) sector uncertainty in percent is calculated from all the countries for which complete sectoral emissions and uncertainties were reported, and this average uncertainty is assigned to the country's sector GHG total reported in its final CRF tables.

The country-level uncertainties presented in this paper, have been compiled using this same processing routine and using the uncertainties and CRF data reported by the countries in the 2021 submission. However, here the

method has been expanded to gap-fill at the individual greenhouse gas level ($CH_4$ and $N_2O$ emissions only) rather than at the aggregate GHG level. Furthermore, the expanded method here assigns the sub-sectoral uncertainties to the emissions and removals of the entire time series (1990-2019), rather than just the base year and latest year of the respective time series. This allows uncertainties to be sensitive to the sub-sectoral contributions to sectoral and national total emissions, which of course change over time. For each year of the time series, uncertainties in the total and



sectoral CH$_4$ and N$_2$O emissions are calculated using Gaussian error propagation, by summing the respective sub-sectoral uncertainties (expressed in kton CH$_4$ and N$_2$O) in quadrature and assuming no error correlation. In contrast, for the EU27+UK bloc, uncertainties in the total and sectoral CH$_4$ and N$_2$O emissions were calculated to take into account error correlations between the respective country estimates at the subsector level. This was done by applying the same methods and assumptions described in the 2022 EU NIR (UNFCCC NIR, 2022). The subsector resolution

applied for gap-filling allows the routine to access respective data on emission factors from CRF Table *Summary 3* and apply correlation coefficients (r) when aggregating the uncertainties. For a given subsector, it is assumed that the errors of countries using default factors are completely correlated (r = 1), while errors of countries using country-specific factors are assumed uncorrelated (r = 0). For countries using a mix of default and country-specific factors at the given subsector level, it is assumed that these errors are partially correlated (r = 0.5) with one another and with the

errors of countries using the default factors only.

   Based on these correlation assumptions, the routine then aggregates CH$_4$ and N$_2$O emissions and uncertainties for the specified subsector resolution at the EU27+UK level. Uncertainties at sector total level are then aggregated from the subsector estimates assuming no correlation between subsectors. However, for countries reporting very coarse resolution estimates (e.g. total sector CH$_4$ and N$_2$O emissions) or where the sector has been partially or

completely gap-filled, it is assumed that these uncertainties are partially correlated (r = 0.5) with one another and with the other reported subsector level estimates. Level uncertainties on the total EU27+UK CH$_4$ and N$_2$O emissions (with and without LULUCF) are then aggregated from the sector estimates assuming no error correlation between sectors.

### *EDGAR v6.0*

The Emissions Database for Global Atmospheric Research (EDGAR) is an independent global emission inventory of greenhouse gases (GHG) and air pollutants developed by the Joint Research Centre of the European Commission (https://edgar.jrc.ec.europa.eu/index.php). The first edition of the Emissions Database for Global Atmospheric Research was published in 1995. The dataset now includes almost all sources of fossil CO$_2$ emissions, is updated annually, and reports data for 1970 to n-1. Estimates are provided by sector. Emissions are estimated fully

based on statistical data from 1970 till 2018 https://data.jrc.ec.europa.eu/dataset/97a67d67-c62e-4826-b873-9d972c4f670b. For complete description see Andrew (2020).

   **Uncertainties:** EDGAR uses emission factors (EFs) and activity data (AD) to estimate emissions. Both EFs and AD are uncertain to some degree, and when combined, their uncertainties need to be combined too. To estimate EDGAR's uncertainties (stemming from lack of knowledge of the true value of the EF and AD), the methodology

devised by IPCC (2006, Chapter 3) is adopted, that is the sum of squares of the uncertainty of the EF and AD (uncertainty of the product of two variables). A log-normal probability distribution function is assumed to avoid negative values, and uncertainties are reported as 95 % confidence interval according to IPCC (2006, chapter 3, equation 3.7). For emission uncertainty in the range 50 % to 230% a correction factor is adopted as suggested by Frey et al (2003) and IPCC (2006, chapter 3, equation 3.4). Uncertainties are published in Solazzo et al., 2021.




***CAPRI***

CAPRI is an economic, partial equilibrium model for the agricultural sector, focused on the EU (Britz and Witzke, 2014[13]; Weiss and Leip, 2012[14]). CAPRI stands for 'Common Agricultural Policy Regionalised Impact analysis', and the name hints at the main objective of the system: assessing the effect of CAP policy instruments not only at the EU or Member State level but at sub-national level too. The model is calibrated for the base year (currently 2012) and then baseline projections are built, allowing the ex-ante evaluation of agricultural policies and trade policies on production, income, markets, trade and the environment.

Among other environmental indicators, CAPRI simulates $CH_4$ emissions from agricultural production activities (enteric fermentation, manure management, rice cultivation, agricultural soils). Activity data is mainly based on FAOSTAT and EUROSTAT statistics and estimation of emissions follows IPCC 2006 methodologies, with a higher or lower level of detail depending on the importance of the emission source. Details on CAPRI methodology for emissions calculations is referenced in the Annex Table A1. For this study CAPRI updated three years: 2014, 2016 and 2018.

**Uncertainties** were calculated for the updated years, 2014, 2016 and 2018. The uncertainty of the spatial allocation of emissions for $CH_4$ and $N_2O$ has been calculated by taking into account the uncertainty of the spatial disaggregation and the uncertainty of the emission sources, assuming:

- the disaggregation having an uncertainty of 50% for $N_2O$ and 20% for $CH_4$.
- the emission processes have uncertainty of:
    - 50%: $N_2O$ soil processes;
    - 30%: $N_2O$ manure processes;
    - 30%: $CH_4$ manure and enteric, and
    - 10%: rice.

Then, for each cell, the uncertainty of the disaggregation and that of the process are combined as they are independent (sum of squares, (see Solazzo et al., 2021) and then the total uncertainty for the grid cell is aggregated using emission weighted sum of squares.

***GAINS***

Specific sectors and abatement technologies in GAINS vary by the specific emitted compound, with source sector definition and emission factors largely following the IPCC methodology at the Tier 1 or Tier 2 level. GAINS includes in general all anthropogenic emissions to air, but does not cover emissions from forest fires, savannah burning and land use / land use change. Emissions are estimated for 174 countries/regions, with the possibility to aggregate to a global emission estimate, and spanning a timeframe from 1990 to 2050 in five-year intervals. Activity drivers for macroeconomic development, energy supply and demand, and agricultural activities are entered externally, GAINS

---

[13] https://www.capri-model.org/docs/CAPRI_documentation.pdf
[14] https://www.sciencedirect.com/science/article/pii/S0167880911004415



extends with knowledge required to estimate "default" emissions (emissions occurring due to an economic activity without emission abatement) and emissions and costs of situations under emission control (see Amann et al., 2001).

The GAINS model covers all source sectors of anthropogenic methane ($CH_4$) emissions; agricultural sector emissions from livestock, rice cultivation and agricultural waste burning, energy sector emissions from upstream and downstream sources in fossil fuel extraction and use, and emissions from handling and treatment of solid waste and wastewater source sectors. A description of the modelling of $CH_4$ emissions in GAINS is presented in Höglund-Isaksson (2020). Generation of solid waste and the carbon content of wastewater are derived within the model in

consistency with the relevant macroeconomic scenario. The starting point for estimations of anthropogenic $CH_4$ is the methodology recommended in the IPCC (2006 and revision in 2019) guidelines, for most source sectors using country-specific information to allow for deriving country- and sector/technology- specific emission factors at a Tier 2 level. Consistent methodologies were further developed to estimate emissions from oil and gas systems (Höglund-Isaksson, 2017) and solid waste (Höglund-Isaksson, 2018; Gómez-Sanabria et al., 2018). Emission factors are specified in a

consistent manner across countries for given sets of technology and with past implementation of emission abatement measures reflected as changes in technology structures. The resulting emission estimates are well comparable across geographic and temporal scales. The GAINS approach to calculate waste emissions is developed in consistency with the First-Order-Decay method recommended by IPCC (2006 and 2019 revision), applying different decay periods when estimating emissions from flows of different types of organic waste, i.e., food & garden, paper, wood, textile

and other. Data on waste generation, composition and treatment are taken from EUROSTAT (2019) and complemented with national information from the UNFCCC (2019) Common Reporting Format tables on the amounts of waste diverted to landfills of various management levels and to treatment e.g., recycling, composting, biodigestion and incineration.

**Uncertainties:** Uncertainty is prevalent among many different dimensions both in the estimations of emissions,

abatement potentials and costs. When constructing global bottom-up emission inventories at a detailed country and source level, it is inevitable that some information gaps will be bridged using default assumptions. As it is difficult to speculate about how such sources of uncertainty affect resulting historical and future emission estimates, we instead address uncertainty in historical emissions by making comparisons to estimates by other publicly available and independently developed bottom-up inventories and various top-down estimates consistent with atmospheric

measurements and inverse model results. Although existing publicly available global bottom-up inventories adhere to the recommended guidelines of the IPCC (2006), the flexibility in these is large and results will depend on the availability and quality of gathered source information. There is accordingly a wide range of possible sources of uncertainty built into estimations in such comprehensive efforts. Having a pool of independently developed inventories, each with its own strengths and weaknesses, can improve the understanding of the scope for uncertainty,

in particular when compared against top-down atmospheric measurements.

### *FAOSTAT*

FAOSTAT: The Food and Agricultural Organization of the United Nations (FAO), provides $CH_4$ emissions totals or per gas/activity from agriculture and LULUCF available at: https://www.fao.org/faostat/en/#data/GT. The





FAOSTAT emissions database is computed following Tier 1 IPCC 2006 Guidelines for National GHG Inventories (http://www.ipcc-nggip.iges.or.jp/public/2006gl/index.html). Country reports to FAO on crops, livestock and agriculture use of fertilizers are the source of activity data. Geospatial data are the source of AD for the estimates from cultivation of organic soils, biomass and peat fires. GHG emissions are provided by country, regions and special groups, with global coverage, relative to the period 1961-present (with annual updates, currently to 2019) and with

projections for 2030 and 2050, expressed as $CO_2e$ for $CH_4$, by underlying agricultural emission sub-domain and by aggregate (agriculture total, agriculture total plus energy, agricultural soils). LULUCF emissions consist of $CH_4$ (methane) associated with biomass and peat fires. Comparison to the UNFCCC submissions is also provided.

**Uncertainties** were computed by Tubiello et al., 2013 but are not available in the FAOSTAT database.

### Top-down $CH_4$ emission estimates

*FLEXPART – FLExKF*

    FLExKF applies an Extended Kalman Filter (Brunner et al. 2012) in combination with backward Lagrangian transport simulations using the model FLEXPART (Stohl et al. 2005; Pisso et al. 2019). It optimizes surface-atmosphere fluxes by assimilating atmospheric observations in a sequential manner, which allows for an analytical solution for relatively large inversion problems (long time periods, number stations O(100)). Since model-observation

residuals typically follow a log-normal distribution, the method optimizes log-transformed emissions, which also guarantees a positive solution. Source-Receptor Matrices (Seibert and Frank, 2004) were computed at 0.25° x 0.25° resolution with FLEXPART driven by ECMWF Era Interim meteorological fields in the same way as for FlexInvert. Backward simulations were limited to 10 days prior to each observation and to the domain 15°W – 35°E, 30°N – 75°N. Fluxes were estimated for this domain on a monthly basis at 0.5° x 0.5° resolution. For the version used in this

study, FLExKF-CAMSv19r_EMPA, the background mole fraction was taken from CAMSv19r which is based on the global TM5-4DVAR assimilation run (Bergamaschi et al. 2018a) where the above domain was cut out following the two-step approach of Rödenbeck et al. (2009).

**Uncertainties:** The uncertainty in the posterior fluxes is composed of random and systematic errors. The random uncertainties are represented by the posterior error covariance matrix provided by the Kalman Filter, which combines

errors in the prior fluxes with errors in the observations and model representation. Systematic uncertainties primarily arise from systematic errors in modelled atmospheric transport and in background mole fractions, but also include aggregation errors, i.e. errors arising from the way the flux variables are discretized in space and time.

    *FLEXINVERT*

The FlexInvert framework is based on Bayesian statistics and optimizes surface-atmosphere fluxes using the maximum probability solution (Rodgers, 2000). Atmospheric transport is modelled using the Lagrangian model FLEXPART (Stohl et al., 2005; Pisso et al., 2019) run in the backwards time mode to generate a so-called Source-Receptor Matrix (SRM). The SRM describes the relationship between the change in mole fraction and the fluxes discretized in space and time (Seibert and Frank, 2004) and was calculated for 8 days prior to each observation. For

use in the inversions, FLEXPART was driven using ECWMF operational analysis wind fields. The state vector consisted of prior fluxes discretized on an irregular grid based on the SRMs (Thompson et al. 2014). This grid has





finer resolution (in this case the finest was 0.25°×0.25°) where the fluxes have a strong influence on the observations and coarser resolution where the influence is only weak (the coarsest was 2°×2°). The fluxes were solved at 10-days temporal resolution. The state vector also included scalars for the background contribution. The background mixing ratio, i.e., the contribution to the mixing ratio that is not accounted for in the 8-day SRMs, was estimated by coupling the termination points of backwards trajectories (modelled using virtual particles) to initial fields of methane simulated with the Lagrangian FLEXPART-CTM model, which was developed at Empa based on FLEXPART (Stohl et al., 2005; Pisso et al., 2019). In these simulations, we applied the data assimilation method described by Groot Zwaaftink et al. (2018) that constrains modelled fields with surface observations through nudging.

**Uncertainties:** The posterior fluxes are subject to systematic errors primarily from: 1) errors in the modelled atmospheric transport; 2) aggregation errors, i.e. errors arising from the way the flux variables are discretized in space and time; 3) errors in the background methane fields; and 4) the incomplete information from the observations and hence the dependence on the prior fluxes. In addition, there is, to a smaller extent, some error due to calibration offsets between observing instruments. Uncertainties in the observation space were inflated to take into account the model representation errors

### *InGOS and TM5-4DVAR*

The atmospheric models used within the European FP7 project InGOS (Integrated non-CO2 Greenhouse gas Observing System) are described by Bergamaschi et al., 2018a and Supplement (https://www.atmos-chem-phys.net/18/901/2018/acp-18-901-2018-supplement.pdf). The models include global Eulerian models with a zoom over Europe (TM5-4DVAR, TM5-CTE, LMDZ), regional Eulerian models (CHIMERE), and Lagrangian dispersion models (STILT,NAME,COMET). The horizontal resolutions over Europe are~1.0–1.2∘ (longitude)×~0.8–1.0∘ (latitude) for the global models (zoom) and ~0.17–0.56∘ (longitude)×~0.17–0.5∘ (longitude) for the regional models. Most models are driven by meteorological fields from the European Centre for Medium-Range Weather Forecasts (ECMWF) ERA-Interim reanalysis (Dee et al., 2011). In the case of STILT, the operational ECMWF analyses were used, while for NAME meteorological analyses of the Met Office Unified Model (UM) were employed. The regional models use boundary conditions (background $CH_4$ mole fractions) from inversions of the global models (STILT from TM3, COMET from TM5-4DVAR, CHIMERE from LMDZ) or estimate the boundary conditions in the inversions (NAME) using baseline observations at MaceHead as prior estimates. In the case of NAME and CHIMERE, the boundary conditions are further optimised in the inversion. The inverse modelling systems applied in this study use different inversion techniques. TM5-4DVAR, LMDZ, and TM3-STILT use 4DVAR variational techniques, which allow optimisation of emissions of individual grid cells. These 4DVAR techniques employ an adjoint model in order to iteratively minimise the cost function using a quasi-Newton (Gilbert and Lemaréchal, 1989) or conjugate gradient (Rödenbeck, 2005) algorithm. The NAME model applies a simulated annealing technique, a probabilistic technique for approximating the global minimum of the cost function. In CHIMERE and COMET, the inversions are performed analytically after reducing the number of parameters to be optimised by aggregating individual grid cells before the inversion. TM5-CTE applies an ensemble Kalman filter (EnKF) (Evensen, 2003), with a fixed-lag smoother (Peters et al., 2005).



**Uncertainty:** In general, the estimated model uncertainties depend on the type of station and for some models (TM5-4DVAR and NAME) also on the specific synoptic situation. In InGOS the uncertainty of the ensemble was calculated

as 1σ estimate. Bergamaschi et al. (2015) showed that the range of the derived total CH$_4$ emissions from north-western and eastern Europe using four different inverse modelling systems was considerably larger than the uncertainty estimates of the individual models because the latter typically use Bayes' theorem to calculate the reduction of assumed a prior emission uncertainties by assimilating measurements (propagating estimated observation and model errors to the estimated emissions). An ensemble of inverse models may provide more realistic overall uncertainty

estimates, since estimates of model errors are often based on strongly simplified assumptions and do not represent the total uncertainty.

### InTEM – NAME

The Inverse Technique for Emission Modelling (InTEM) (Arnold et al., 2018) uses the NAME (Numerical Atmospheric dispersion Modelling Environment) (Jones et al, 2007) atmospheric Lagrangian transport model. NAME

is driven by analysis 3-D meteorology from the UK Met Office Unified Model (Cullen, 1993). The horizontal and vertical resolution of the meteorology has improved over the modelled period from 40 km to 12 km (1.5 km over the UK). InTEM is a Bayesian system that minimises the mismatch between the model and the atmospheric observations given the constraints imposed by the observation and model uncertainties and prior information with its associated uncertainties. The direction (latitude and longitude) and altitude varying background concentration and observation

station bias are solved for within the inverse system along with the spatial distribution and magnitude of the emissions. The time-varying prior background concentration for the DECC network stations is derived from the MHD observations when they are very largely sensitive only to Northern Canada (Arnold et al., 2018). The prior bias (that can be positive or negative) for each station is set to zero with an uncertainty of 1 ppb. The observations from each station are assumed to have an exponentially decreasing 12-hr time correlation coefficient and, between stations, a

200 km spatial correlation coefficient. The observations are averaged into 2-hr periods. The uncertainty of the observations is derived from the variability of the observations within each 2-hr period. The modelling uncertainty for each 2-hr period at each station varies and is defined as the larger of; the median pollution events in that year at that station, or 16.5% of the magnitude of the pollution event. These values have been derived from analysis of the observations of methane at multiple heights at each station across the DECC network. Each inversion is repeated 24

times, each time 10% of the observations per year per station are randomly removed in 5-day intervals and the results and uncertainty averaged.

**Uncertainty:** This random removal of observations allows a greater exploration of the uncertainty, given the potential for some of the emission sources to be intermittent within the time-period of the inversion.

### CTE-CH$_4$ Europe, CTE-SURF and CTE-GOSAT

CarbonTracker Europe CH$_4$ (CTE-CH$_4$) (Tsuruta et al., 2017) applies an ensemble Kalman filter (Peters et al. 2005) in combination with the Eulerian transport model TM5 (Krol et al. 2005). It optimizes surface fluxes weekly , and assimilates atmospheric CH$_4$ observations. TM5 was run at 1° x 1° resolution over Europe and 6° x 4° resolution globally, constrained by 3-hourly ECMWF ERA-Interim meteorological data. The photochemical sink of CH$_4$ due to



tropospheric and stratospheric OH, and stratospheric Cl and O($^1$D) was pre-calculated based on Houweling et al.
(2014) and Brühl and Crutzen (1993) and not adjusted in the optimization scheme.

Three experiments were conducted, which differ in (1) sets of prior fluxes, (2) sets of assimilated observations, and (3) optimization resolution over the Northern Hemisphere. CTE-FMI uses sets of prior fluxes from LPX-Bern DYPTOP (Stocker et al., 2014) for biospheric, EDGAR v4.2 FT2010 (Janssens-Maenhout et al., 2013) for anthropogenic, GFED v4 (Giglio et al., 2013) for biomass-burning, Ito and Inatomi (2012) for termites and Tsuruta et
al. (2017) for ocean sources. CTE-SURF and CTE-GOSAT use sets of prior fluxes from Global Carbon Project (Saunois et al., 2020). CTE-FMI and CTE-SURF assimilated ground-based surface CH$_4$ observations, while CTE-GOSAT assimilated GOSAT XCH$_4$ retrievals from NIES v2.72. CTE-FMI optimized fluxes at 1° x 1° resolution over Northern Europe, northeast Russia and southeast Canada, 6° x 4° resolution over other parts of the Northern Hemisphere land, and region-wise (combined TransCom regions and soil-type) over the Southern Hemisphere and
ocean. CTE-SURF and CTE-GOSAT fluxes were optimized at 1° x 1° resolution over Europe and region-wise elsewhere globally.

**Uncertainty:** The prior uncertainty is assumed to be a Gaussian probability distribution function, where the error covariance matrix includes errors in prior fluxes, observations and transport model representations. The uncertainty for the prior fluxes were assumed to be 80 % of the fluxes over land and 20 % over ocean, with correlation between
grid cells or regions to be 100-500 km over land and 900 km over ocean. The uncertainty for observations and transport model representations vary between observations, with min. aggregated uncertainty to be 7.5 ppb for surface observations and 15 ppb for GOSAT data. The posterior uncertainty is calculated as standard deviation of the ensemble members, where the posterior error covariance matrix is driven by the ensemble Kalman filter.

***MIROC4-ACTM:***

The MIROC4-ACTM time dependent inversions solve for emissions from 53 regions for CH$_4$ and 84 regions for N$_2$O. The inversion framework is based on Bayesian statistics and optimizes surface-atmosphere fluxes using the maximum probability solution. Atmospheric transport is modelled using the JAMSTEC's Model for Interdisciplinary Research on climate, version 4 based atmospheric chemistry-transport model (MIROC4-ACTM) (Watanabe et al. 2008; Patra
et al. 2018). The Source-Receptor Matrix (SRM) is calculated by simulating unitary emissions from 53 or 84 basis regions, for which the fluxes are optimised. The SRM describes the relationship between the change in mole fraction at the measurement locations for the unitary basis region fluxes. The MIROC4-ACTM meteorology was nudged to the JMA 55-year reanalysis (JRA55) horizontal wind fields and temperature. The calculation of photo-chemical losses is performed online. The hydroxyl (OH) radical concentration for reaction with CH$_4$ vary monthly but without any
interannual variations. The simulated mole fractions for the total a priori fluxes are subtracted from the observed concentrations before running the inversion calculation (as in Patra et al., 2016 for CH$_4$ inversion). Both the inversion results are contributed to the GCP-CH$_4$ and GCP-N2O activities (Saunois et al., 2020; Thompson et al., 2019; Tian et al., 2020).

**Uncertainties:** The posterior fluxes are subject to systematic errors primarily from: 1) errors in the modelled
atmospheric transport; 2) aggregation errors, i.e. errors arising from the way the flux variables are discretized in space



(84 regions) and time (monthly-means); 3) errors in the background mole fractions (assumed to be a minor factor); and 4) the incomplete information from the sparse observational network and hence the dependence on the prior fluxes. In addition, there is, to a much smaller extent, some error due to calibration offsets between observing instruments, which is more pertinent for $N_2O$ than for other GHGs. We have validated model transport in troposphere using $SF_6$

for the inter-hemispheric exchange time, and the using $SF_6$ and $CO_2$ for the age of air in the stratosphere. The simulated $N_2O$ concentrations are also compared with aircraft measurements in the upper troposphere and lower stratosphere for evaluating the stratosphere-troposphere exchange rates. Comparisons with ACE-FTS vertical profiles in the stratosphere and mesosphere indicate good parameterisation of $N_2O$ loss by photolysis and chemical reactions, and thus the lifetime, which affect the global total $N_2O$ budgets. Random uncertainties are calculated by the inverse model

depending on the prior flux uncertainties and the observational data density and data uncertainty. Only 37 sites are used in the inversion and thus the reduction in priori flux uncertainties have been minimal. The net fluxes from the inversion from individual basis regions are less reliable compared to the anomalies in the estimated fluxes over a period of time.

***Global Carbon Project – Global Methane Budget (GMB)***

GMB uses an ensemble of 22 top-down global inversions for anthropogenic $CH_4$ emissions presented in Saunois et al. (2020) for the Global Methane Budget. These inversions were simulated by nine atmospheric inversion systems based on various chemistry transport models, differing in vertical and horizontal resolutions, meteorological forcing, advection and convection schemes, and boundary layer mixing. Surface-based inversions were performed

over the period 2000-2017 while satellite-based inversions cover the GOSAT data availability 2010-2017. The protocol established for these simulations was not stringent as the prior emission flux data set was not mandatory, and each group selected their constraining observations. More information can be found in Saunois et al. (2020) in particular in their Table 6 and S6.

**Uncertainties**: currently there are no uncertainties reported for the GMB models. This study uses the median and the

min/mas as uncertainty range estimation from the 22 models ensemble. In general uncertainties might be due to factors like: different transport models, physical parametrizations, prior fluxes, observation data sets etc.

## A2: Natural $CH_4$ emissions
### *Bottom-up $CH_4$ emissions estimates*

### *$CH_4$ emissions from inland waters*

The $CH_4$ estimate from inland waters represents a climatology of diffusive and ebullitive $CH_4$ emissions from rivers, lakes and reservoirs. It is based on two approaches. The first approach synthesizes average annual $CH_4$ emissions fluxes for Europe that were rescaled to a consistent set of inland water surface area (Lauerwald et al., in

prep.) and corrected for the effect of seasonal ice cover. To obtain fluxes for the EU27+UK domain, the median and first interquartile range values are scaled down by a factor 0.75 based on the surface area of the two domains. The second approach provides a spatially-resolved climatology of inland water fluxes at the resolution of 0.1°. The river



estimate relies on the 0.1°x0.1° river water surface area of Lauerwald et al. (2015) and 3 observation-based assessments (Bastviken et al., 2011; Stanley et al., 2016; Rosentreter et al., 2021) of mean $CH_4$ flux densities for

European rivers. Note the very large range encompassed by the 3 studies (0.07-0.66 Tg $CH_4$ yr$^{-1}$ for EU27+UK), reflecting high uncertainty in the assessment of the river $CH_4$ flux. The lake estimate not only resolves the spatial variability, but also the temporal variability in $CH_4$ emissions. To do so, the mechanistic-stochastic-modeling (MSM) approach of Maavara et al. (2017, 2019) and Lauerwald et al. (2019) was expanded to resolve the lake seasonal dynamics and the biogeochemical processes of the $CH_4$ and $O_2$ cycles occurring in the water column and sediments.

To constrain the lake physics, the Canadian Small Lake Model (CSLM) was used (MacKay, 2012). CSLM represents lake stratification and mixing events by simulating vertical temperature profiles, thermocline and light penetration depths, and lake ice dynamics. For carbon, the MSM simulates a lake-mean trophic state from the balance between Net Primary Production (limited by light and nutrient inputs from the watershed) and heterotrophic decomposition of organic matter. It was upgraded to simulate vertical profiles of $O_2$ and $CH_4$ by accounting for eddy-diffusive transport

and the set of consumption/production processes of the $O_2$-$CH_4$ cycles at the (sub)-daily resolution with a vertical resolution of less than one meter (Maisonnier et al., in prep.). In the sediment, net $CH_4$ production was split into diffusive and ebullitive pathways using an approach modified from Langenegger et al. (2019). The new process-based model was then applied to the European domain, using a lake clustering approach whereby within each grid of the simulation domain (2.5°x2.5°), lakes are binned into different classes as a function of the key drivers of $CH_4$ fluxes

that are lake-size and depth (Messager et al., 2016), and lake trophic status. Then, for each grid and each class, a representative simulation forced by high-resolution local climate forcings extracted from the lake sector of ISIMIP was performed. To carry out the spatial upscaling, the resulting diffusive and ebullitive areal $CH_4$ fluxes through the water-air interface were multiplied by the surface area of each lake class in each grid of the domain, extracted from the HydroLAKES database (Messager et al., 2016).


### *JSBACH-HIMMELI*

     The model framework, JSBACH-HIMMELI (Raivonen et al., 2017; Susiluoto et al., 2018) is used to estimate wetland and mineral soil emissions, and an empirical model is used to estimate the emissions from inland water bodies.

     JSBACH-HIMMELI is a combination of two models, JSBACH, that is the land-surface model of MPI-ESM

(Reick et al., 2013), and HIMMELI, that is a specific model for northern peatland emissions of $CH_4$ (Raivonen et al., 2017). HIMMELI (HelsinkI Model of MEthane buiLd-up and emIssion for peatlands) has been developed especially for estimating $CH_4$ production and transport in northern peatlands. It simulates both $CH_4$ and $CO_2$ fluxes and can be used as a module within different modelling environments (Raivonen et al., 2017; Susiluoto et al., 2018). HIMMELI is driven with soil temperature, water table depth, the leaf area index and anoxic respiration. These parameters are

provided to HIMMELI from JSBACH, which models hydrology, vegetation and soil carbon input from litter and root exudates. $CH_4$ emission and uptake of mineral soils are calculated applying the method by Spahni et al. (2011) based on soil moisture estimated by JSBACH.



The distribution of terrestrial vegetation types in JSBACH-HIMMELI is adopted from CORINE land cover data and from native JSBACH land cover for the areas that CORINE does not cover. The HIMMELI methane model

is applied for peatlands and the mineral soil approach for the rest. The map of inland water $CH_4$ emissions has been combined with JSBACH-HIMMELI land use map so that the map of inland waters is preserved and JSBACH grid-based fractions of different land use categories adjusted accordingly. In order to avoid double-counting the terrestrial $CH_4$ flux estimates have been normalized by the ratio of the two inland water body distributions.

**Uncertainties:** As in any process modeling the uncertainties of the bottom up modeling of $CH_4$ arise from three

primary sources: parameters, forcing data (including spatial and temporal resolution), and model structure. An important source of uncertainty in the case of terrestrial $CH_4$ flux modeling is the spatial distribution of peatlands.

The uncertainties of JSBACH-HIMMELI peatland emissions were estimated by comparing the annual totals of measured and simulated methane fluxes at five European observation sites. Two of the sites are located in Finnish Lapland, one in middle Sweden, one in southern Finland and one in Poland.

For the sensitivity of mineral soil fluxes Spahni et al. (2011) tested two soil moisture thresholds, 85% or 95% of water holding capacity, below which mineral soils were assumed to be only $CH_4$ sinks, above which sources. We used the higher value, 95% of water holding capacity. The uncertainty was estimated using $CH_4$ flux simulations of one year (2005). We did two new model runs, using moisture thresholds 95±15%, and derived the uncertainty from the resulting range in the annual emission sum.


### Geological fluxes

### Framework and previous works

Geological methane emissions to the atmosphere, including natural gas seepage in petroliferous sedimentary basins and geothermal exhalations, were estimated at the global scale by multiple authors, based on bottom-up and

top-down procedures (see review, including discussions on conflicting estimates, in Etiope and Schwietzke, 2019; Thornton et al. 2021 and references therein), and accounting for about 40-50 Tg $CH_4$ yr$^{-1}$.

For the European continent, a first geo-methane emission estimate was proposed by Simpson et al. (1999) with "a best guess" of 0.01 Tg yr$^{-1}$ and a speculative upper limit of 3 Tg yr$^{-1}$, only on the basis of an extrapolation of a few submarine emission data. At the time of the Simpson's work, very few data on geological methane fluxes on

land were available, and emission factors were basically unknown. Bottom-up emission estimates at European level including onshore seepage and geothermal exhalations were proposed ten years later, by Etiope (2009), on the basis of published regional emission estimates, suggesting around 3 Tg yr$^{-1}$ for geographic, onshore and offshore, Europe (including Azerbaijan; practically corresponding to present EU49).

Again ten years later, thanks to a wider data-set of $CH_4$ seepage flux from different geological environments

in different countries and global inventories of geo-$CH_4$ emission sites, a process-based model using statistically derived emission factors and activity (areas) was developed to derive a global grid map of geo-$CH_4$ emissions (Etiope et al. 2019). The global grid model, developed by ArcGIS at 1°x1° resolution, can be, in theory, used to derive (scale-up) geo-$CH_4$ emissions at continental or regional scale.





This exercise was done, for Europe (EU27+UK) by Petrescu et al. (2021a), obtaining a value of 8.8 Tg yr$^{-1}$.

The authors wished, however, to scale-down this value taking into account a global top-down estimate based on radiocarbon-free $CH_4$ in ice cores by Hmiel et al. (2020), who suggested, for the entire planet, 1.6 Tg yr$^{-1}$. This global estimate has more recently been contextualized and questioned by Thornton et al. (2021); in fact, the Hmiel et al. (2020) estimate has the same order of magnitude of emissions estimated by multiple authors for single local and regional seepage areas, so that the global emission must be considerable higher.

Petrescu et al. (2021) used however the upper limit of Hmiel et al. (2020), 5.4 Tg yr$^{-1}$, to scale-down the global gridded emission of Etiope et al. (2019), i.e. 37.4 Tg yr$^{-1}$, which is not the estimated global emission (43-50 Tg yr$^{-1}$, which included some factors that could not be considered in the grid model). From the 8.8 Tg yr$^{-1}$ (EU27 + UK), Petrescu et al. (2021) obtained then 1.3 Tg yr$^{-1}$ (for marine and land geological emissions) as follows:

$$8.8 \times 5.4/37.4 = 1.3 \text{ Tg } CH_4 \text{ yr}^{-1} \quad (1)$$

Besides the subjective use of the upper limit of Hmiel et al. (2020) and of the gridded (not the estimated global) value of Etiope et al. (2019), it is important to note that the initial gridded value derived for EU27+UK, 8.8 yr$^{-1}$, is affected by the relatively low precision, at the European scale, of the input data used for the global grid model. The global model was, in fact, developed on the basis of a global, large scale distribution of geological factors (for example the area of petroleum fields which determines the microseepage area), which lack the necessary precision for

lower (continental and country) scale application. The main purpose of the global gridding was to offer a global spatial distribution of geo-$CH_4$ sources, with emission potential and methane isotopic values; it could not provide locally precise geo-$CH_4$ emissions because the datasets, developed for gridding purposes, was not complete and did not contain all the information necessary for improving previous estimates. A refinement of bottom-up estimates was possible only for mud volcanoes and microseepage because their gridding implied a more careful assessment of the

spatial distribution and emission factors.

### Re-assessment of geo-CH₄ emissions in Europe

For the current study,. Etiope and Ciotoli applied the global grid model of Etiope et al. (2019) using more detailed input data for Europe, with reference to the potential area of microseepage (activity) derivable by a more precise

estimate of the continental oil-gas field area. They used the same microseepage emission factors statistically derived on global scale. For the other categories of geo-$CH_4$ sources (mud volcanoes, onshore seeps, submarine seepage and geothermal manifestations) they applied European country masks from VERIFY (relative to EU27+UK and EU49) for the calculation of the onshore emission, and related EEZ (Exclusive Economic Zone) for the calculation of the sub-marine seepage to derive the global emission grid.

In the global microseepage emission model, the area where microseepage can potentially exist (named EMA - effective microseepage area) was estimated taking into account the distribution of microseepage observed by direct measurements in several oil-gas fields: statistical analysis of available data (summarized in Etiope et al. 2019) suggested that microseepage fluxes occur in about 57% of the petroleum field area (PFA). In theory, microseepage is expected also in the regions where seepage phenomena, manifested by macro-seeps (oil and gas seeps), exist. In Etiope





et al. (2019) the global PFA was derived from Petrodata (Lujala et al. 2007), based on an oil-gas field digital map of
        USGS (Pawlewicz et al. 1997), which considered for each oil-gas field center point a buffer area of 30 km radius. The
        reason for this radius was not explained in Lujala et al. (2007).

        In this work, the PFA of Europe (EU49) was derived by digitising all petroleum field center points from the
USGS map (Pawlewicz et al. 1997). The microseepage $CH_4$ emission related to EMA is then derived by using the
same emission factors (four levels of microseepage; Table 2) established on global scale in Etiope et al. (2019). The
results of the gridding, integrating activity (EMA) and emission factors, for EU49 are reported in Table A2.1.

*Table A2.1: Results of microseepage gridding (0.05°x0.05°) for EU49*

|  | N. cell | Area (km$^2$) | MS (t km$^{-2}$ yr$^{-1}$) | Total output (t yr$^{-1}$) |
|---|---|---|---|---|
| Gridded EMA | 9457 | 199,703 |  | 2,985,570 |
| Gridded Level 1 | 6843 | 143,307 | 0.474 | 67,927 |
| Gridded Level 2 | 1899 | 40,094 | 11.366 | 455,708 |
| Gridded Level 3 | 134 | 2959 | 40.15 | 118,804 |
| Gridded Level 4 | 581 | 13,008 | 180.13 | 2,343,131 |

*MS: Microseepage emission factor statistically derived as described in Etiope et al. (2019).*

The EMA and related microseepage emission for EU27+UK are derived by applying the related mask on
EU49, resulting in:

        EU27+UK EMA= 177,439 km$^2$ and microseepage EU27+UK = 2,161,060 t yr$^{-1}$.

The total microseepage emission is quite sensitive to the activity (area); This estimate can be improved by increasing
the number of measurements worldwide and related spatial (activity) and emission factor statistics.

For the other geological sources categories, i.e., onshore seeps (OS; including mud volcanoes), geothermal
manifestations (GM) and submarine seepage (SS), we applied the masks of the European territories (for EU27+UK
and EU49, and the EEZ for the marine areas) on the global 1°x1° emission grid of Etiope et al. (2019). The result is
summarised in Table A2.2.

*Table A2.2: Gridded European geo-CH$_4$ emissions (t yr$^{-1}$).*

|  | Microseepage (MS) | Onshore seeps (OS) | Geothermal (GM) | Submarine Seeps (SS) | TOTAL |
|---|---|---|---|---|---|
| EU49 | 2,985,570 | 2,162,539 | 404,205 | 1,653,049 | 7,205,363 |
| EU27+UK | 2,183,733 | 69,618 | 206,705 | 863,368 | 3,323,424 |




Table A2.3 compares the new results with previous European estimates (Etiope, 2009; Petrescu et al. 2021a).

*Table A2.3: European geo-CH$_4$ emission estimates* (Tg yr$^{-1}$)

| | Etiope (2009) | Petrescu et al. (2021a) | This work |
|---|---|---|---|
| Geographic Europe (onshore+ offshore, including Azerbaijan) | 3 | | |
| EU27+UK onshore | | 8.8 (from global grid model) 1.3 (scaled-down as Hmiel, 2020) | 2.4 |
| EU27+UK onshore + offshore | | | 3.3 |
| EU49 onshore | | | 5.5 |
| EU49 onshore + offshore | | | 7.2 |

The overall uncertainties of the spatial distribution of the geo-CH$_4$ sources and CH$_4$ emissions depend on
individual uncertainties of the four categories of seepage, which are discussed in Etiope et al. (2019). Compared to
the global grid model, we have reduced the uncertainty of the microseepage at European scale by refining the activity
(microseepage area).

The new EU27+UK geo-CH$_4$ emission estimate is lower than the one derived by Petrescu et al. (2021a) using
the global gridding, but higher than the scaled-down value (eq. 1). The EU49 (onshore+offshore) emission is higher
than, but of the same order of magnitude of, the preliminary, rough estimate of geographic Europe, which included
Azerbaijan, by Etiope (2009). Onshore geo-CH$_4$ emissions occur mostly in Azerbaijan, Italy, Romania, which are
actually the EU49 countries with major onshore oil-gas reserves and production (BP, 2020), and thus with greater
natural seepage potential. Offshore emissions are dominated by the large estimates published for the UK shelf (Judd
et al, 1997, revised by Tizzard, 2008). These estimates need to be verified and improved. Beyond the emission values,
our gridding provides, however, the first detailed map of natural geological methane emission in Europe, which can
be used for continental scale methane budget modelling.

### *Top-down CH$_4$ emissions estimates*

***Global Carbon Project - Global Methane Budget (Saunois et al., 2020)***

For this study, none of the global inversions were updated.

GMB uses an ensemble of thirteen monthly gridded estimates of wetland emissions based on different land
surface models as calculated for Saunois et al. (2020). Each model conducted a 30-year spin-up and then simulated



net methane emissions from wetland ecosystems over 2000-2017. The models were forced by CRU-JRA reconstructed
climate fields (Harris, 2019), and by the remote sensing-based wetland dynamical area dataset WAD2M (Wetland
Area Dynamics for Methane Modeling). This data set provides monthly global areas over 2000-2017 based on a
combination of microwave remote sensing data from Schroeder et al. (2015) and various regional inventory data sets.
More information is available in Saunois et al. (2020) and more details will be presented in a future publication led
by Poulter et al., 2017 and colleagues.

**Uncertainty:** As described by Saunois et al. (2020) uncertainties are reported as minimum and maximum values of
the available studies, in brackets. They do not take into account the uncertainty of the individual estimates, but rather
express the uncertainty as the range of available mean estimates, i.e., the standard error across
measurements/methodologies considered.

## A3: Anthropogenic and natural N₂O emissions

### *Bottom-up N₂O emission estimates*

***UNFCCC NGHGI (2019), EDGAR v6.0 and CAPRI:*** descriptions are found in Appendix A1.

### *ECOSSE*

ECOSSE is a biogeochemical model that is based on the carbon model ROTH-C (Jenkinson and Rayner,
1977; Jenkinson et al. 1987; Coleman and Jenkinson, 1996) and the nitrogen-model SUNDIAL (Bradbury et al. 1993;
Smith et al. 1996). All processes of the carbon and nitrogen dynamics are considered (Smith et al., 2010a,b).
Additionally, in ECOSSE processes of minor relevance for mineral arable soils are implemented as well (e.g. methane
emissions) to have a better representation of processes that are relevant for other soils (e.g. organic soils). ECOSSE
can run in different modes and for different time steps. The two main modes are site specific and limited data. In the
later version, basis assumptions/estimates for parameters can be provided by the model. This increases the uncertainty
but makes ECOSSE a universal tool that can be applied for large scale simulations even if the data availability is
limited. To increase the accuracy in the site-specific version of the model, detailed information about soil properties,
plant input, nutrient application and management can be added as available.

During the decomposition process, material is exchanged between the SOM pools according to first order
rate equations, characterised by a specific rate constant for each pool, and modified according to rate modifiers
dependent on the temperature, moisture, crop cover and pH of the soil. The N content of the soil follows the
decomposition of the SOM, with a stable C:N ratio defined for each pool at a given pH, and N being either mineralised
or immobilised to maintain that ratio. Nitrogen released from decomposing SOM as ammonium ($NH4+$) or added to
the soil may be nitrified to nitrate ($NO3-$).

For spatial simulations the model is implemented in a spatial model platform. This allows us to aggregate the
input parameter for the needed resolution. ECOSSE is a one-dimensional model and the model platform provides the
input data in a spatial distribution and aggregates the model outputs for further analysis. While climate data are



interpolated, soil data are represented by the dominant soil type or by the proportional representation of the different soil types in the spatial simulation unit (this is in VERIFY a grid cell).

**Uncertainties** in ECOSSE arise from three primary sources: parameters, forcing data (including spatial and temporal resolution), and model structure.

*DayCent*

DayCent was designed to simulate soil C dynamics, nutrient flows (N, P, S) and trace gas fluxes ($CO_2$, $CH_4$, $N_2O$, $NO_x$, $N_2$) between soil, plants and the atmosphere at daily time-step. Submodels include soil water content and temperature by layer, plant production and allocation of net primary production (NPP), decomposition of litter and soil organic matter, mineralization of nutrients, N gas emissions from nitrification and denitrification, and $CH_4$ oxidation in non-saturated soils.

The DayCent modelling application at the EU level is a consolidated model framework running on LUCAS point (Orgiazzi, 2018) which was extensively explained in previous works (Lugato et al., 2017, 2018; Quemada et al., 2020) where a detailed description of numerical and geographical datasets and uncertainty estimations is reported.

Information directly derived from LUCAS (2009-2015) included the soil organic carbon content (SOC), particle size distribution and pH. Hydraulic properties and bulk density was also calculated with an empirically-derived pedotransfer. Management information was derived from official statistics (Eurostat, 2019) and included crop shares at NUTS2 level. The amount of mineral N was partitioned according to the regional crop rotations and agronomic crop requirements. Organic fertilization and irrigated areas were derived from the 'Gridded Livestock of the World' FAO dataset and the FAO-AQUASTAT product.

Meteorological data were downloaded from the E-OBS gridded dataset (http://www.ecad.eu) at 0.1° resolution. For the climatic projection, the gridded data from CORDEX database (https://esgf-node.ipsl.upmc.fr/search/cordex-ipsl/) were used. The average annual (2006-2010) atmospheric N deposition from the EMEP model (rv 4.5) were also implemented into the simulations. The results were updated to 2015-2019 mean.

**Uncertainty:** The starting year of the simulation was set in 2009 and projected in the future. The uncertainty analysis, based on the Montecarlo approach, was done running the model 52 times in each point and, contemporary, randomly sampling model inputs from probability density functions for: SOC pool partition, irrigation and both mineral and organic fertilization rates. The model outputs (including uncertainties) at point level were up-scaled regionally at 1 km resolution by a machine learning approach based on Random Forest regression.

*$N_2O$ emissions from inland waters*

The $N_2O$ estimate represents a climatology of average annual $N_2O$ emissions from rivers, lakes, reservoirs and estuaries at the spatial resolution of 0.1°. Based on a spatially explicit representation of water bodies and point and non-point sources of N and P, this model quantifies the global scale spatial patterns in inland water $N_2O$ emissions in a consistent manner at 0.5° resolution, which were then downscaled to 0.1° using the spatial distribution of European inland water bodies. The procedure to calculate the cascading loads of N and P delivered to each water body along the river–reservoir–estuary continuum and to topologically connect 1.4 million lakes (extracted from the HYDROLAKES



database) is described in Maavara et al., 2019 and Lauerwald et al., 2019. The methodology to quantify $N_2O$ emissions is based on the application of a mechanistic stochastic model (MSM) to estimate inland water C-N-P cycling as well as $N_2O$ production and emission generated by nitrification and denitrification. Using a Monte Carlo analysis, the MSM
allows to generate relationships relating N processes and $N_2O$ emissions to N and P loads and water residence time from the mechanistic model outputs, which are subsequently applied for the spatially resolved upscaling. For the estimation of $N_2O$ emission, we ran two distinct model configurations relying on EFs scaling to denitrification and nitrification rates: one assuming that $N_2O$ production equals $N_2O$ emissions, the other taking into account the kinetic limitation on $N_2O$ gas transfer and progressive $N_2O$ reduction to $N_2$ during denitrification in water bodies with
increasing residence time (Maavara et al., 2019). The model outputs from the two scenarios are used to constrain uncertainties in $N_2O$ emission estimates. For this study, the upscaled RECCAP2 estimates were used (Lauerwald et al., in prep)

### GAINS

Specific sectors and abatement technologies in GAINS vary by the specific emitted compound, with source sector definition and emission factors largely following the IPCC methodology at the Tier 1 or Tier 2 level. GAINS includes in general all anthropogenic emissions to air, but does not cover emissions from forest fires, savannah burning and land use / land use change. Emissions are estimated for 174 countries/regions, with the possibility to aggregate to a global emission estimate, and spanning a timeframe from 1990 to 2050 in five-year intervals. Activity drivers for
macroeconomic development, energy supply and demand, and agricultural activities are entered externally, GAINS extends with knowledge required to estimate "default" emissions (emissions occurring due to an economic activity without emission abatement) and emissions and costs of situations under emission control (Amann et al., 2001).

Emissions of nitrous oxide derive from energy, industry, agriculture, and waste. Land use change emissions are not included. In the energy sector, certain technologies implemented to improve air quality affect $N_2O$ emission
factors (like catalytic converters in vehicles), sometimes also negatively. That is also the case for non-selective catalytic reduction devices for NOx abatement in power plants, or for fluidized bed combustion. Relevant industrial processes cover nitric acid and adipic acid, with other processes (glyoxal, if relevant, or caprolactam) included. Both processes allow for two different levels of abatement technologies, which both are relatively easily accessible and low cost. The use of $N_2O$ in gaseous form, often as an anesthetic for medical purposes, is associated with population
numbers and scaled by availability of hospital beds. Marked emission reductions (at low costs) as well as complete phase out of emissions (high costs) are implemented as technologies. Agricultural emissions in part derive from manure handling, where different management strategies have repercussions on emissions. The larger fraction of emission is from application of nitrogen compounds in different forms to grassland, crops and rice, with rice using a different emission factor. Application of manure and of mineral fertilizer in GAINS can be reduced by advanced
computer technology such as automatic steering and variable rate application, or by agrochemistry (nitrification inhibitors). Costs of implementation are considered to depend on the size of a farm, hence farm size is an important parameter. In the waste sector, composting and wastewater treatment are considered relevant sources. For wastewater treatment, GAINS also considers a specific emission reduction option when optimizing processes towards $N_2O$



reduction (e.g. via favoring the anammox process). All details have been reported by Winiwarter et al. (2018) in their supplementary material.

**Uncertainties**: The same paper provides full information on the uncertainty of $N_2O$ emissions in the GAINS model, which is a consequence of uncertainty provided in the activity data, in the emission factors, and in the actual structure of the respective management strategies that also include the share of abatement technology already implemented. Further parameters also described (on uncertainty of future projections and on costs) are not relevant here.

### FAOSTAT

FAOSTAT: Statistics Division of the Food and Agricultural Organization of the United Nations, provides $N_2O$ emissions from agriculture: https://www.fao.org/faostat/en/#data/GT and its sub-domains, as well as $N_2O$ emissions from land use linked to biomass burning. The FAOSTAT emissions database is computed following Tier 1 IPCC 2006 Guidelines for National GHG Inventories (http://www.ipcc-nggip.iges.or.jp/public/2006gl/index.html). Country reports to FAO on crops, livestock and agriculture use of fertilizers are the source of activity data. Geospatial data are the source of AD for the estimates from cultivation of organic soils, biomass and peat fires. $N_2O$ emissions are provided by country, regions and special groups, with global coverage, relative to the period 1961-present (with annual updates, currently 2019) and with projections for 2030 and 2050 for agriculture only, expressed in both $CO_2e$ and $N_2O$ by underlying agricultural and land use emission sub-domain and by aggregate (agriculture total, agriculture total plus energy, agricultural soils). The main $N_2O$ emissions are reported for the following agricultural activities: manure management, synthetic fertilizers, manure applied to the soils, manure left in pasture, crop residues, cultivation of organic soils and burning crop residues. LULUCF emissions consist of $N_2O$ associated with burning biomass and peat fires, as well as from the drainage of organic soils. Comparison to the UNFCCC submissions is also provided.

**Uncertainties** were computed by Tubiello et al., 2013 but are not available in the FAOSTAT database.

### Top-down N₂O emission estimates
### FLEXINVERT

The FlexInvert framework is based on Bayesian statistics and optimizes surface-atmosphere fluxes using the maximum probability solution (Rodgers 2000). Atmospheric transport is modelled using the Lagrangian model FLEXPART (Stohl et al. 2005; Pisso et al. 2019) run in the backwards time mode to generate a so-called Source-Receptor Matrix (SRM). The SRM describes the relationship between the change in mole fraction and the fluxes discretized in space and time (Seibert and Frank, 2004) and was calculated for 7 days prior to each observation. For use in the inversions, FLEXPART was driven using ECMWF Era Interim wind fields.

The state vector consisted of flux increments (i.e. offsets to the prior fluxes) discretized on an irregular grid based on the SRMs (Thompson et al. 2014). This grid has finer resolution (in this case the finest was $0.5°×0.5°$) where the fluxes have a strong influence on the observations and coarser resolution where the influence is only weak (the coarsest was $2°×2°$). The flux increments were solved at 2-weekly temporal resolution. The state vector also included scalars for the background mole fractions. The optimal (posterior) fluxes were found using the Conjugate Gradient method (e.g. Paige and Saunders, 1975).



The background mole fractions, i.e., the contribution to the modelled mole fractions that is not accounted for in the 7-day SRMs, was estimated by coupling the termination points of backwards trajectories (modelled using virtual particles) to initial fields of mole fractions from the optimized Eulerian model LMDz (i.e. the CAMS $N_2O$ mole fraction product v18r1) following the method of Thompson et al. 2014.

**Uncertainties:** The posterior fluxes are subject to systematic errors primarily from: 1) errors in the modelled
atmospheric transport; 2) aggregation errors, i.e. errors arising from the way the flux variables are discretized in space and time; 3) errors in the background mole fractions; and 4) the incomplete information from the observations and hence the dependence on the prior fluxes. In addition, there is, to a smaller extent, some error due to calibration offsets between observing instruments, which is more pertinent for $N_2O$ than for other GHGs. Random uncertainties are calculated from a Monte Carlo ensemble of inversions following Chevallier et al. (2007) and uncertainties in the
observation space were inflated to take into account the model representation errors.

***Global N₂O Budget – GCP (Tian et al., 2020)***

***CAMS-N₂O***

        Within the GCP 2019 results, $N_2O$ fluxes are estimated using the atmospheric inversion framework, CAMS-
$N_2O$. Atmospheric inversions use observations of atmospheric mixing ratios, in this case, of $N_2O$, and provide the fluxes that best explain the observations while at the same time being guided by a prior estimate of the fluxes. In other words, the fluxes are optimized to fit the observations within the limits of the prior and observation uncertainties. To produce the optimized (*a posteriori*) fluxes a number of steps are involved: first, the observations are pre-processed, second, a prior flux estimate is prepared, third mixing ratios are simulated using the prior fluxes and are used to
estimate the model representation error, and fourth, the inversion is performed. In total 140 ground-based sites, ship and aircraft transects are included in the inversion. The term "site" refers to locations where there is a long-term record of observations and includes ground-based measurements, both from discrete samples (or "flasks") and quasi-continuous sampling by in-situ instruments, as well as aircraft measurements. A prior estimate of the total $N_2O$ flux with monthly resolution and inter-annually varying fluxes is prepared from a number of models and inventories. For
the soil fluxes (including anthropogenic and natural) an estimate from the land surface model OCN-v1.2 is used, which is driven by observation-based climate data, N-fertilizer statistics and modelled N-deposition (Zaehle et al. 2011). For the ocean fluxes, an estimate from the ocean biogeochemistry model PlankTOM-v10.2 is used, which is a prognostic model (Buitenhuis et al. 2018). Atmospheric transport is modelled using an offline version of the Laboratoire de Meteorologie Dynamique model, LMDz5, which computes the evolution of atmospheric compounds using archived
fields of winds, convection mass fluxes and planetary boundary layer (PBL) exchange coefficients that have been calculated using the online version nudged to ECMWF ERA interim winds.

CAMS-$N_2O$ uses the Bayesian inversion method to find the optimal fluxes of $N_2O$ given prior information about the fluxes and their uncertainty, and observations of atmospheric $N_2O$ mole fractions. The method is the same as that used in Thompson et al. (2014)

**Uncertainty:** Uncertainties in CAMS-$N_2O$ simulations pertain to observation space and to state space. Uncertainty in the observation space is calculated as the quadratic sum of the measurement and transport uncertainties. The measurement uncertainty is assumed to be 0.3 ppb (approximately 0.1%) based on the recommendations of data



providers. The transport uncertainty includes estimates of uncertainties in advective transport (based on the method of Rödenbeck et al. (2003)) and from a lack of subgrid-scale variability (based on the method of Bergamaschi et al. (2010)). For the error in each land grid cell, the maximum magnitude of the flux in the cell of interest and its 8 neighbours is used, while for ocean grid cells the magnitude of the cell of interest only is used. Posterior flux uncertainties are calculated from a Monte Carlo ensemble of inversions, based on the method of Chevallier et al. (2005).

### TOMCAT-INVICAT

TOMCAT-INVICAT (Wilson et al., 2014) is a variational inverse transport model, which is based on the global chemical transport model TOMCAT, and its adjoint. It uses a 4-D variational (4D-VAR) optimization framework based on Bayesian theory which seeks to minimize model-observation differences by altering surface fluxes, while allowing for prior knowledge of these fluxes to be retained. TOMCAT (Monks et al., 2017) is an offline chemical transport model, in which meteorological data is taken from ECMWF ERA-Interim reanalyses (Dee et al., (2011)). The model grid resolution, and therefore the optimised surface flux estimates, have a horizontal resolution of 5.6 x 5.6 degrees. The model has 60 vertical levels running from the surface to 0.1 hPa. For each individual year's fluxes, which are optimised on a monthly basis, 30 minimisation iterations are carried out.

**Uncertainty:** Uncertainties in TOMCAT-INVICAT $N_2O$ inversions are described as follows and further in Thompson et al., (2019). Uncertainty in the observations is calculated as the quadratic sum of the measurement and transport uncertainties. The measurement uncertainty for each observation is assumed to be 0.4 ppb. For the transport error for each observation is assumed to be the mean difference between the observation grid cell and its 8 neighbours. Prior flux errors are assumed to be 100% or the prior estimate, and are uncorrelated in space and time. Posterior flux uncertainties are not currently able to be calculated.

### MIROC4-ACTM

The MIROC4-ACTM time dependent inversion for 84 regions (TDI84) framework is based on Bayesian statistics and optimizes surface-atmosphere fluxes using the maximum probability solution (Rodgers 2000). Atmospheric transport is modelled using the JAMSTEC's Model for Interdisciplinary Research on climate, version 4 based atmospheric chemistry-transport model (MIROC4-ACTM) (Watanabe et al. 2008; Patra et al. 2018, 2022). The Source-Receptor Matrix (SRM) is calculated by simulating unitary emissions from 84 basis regions, for which the fluxes are optimised. The SRM describes the relationship between the change in mole fraction at the measurement locations for the unitary basis region fluxes (similar to Rayner et al., 1997). The MIROC4-ACTM meteorology was nudged to the JMA 55-year reanalysis (JRA55) horizontal wind fields and temperature.

The simulated mole fractions for the total a priori fluxes are subtracted from the observed concentrations before running the inversion calculation (as in Patra et al., 2016 for $CH_4$ inversion). In this study, the simulation have been updated to 2019 (Patra et al., 2022).

**Uncertainties:** The posterior fluxes are subject to systematic errors primarily from: 1) errors in the modelled atmospheric transport; 2) aggregation errors, i.e. errors arising from the way the flux variables are discretized in space



(84 regions) and time (monthly-means); 3) errors in the background mole fractions (assumed to be a minor factor); and 4) the incomplete information from the sparse observational network and hence the dependence on the prior fluxes. In addition, there is, to a much smaller extent, some error due to calibration offsets between observing instruments, which is more pertinent for $N_2O$ than for other GHGs. We have validated model transport in the troposphere using SF6 for the inter-hemispheric exchange time, and the using SF6 and $CO_2$ for the age of air in the stratosphere. The

simulated $N_2O$ concentrations are also compared with aircraft measurements in the upper troposphere and lower stratosphere for evaluating the stratosphere-troposphere exchange rates. Comparisons with ACE-FTS vertical profiles in the stratosphere and mesosphere indicate good parameterisation of $N_2O$ loss by photolysis and chemical reactions, and thus the lifetime, which affect the global total $N_2O$ budgets.

Random uncertainties are calculated by the inverse model depending on the prior flux uncertainties and the

observational data density and data uncertainty. Only 37 sites are used in the inversion and thus the reduction in priori flux uncertainties have been minimal. The net fluxes from the inversion from individual basis regions are less reliable compared to the anomalies in the estimated fluxes over a period of time.

**B1: Overview figures**


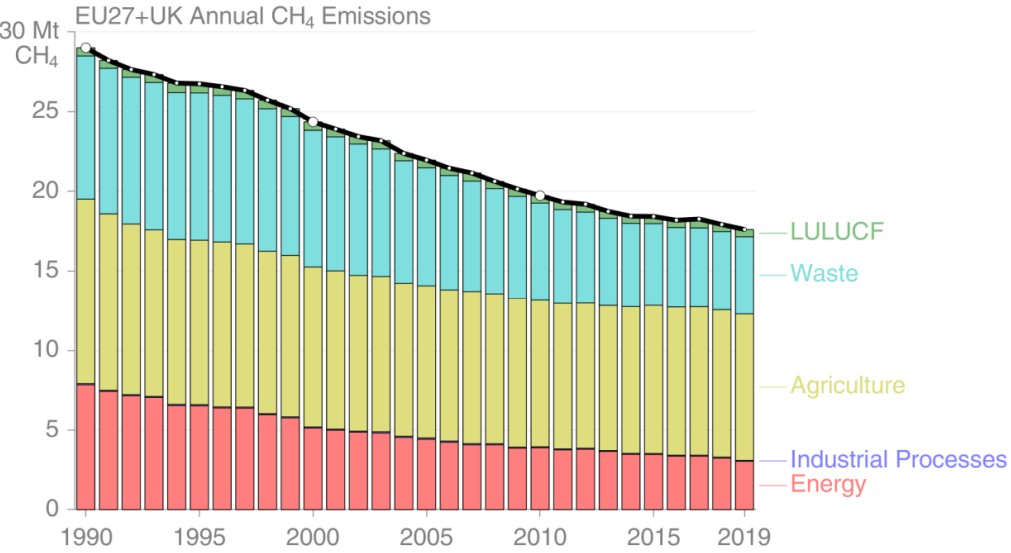

*Figure B1a: EU27+UK total CH₄ emissions time series per sectors as reported by UNFCCC NGHGI (2021).*



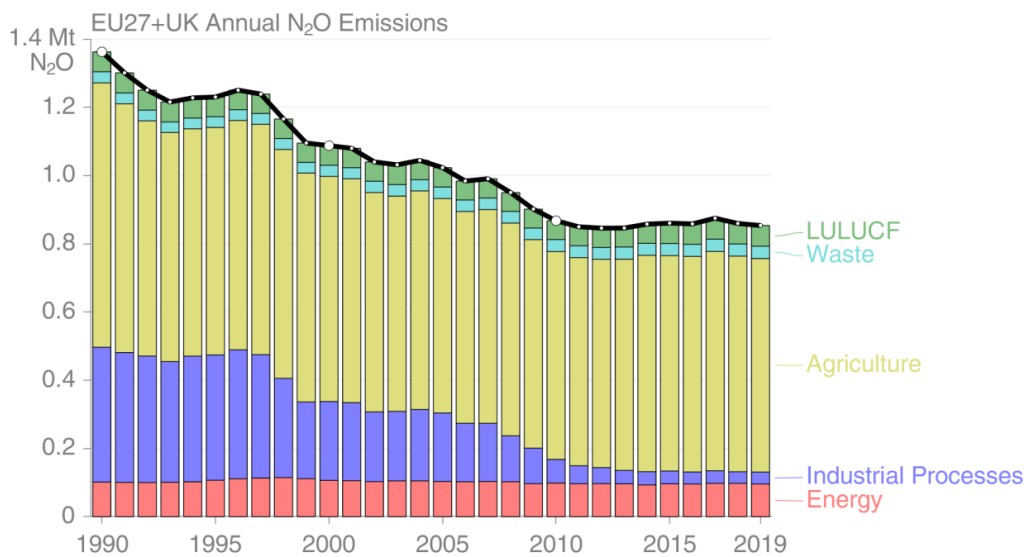


*Figure B1b: EU27+UK total N$_2$O emissions time series per sectors as reported by UNFCCC NGHGI (2021).*

### B2: Source specific methodology: AD, EF and uncertainties

*Table B2.1: Source specific activity data (AD), emission factors (EF) and uncertainty methodology for all current*
*VERIFY and non-VERIFY 2021 data product collection.*

| CH$_4$ bottom-up anthropogenic emissions | | | | |
|---|---|---|---|---|
| **Data source** | **AD/Tier** | **EFs/Tier** | **Uncertainty assessment method** | **Emission data availability** |
| **UNFCCC NGHGI (2021)** | Country-specific information consistent with the IPCC GLs. | IPCC GLs/country-specific information for higher tiers. | IPCC GLs (https://www.ipcc-nggip.iges.or.jp/public/ 2006gl/, last access: December 2019) for calculating the uncertainty of emissions based on the uncertainty of AD and EF, two different approaches: (1) error propagation and (2) Monte Carlo simulation.<br><br>UBA-Vienna provided yearly harmonized and gap-filled uncertainties | NGHGI official data (CRFs) are found at https://unfccc.int/ghg-inventories-annex-i-parties/2021 |



| EDGAR v6.0 | International Energy Agency (IEA) for fuel combustion Food and Agricultural Organisation (FAO) for agriculture US Geological Survey (USGS) for industrial processes (e.g. cement, lime, ammonia and ferroalloys) GGFR/NOAA for gas flaring World Steel Association for iron and steel production International Fertilisers Association (IFA) for urea consumption and production Complete description of the data sources can be found in Janssens-Maenhout et al. 2019 and in Crippa et al. (2019b). | IPCC 2006, Tier 1 or Tier 2 depending on the sector | Tier 1 with error propagation by sectors for $CH_4$ | https://edgar.jrc.ec.europa.eu/dataset_ghg60 |
| --- | --- | --- | --- | --- |
| CAPRI | Farm and market balances, economic parameters, crop areas, livestock population and yields from EUROSTAT, parameters for input-demand functions at regional level from FADN (EC), data on trade between world regions from FAOSTAT, policy variables from OECD. | IPCC 2006: Tier 2 for emissions from enteric fermentation of cattle and from manure management of cattle. Tier 1 for all other livestock types and emission categories. N-flows through agricultural systems (including N excretion) calculated endogenously. | Spatial uncertainties computed for 2014, 2016 and 2018 | Detailed gridded data $CH_4$ and $N_2O$ emissions can be obtained by contacting the data provider: Adrian.Leip@ec.europa.eu |





| GAINS | Livestock numbers by animal type (FAOSTAT, 2010; EUROSTAT, 2009; UNFCCC, 2010) Growth in livestock numbers from FAOSTAT (2003), CAPRI model (2009) Rice cultivation Land area for rice cultivation (FAOSTAT, 2010) Projections for EU are taken from the CAPRI Model | Country-specific information and: Livestock - Implied EFs reported to UNFCCC and IPCC Tier 1 (2006, Vol.4, Ch. 10) default factors Rice cultivation - IPCC Tier 1–2 (2006, Vol. 4, p. 5.49 Agricultural waste burning - IPCC Tier 1 (2006, Vol. 5, p. 520 | IPCC (2006, Vol.4, p.10.33) uncertainty range | Detailed gridded data $CH_4$ and $N_2O$ emissions can be obtained by contacting the data providers: for $CH_4$, contact Lena Höglund Isaksson (hoglund@iiasa.ac.at); for $N_2O$, contact Wilfried Winiwarter (winiwart@iiasa.ac.at). |
| **FAOSTAT** | FAOSTAT Crop and Livestock Production domains from country reporting; FAOSTAT Land Use Domain; Harmonized world soil; ESA CCI and Copernicus Global Land Cover Service (C3S) maps; MODIS MCD12Q1 v6; FAO Gridded Livestock of the World; MODIS MCD64A1.006 burned area products | IPCC guidelines Tier 1 | IPCC (2006, Vol.4, p.10.33) Uncertainties in estimates of GHG emissions are due to uncertainties in emission factors and activity data. They may be related to, inter alia, natural variability, partitioning fractions, lack of spatial or temporal coverage, or spatial aggregation. | Agriculture total and subdomain specific GHG emissions are found for download at http://www.fao.org/faostat/en/#data/GT (last access: April 2022). |
| **$CH_4$ bottom-up natural emissions** | | | | |
| **Data source** | **AD/Tier** | **EFs/Tier** | **Uncertainty assessment method** | **Emission data availability** |
| | | | | |

| Mechanistic Stochastic Model CH₄ emissions from inland waters | Hydrosheds 15s (Lehner et al., 2008) and Hydro1K (USGS, 2000) for river network, HYDROLAKES for lakes and reservoirs network and surface area (Messager et al., 2016); Worldwide Typology of estuaries by Dürr et al. (2011) | N/A | Four model configurations for CH₄ | Detailed gridded data can be obtained by contacting the data providers: Ronny Lauerwald ronny.lauerwald@inrae.fr Pierre Regnier Pierre.Regnier@ulb.ac.be |
|---|---|---|---|---|
| **JSBACH-HIMMELI** | JSBACH vegetation and soil carbon and physical parameters provided to HIMMELI to simulate wetland methane fluxes HydroLAKES database (Messager et al., 2016). CORINE land cover data VERIFY climate drivers 0.1° × 0.1° | CH₄ fluxes from peatlands and mineral soils | the standard deviation and the resulting range in the annual emission sum represents a measure of uncertainty. | Detailed gridded data CH₄ emissions can be obtained by contacting the data providers: Tuula.Aalto@fmi.fi tiina.markkanen@fmi.fi |
| Geological emissions, including marine and land geological) | Areal distribution activity: 1° × 1° maps include the four main categories of natural geo-CH₄ emission: (a) onshore hydrocarbon macro-seeps, including mud volcanoes, (b) submarine (offshore) seeps, (c) diffuse microseepage and (d) geothermal manifestations. | CH₄ fluxes, measurements and estimates based on size and activity | 95% confidence interval of the median emission-weighted mean sum of individual regional values | Etiope et al, 2019 with updated activity for current study) Detailed gridded data on geological CH₄ emissions can be obtained by contacting the data providers: Giuseppe Etiope: giuseppe.etiope@ingv.it Giancarlo Ciotoli giancarlo.ciotoli@gmail.com |



| CH$_4$ Top-down inversions | | | | |
|---|---|---|---|---|
| Regional inversions over Europe ( high transport model resolution ) | | | | |
| **Data source** | **AD/Tier** | **EFs/Tier** | **Uncertainty assessment method** | **Emission data availability** |
| **FLEXPART - FLExKF** | Extended Kalman Filter in combination with backward Lagrangian transport simulations using the model FLEXPART Atmospheric observations ECMWF Era Interim meteorological fields | FLExKF-CAMSv19r_EMPA specific background | The random uncertainties are represented by the posterior error covariance matrix provided by the Kalman Filter, which combines errors in the prior fluxes with errors in the observations and model representation (see description in Appendix A1) | Detailed gridded data can be obtained by contacting the data provider: Dominik.Brunner@empa.ch |
| **TM5-4DVAR** | Global Eulerian models with a zoom over Europe, ERA-Interim reanalysis | 4DVAR variational techniques | Uncertainty was calculated as 1σ estimate. See descriptions in Appendix A1 | Detailed gridded data can be obtained by contacting the data provider: Peter.BERGAMASCHI@ext.ec.europa.eu |
| **FLEXINVERT** | Bayesian statistics Atmospheric transport is modelled using the Lagrangian model FLEXPART | prior fluxes from LPX-Bern DYPTOP, EDGAR v4.2 FT2010 GFED v4 Termites and ocean fluxes ground-based surface CH$_4$ observations. Background fields based on nudged FLEXPART-CTM simulations (Groot Zwaaftink et al., 2018) | | Detailed gridded data CH$_4$ emissions can be obtained by contacting the data provider: Christine Groot Zwaaftink cgz@nilu.no |
| **InTEM-NAME** | Atmospheric Lagrangian trans port model analysis 3-D meteo rology from the UK Met Office Unified Model. | (a) the UK National Atmospheric Emissions Inventory (NAEI) 2015 within the UK. (b) Outside the UK – EDGAR 2010 emissions distributed uni formly over land (excluding the UK). | Derived from the variability of the observations within each 2 h period: a) 40 %; b) 50 %. | Detailed gridded data can be ob tained by contacting the data provider: Alistair Manning (alistair.manning@ metoffice.gov.uk). |
| **CTE-FMI** | Ensemble Kalman filter Eulerian transport model TM5 ECMWF ERA-Interim meteorological data | prior fluxes from LPX-Bern DYPTOP, EDGAR v4.2 FT2010 GFED v4 Termites and ocean fluxes ground-based surface CH$_4$ observations GOSAT XCH$_4$ retrievals from NIES v2.72 | The prior uncertainty is assumed to be a Gaussian probability distribution function The posterior uncertainty is | Detailed gridded data can be obtained by contacting the data provider: aki.tsuruta@fmi.fi |

| | | | calculated as standard deviation of the ensemble members, where the posterior error covariance matrix are driven by the ensemble Kalman filter. | |
|---|---|---|---|---|
| **InGOS** | 18 European monitoring stations EDGARv4.2FT-InGOS wetland inventory of J.Kaplan and LPX-Bern v1.0 ERA-Interim reanalysis Met Office Unified Model | For Priors please see Table B4 | The uncertainty of the model ensemble was calculated as 1σ estimate. Individual models use Bayes' theorem to calculate the reduction of assumed a priori emission uncertainties by assimilating measurements. | Detailed gridded data can be obtained by contacting the data provider: Peter.BERGAMASCHI@ext.ec.europa.eu |
| **Global inversions from the Global Carbon Project CH₄ budget (Saunois et al. 2020)** | | | | |
| **GCP-CH₄ 2019 anthropogenic and natural partitions from inversions** | ensemble of inversions gathering various chemistry transport models surface or satellite data | For Priors please see Table B4 | Uncertainties are reported as minimum and maximum values of the available studies, as the range of available mean estimates, i.e., the standard error across measurements/methodologies considered. Posterior uncertainty mostly use Monte Carlo methods | Detailed gridded data can be obtained by contacting the data provider: Marielle Saunois marielle.saunois@lsce.ipsl.fr |
| **N₂O bottom-up anthropogenic emissions** | | | | |
| **Data source** | **AD/Tier** | **EFs/Tier** | **Uncertainty assessment method** | **Emission data availability** |
| **UNFCCC NGHGI (2021), EDGAR v6.0, CAPRI, GAINS and FAOSTAT see above** | | | | |
| **ECOSSE** | The model is a point model, which provides spatial results by using spatial distributed input data (lateral fluxes are not considered). The model is a TIER 3 approach that is applied on | IPCC 2006: Tier 3 The simulation results will be allocated due to the available information (size of spatial unit, representation of considered land use, etc.). | N/A | Detailed gridded data can be obtained by contacting the data provider: Kuhnert, Matthias matthias.kuhnert@abdn.ac.uk |





| | | | | |
|---|---|---|---|---|
| | grid map data, polygon organized input data or study sites. | | | |
| **DayCent** | Spatial explicit simulations at point level, up-scaled at 1km for agricultural areas. | Tier 3; Land management and input factors for the cropland remaining cropland category based on datasets covering the 2005-2015 period. | Monte Carlo | Detailed gridded data can be obtained by contacting the data provider: Emanuele.LUGATO@ec.europa.eu |
| **N₂O bottom-up natural emissions** | | | | |
| **Mechanistic Stochastic Model for N₂O emissions from inland waters** | Hydrosheds 15s (Lehner et al., 2008) and Hydro1K (USGS, 2000) for river network, HYDROLAKES for lakes and reservoirs network and surface area (Messager et al., 2016); Worldwide Typology of estuaries by Dürr et al. (2011); terrestrial N and P loads by Global-NEWS (Van Drecht et al., 2009; Bouwman et al., 2009), resdistributed at 0.5° resolution by Maavara et al., 2019. | EFs applied to denitrification and nitrification rates for N2O emissions. Values constrained from the range reported in Beaulieu et al., 2011. | Upscaled emission estimates from RECCAP2 | Detailed gridded data can be obtained by contacting the data providers: Ronny Lauerwald ronny.lauerwald@inrae.fr Pierre Regnier Pierre.Regnier@ulb.ac.be |
| **Regional N₂O inversions over Europe ( high transport model resolution )** | | | | |
| **FLEXINVERT** | Bayesian statistics Atmospheric transport is modelled using the Lagrangian model FLEXPART | background mole fractions | Random uncertainties are calculated from a Monte Carlo ensemble of inversions | Detailed gridded N₂O data can be obtained by contacting the data provider: Rona Thompson rlt@nilu.no |
| **Global N₂O inversions over Europe from GN₂OB (Tian et al., 2020)** | | | | |
| **CAMS-N₂O** | Bayesian inversion method observations of atmospheric mixing ratios fluxes from ground-based | Fires emission factors from Akagi et al., 2011 | Uncertainty in the observation space is calculated as the quadratic sum of the measurement and transport uncertainties. | Detailed gridded N₂O data can be obtained by contacting the data provider: Rona Thompson rlt@nilu.no |

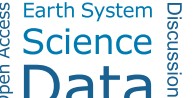

| Model | | | |
|---|---|---|---|
| | sites, ship and aircraft transects soil fluxes OCN-v1.2 ocean biogeochemistry model PlankTOM-v10.2 GFED-v4.1s EDGAR-4.32 ECMWF ERA interim | | For the error in each land grid cell, the maximum magnitude of the flux in the cell of interest and its 8 neighbours is used; for ocean grid cells the magnitude of the cell of interest only is used. | |
| **TOMCAT-INVICAT** | Variational Bayesian inverse model assimilating surface flask observations of atmospheric mixing ratios. ECMWF ERA-Interim meteorological driving data. | Prior emissions estimates are from OCN-v1.1 model (soils), EDGARv4.2FT2010 (anthro. non-soil), PlankTOM5 (oceans) and GFEDv4.1s (biomass burning). | Uncertainty in the observation space is calculated as the quadratic sum of the measurement and transport uncertainties. For the error in each land grid cell, the maximum magnitude of the flux in the cell of interest and its 8 neighbours is used. Prior emission uncertainties are 100% and uncorrelated. | Detailed gridded N$_2$O data can be obtained by contacting the data provider: Christopher Wilson [GEO] C.Wilson@leeds.ac.uk |
| **MIROC4-ACTM** | Matrix inversion for calculation of fluxes from 53 and 84 partitions of the globe for CH$_4$ and N$_2$O, respectively. Forward model transport is nudged to JRA-55 horizontal winds and temperature. | Fire emissions for CH$_4$ are taken from GFEDv4s | A posteriori uncertainties are obtained from the Bayesian statistics model. A priori emissions uncertainties are uncorrelated. | Detailed gridded data can be obtained by contacting the data provider: Prabir Patra prabir@jamstec.go.jp |

*Table B2.2: Biogeochemical models that computed wetland emissions used in this study. Runs were performed for the whole period 2000-2017. Models run with prognostic (using their own calculation of wetland areas) and/or diagnostic (using WAD2M) wetland surface areas (see Sect 3.2.1) From Saunois et al., 2020.*


| Model | Institution | Prognostic | Diagnostic | References |
|---|---|---|---|---|

| | | | | |
|---|---|---|---|---|
| CLASS-CTEM | Environment and Climate Change Canada | y | y | Arora, Melton and Plummer (2018) Melton and Arora (2016) |
| DLEM | Auburn University | n | y | Tian et al., (2010;2015) |
| ELM | Lawrence Berkeley National Laboratory | y | y | Riley et al. (2011) |
| JSBACH | MPI | n | y | Kleinen et al. (2019) |
| JULES | UKMO | y | y | Hayman et al. (2014) |
| LPJ GUESS | Lund University | n | y | McGuire et al. (2012) |
| LPJ MPI | MPI | n | y | Kleinen et al. (2012) |
| LPJ-WSL | NASA GSFC | y | y | Zhang et al. (2016b) |
| LPX-Bern | University of Bern | y | y | Spahni et al. (2011) |
| ORCHIDEE | LSCE | y | y | Ringeval et al. (2011) |
| TEM-MDM | Purdue University | n | y | Zhuang et al. (2004) |
| TRIPLEX_GHG | UQAM | n | y | Zhu et al., (2014;2015) |
| VISIT | NIES | y | y | Ito and Inatomi (2012) |

*Table B2.3: Top-down studies used in our new analysis, with their contribution to the decadal and yearly estimates noted. For decadal means, top down studies have to provide at least 8 years of data over the decade to contribute to the estimate, from Saunois et al., 2020*

| Model | Institution | Observation used | Time period | Number of inversions | References |
|---|---|---|---|---|---|
| Carbon Tracker-Europe CH$_4$ | FMI | Surface stations | 2000-2017 | 1 | Tsuruta et al. (2017) |
| Carbon Tracker-Europe CH$_4$ | FMI | GOSAT NIES L2 v2.72 | 2010-2017 | 1 | Tsuruta et al. (2017) |
| GELCA | NIES | Surface stations | 2000-2015 | 1 | Ishizawa et al. (2016) |
| LMDz-PYVAR | LSCE/CEA | Surface stations | 2010-2016 | 2 | Yin et al. (2019) |
| LMDz-PYVAR | LSCE/CEA | GOSAT Leicester v7.2 | 2010-2016 | 4 | Yin et al. (2019) |





| | | | | | |
|---|---|---|---|---|---|
| LMDz-PYVAR | LSCE/CEA | GOSAT Leicester v7.2 | 2010-2017 | 2 | Zheng et al. (2018a, 2018b) |
| MIROC4-ACTM | JAMSTEC | Surface stations | 2000-2016 | 1 | Patra et al. (2016; 2018) |
| NICAM-TM | NIES | Surface stations | 2000-2017 | 1 | Niwa et al. (2017a; 2017b) |
| NIES-TM-FLEXPART-VAR (NTFVAR) | NIES | Surface stations | 2000-2017 | 1 | Maksyutov et al. (2020); Wang et al. (2019b) |
| NIES-TM-FLEXPART-VAR (NTFVAR) | NIES | GOSAT NIES L2 v2.72 | 2010-2017 | 1 | Maksyutov et al. (2020); Wang et al., (2019b) |
| TM5-CAMS | TNO/VU | Surface stations | 2000-2017 | 1 | Segers and Houweling (2018); Bergamaschi et al. (2010; 2013), Pandey et al. (2016) |
| TM5-CAMS | TNO/VU | GOSAT ESA/CCI v2.3.8 (combined with surface observations) | 2010-2017 | 1 | Segers and Houweling (2018,report); Bergamaschi et al. (2010; 2013), Pandey et al. (2016) |
| TM5-4DVAR | EC-JRC | Surface stations | 2000-2017 | 2 | Bergamaschi et al. (2013, 2018) |
| TM5-4DVAR | EC-JRC | GOSAT OCPR v7.2 (combined with surface observations) | 2010-2017 | 2 | Bergamaschi et al. (2013, 2018) |
| TOMCAT | Uni. of Leeds | Surface stations | 2003-2015 | 1 | McNorton et al. (2018) |


*Table B2.4: List of prior datasets for natural CH$_4$ emissions used by all inverse models*

| Project | Model | Prior | | | | | | |
|---|---|---|---|---|---|---|---|---|
| | | Wetlands | Geological | Fire | Termites | Soil sink | Ocean/Lakes | Wild animals |
| VERIFY | CTE_FMI | LPX-Bern DYPTOP (Stocker et al., 2014) | | GFED4s | Ito and Inatomi 2012 | LPX-Bern DYPTOP (Stocker et al., 2014) | Tsuruta et al., 2017 | |
| VERIFY | FLEXPART(FLExKF-CAMSv19r)_EMPA | JSBACH-HIMMELI | | | GCP | Ridgwell /GCP | GCP/ULB | |



| VERIFY | FLEXINVERT | LPX-Bern DYPTOP (Stocker et al., 2014) | | GFED4s | Ito and Inatomi, 2012 | LPX-Bern DYPTOP (Stocker et al., 2014) | Tsuruta et al., 2017 | |
|---|---|---|---|---|---|---|---|---|
| VERIFY | TM5_4DVAR JRC | GCP_CH4_2019 | GCP_CH4 2019 (global total: 15 Tg CH4 yr-1) | | GCP_CH4_2019 | GCP_CH4_ 2019 | GCP_CH4_ 2019 | |
| InGOS | INGOS-CTE-S4_EC | LPX-Bern v1.0 (Spahni et al., 2013) | | GFED | Ito and Inatomi 2012 | LPX-Bern v1.0 (Spahni et al., 2013) | Tsuruta et al., 2015 | |
| InGOS | INGOS-LMDZEU-S4_EC | wetland inventory of J. Kaplan (Bergamaschi et al., 2007) | | | | | | |
| InGOS | INGOS-TM3STILT-S4_EC | wetland inventory of J. Kaplan (Bergamaschi et al., 2007) | | | | | | |
| InGOS | INGOS-TM5VAR-S4_EC | wetland inventory of J. Kaplan (Bergamaschi et al., 2007) | | | Sanderson /GCP | Ridgwell /GCP | Lambert /GCP | Oslson climatology |
| InGOS | INGOS-NAME-S4_EC | wetland inventory of J. Kaplan (Bergamaschi et al., 2007) | | | | | | |
| GCP | GELCA-SURF_NIES | VISIT (Ito and Inatomi, 2012) | n/a | GFEDv3.1 then GFAS v1.2 after 2011 | Sanderson (TransCom-CH4 / GCP) | VISIT (Ito and Inatomi, 2012) | n/a | |
| GCP | MIROCv4-SURF_JAMASTEC | VISIT (Ito and Inatomi, 2012) (global total range : 173-197 Tg CH4 yr-1) | Etiope and Milkov, 2004 (global total: 7.5 Tg CH4 yr-1) | GFEDv4s (global total range : 14-35 Tg CH4 yr-1) | Sanderson (TransCom-CH4) (global total: 20.5 Tg CH4 yr-1) | VISIT (Ito and Inatomi, 2012) | Lambert/Houweling (TransCom-CH4) (global total: 18.5 Tg CH4 yr-1) | |



| GCP | NICAM-SURF_NIES | VISIT (Ito and Inatomi, 2012) | GCP based on Etiope 2015 | GFEDv4s / GCP | Sanderson (TransCom-CH4 / GCP) | VISIT (Ito and Inatomi, 2012) | Lambert/Houweling (TransCom-CH4 / GCP) | |
|---|---|---|---|---|---|---|---|---|
| GCP | TOMCAT-SURF_ECMWF | JULES emissions from Mc Norton 2016a | Tomcat 2006 | GFED V4 | Matthews and Fung 2006 | Patra et al. 2011 | Tomcat 2006 Matthews and Fung 1987 - all emissions total rescaled to Schwietzke et al. 2016 | |
| GCP | NTFVAR-GOSAT_NIES | VISIT (Ito and Inatomi, 2012) | Etiope and Milkov, 2004 | GFAS v1.2 | Ito and Inatomi 2012 | VISIT (Ito and Inatomi, 2012) | TransCom-CH4 | |
| GCP | NTFVAR-SURF_NIES | VISIT (Ito and Inatomi, 2012) | Etiope and Milkov, 2004 | GFAS v1.2 | Ito and Inatomi 2012 | VISIT (Ito and Inatomi, 2012) | TransCom-CH4 | |
| GCP | LMDZ-GOSAT1_LSCE | Bloom 2017 | n/a | GFED V41s | Sanderson /GCP | Ridgwell /GCP | Lambert /GCP | |
| GCP | LMDZ-GOSAT2_LSCE | GCP - ensemble mean ESSD Saunois et al . 2016 | GCP based on Etiope 2015 | GFED V41s | Sanderson /GCP | Ridgwell /GCP | Lambert /GCP | |
| GCP | LMDZ-GOSAT LMDZ-GOSAT LMDZ-GOSAT LMDZ-GOSAT LMDZ-SURF1 LMDZ-SURF2 | Kaplan 2002 rescaled by Bergamaschi 2007 | n/a | GFED V41 | Sanderson 1996 /GCP | Ridgwell /GCP | Lambert and Schmidt 1993 | |
| GCP | TM5-CAMS-GOSAT_TNO | Kaplan climatology | n/a | GFED V31 climatology after 2011 | Sanderson /GCP | Ridgwell /GCP | Lambert /GCP | Oslson climatology |





| GCP | TM5-GOSAT1_EC | WETCHIMP ensemble mean; | GCP_CH$_4$ 2019 (global total: 15 Tg CH$_4$ yr$^{-1}$) | | Sanderson /GCP | Ridgwell /GCP | Lambert /GCP | Oslson climatology |
|---|---|---|---|---|---|---|---|---|
| GCP | TM5-GOSAT2_EC | GCP_CH$_4$_2019 | GCP_CH$_4$ 2019 (global total: 15 Tg CH$_4$ yr$^{-1}$) | GCP_CH$_4$_2019 | GCP_CH$_4$_2019 | GCP_CH4_2019 | GCP_CH$_4$_2019 | |
| GCP | TM5-SURF1_EC | WETCHIMP ensemble mean; | GCP_CH$_4$ 2019 (global total: 15 Tg CH$_4$ yr$^{-1}$) | | Sanderson /GCP | Ridgwell /GCP | Lambert /GCP | Oslson climatology |
| GCP | TM5-SURF2_EC | GCP_CH$_4$_2019 | GCP_CH$_4$ 2019 (global total: 15 Tg CH$_4$ yr$^{-1}$) | GCP_CH$_4$_2019 | GCP_CH$_4$_2019 | GCP_CH$_4$_2019 | GCP_CH$_4$_2019 | |
| GCP | CTE-GOSAT_FMI | GCP_CH$_4$_2019 | Etiope 2015 | GCP_CH$_4$_2019 (=GFED4s) | GCP_CH$_4$_2019 | GCP_CH$_4$_2019 | GCP_CH$_4$_2019 | |
| GCP | CTE-SURF_FMI | GCP_CH$_4$_2019 | Etiope 2015 | GCP_CH$_4$_2019 (=GFED4s) | GCP_CH$_4$_2019 | GCP_CH$_4$_2019 | GCP_CH$_4$_2019 | |
| | NAME-SURF_MetOffice | | | | | | | |

**Author contributions:**

AMRP designed this research and led the discussions with all co-authors, AMRP wrote the initial draft of the paper and edited all the following versions; CQ made the figures and helped with the review process, PPe, MJM, V B processed the original data submitted to the VERIFY portal, MJM, PPe and PB designed and are managing the VERIFY web portal, BM provided the new gap-filled yearly UNFCCC NGHGI Member States uncertainty calculations and checked language and content of the final version, GPP provided the figures in Appendix B1 and checked language and content of the final version, DBr and AT provided the data for the gridded figures in section

3.1.5. and AMRP made the figures from section 3.1.5., PPa, OT, RLT, CQ, GE, MK, LH-J, PR, RL, DBa, WW, provided in depth advice and commented/edited the initial versions of the manuscript, MB, RLT, AT, D Br, MK, PGE, GC, PR, RL, LH-J, WW, TA, AB, GC, MC, FD, CDGZ, AL, EL, AJM, MMa, TM, GDO, PKP, MS, AJJ, ES, HT, FNT, SZ. are data providers.

*Competing interests*

The authors declare that they have no conflict of interest.



**Acknowledgements**

We thank Aurélie Paquirissamy, Géraud Moulas and all ARTTIC team, for the great managerial support offered during the project. FAOSTAT statistics are produced and disseminated with the support of its member countries to the FAO regular budget. The views expressed in this publication are those of the author(s) and do not necessarily reflect the views or policies of FAO. We acknowledge the work of other members of the EDGAR group (Edwin Schaaf, Jos Olivier) and the outstanding scientific contribution to the VERIFY project of Peter Bergamaschi. Timo Vesala thanks the ICOS-Finland, University of Helsinki.

**Financial support**

This research has been supported by the European Commission, Horizon 2020 Framework Programme (VERIFY, grant no. 776810).

Ronny Lauerwald thanks the CLand Convergence Institute. Prabir Patra acknowledges support of the Environment Research and Technology Development Fund (JPMEERF20182002) of the Environmental Restoration and Conservation Agency of Japan. Pierre Regnier acknowledges the ESM 2025. David Basviken acknowledges support of the European Research Council (ERC) under the European Union's Horizon 2020 research and innovation programme (grant agreement No 725546 METLAKE). Greet Janssens-Maenhout acknowledges the European Union's Horizon 2020 research and innovation programme (CoCO2, grant no. 958927). Tuula Aalto acknowledges support from the Finnish Academy grants no. 351311, 345531. Sonke Zhaele acknowledges support from the ERC consolidator grant QUINCY (grant no. 647204).

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
