# Peer review of "The consolidated European synthesis of $CH_4$ and $N_2O$ emissions for EU27 and UK: 1990-2019"

_Earth System Science Data, 2022_

## Author Comment (AC1)

**Referee #1**

**General comments:**

This paper presents a European inventory of CH4 and N2O fluxes, which is an update from a recently-published inventory by the same lead author (released in 2021). The paper thoroughly documents reported emissions, as well as emissions inferred from top-down inversions. I think the authors could have done a stronger job explaining the differences between this current manuscript and the data synthesis they published just last year in 2021. This justification was distributed throughout the text, but I think the paper would really benefit from the authors laying out a strong case in the introduction for why this 2022 manuscript is necessary over the 2021 paper. This being said, I do agree that updating datasets often is valuable for the community, whether or not this includes a thorough re-analysis of trends. Given the importance of CH4 and N2O to climate change, the keeping emissions inventories current is clearly valuable.

We thank Referee #1 for his review comments and we acknowledge that the manuscript would indeed beneficiate from a stronger explanation regarding the updates. Therefore, we made some changes the final paragraph in the Introduction (L192-201) as following:

*"As Petrescu et al. (2021a) is the most comprehensive comparison of the NGHGI and research datasets (including both TD and BU approaches) for the EU27+UK to date, the focus of the current paper is on improvement of estimates in the most recent version in comparison with the previous one, including changes in the uncertainty estimates and identification of the knowledge gaps and added value for policy making. Such exercises of yearly updates are needed to improve the different respective approaches and furthermore can inform the development of formal verification systems. Official NGHGI emissions are compared with research datasets, including necessary harmonization of the latter on total emissions to ensure consistency. Differences and inconsistencies between emission estimates were analyzed, and recommendations were made towards future evaluation of NGHGI data. While NGHGIs include uncertainty estimates, the "uncertainty analysis should be seen, first and foremost, as a means to help prioritize national efforts to reduce the uncertainty of inventories in the future, and guide decisions on methodological choice" (Volume 1, Chapter 3, IPCC, 2006) and were therefore not developed to enable comparisons between countries or other datasets. In addition, individual spatially disaggregated research emission datasets often lack quantification of uncertainty. Here, the focus is on the median and minimum/maximum (min/max) range of different research products of the same type to get a first estimate of uncertainty (see Sect. 2). For those datasets providing uncertainties, new uncertainty reduction maps are presented (see Section 3.1.5). For those models/inventories who did not provided an update for this study, the previously published timeseries are shown."*

**Specific comments:**

Line 216: Please define LULUCF

Yes, thank you for noticing it. We added the definition.

Table 1 – since this table is meant to highlight data sources, please include references to the data sources. I also suggest reformatting the table structure. You could have 3 columns (Emissions sector, data source, data source reference), and then vertically stack the sections for the Anthropogenic BU, Natural BU, and TD

Thank you for your suggestion, the references to the data sources are all included in Table 2. Table 1 was meant to highlight the sectors covered by the current study, while data sources are of secondary importance here.

Line 298 – Here you say "to a small extent" is this extend calculation in national inventories? If so, I suggest including the value here instead of a vague description. If this contribution isn't calculated and you are assuming its relevance, can you provide a reference that eutrophication only contributes a small amount to total inland water CH4 emissions? If not, please rephrase this sentence.

Thank you for your comment. The natural $CH_4$ emissions are not included in the countries' NGHGIs. The leaching/run-off are reported under Agriculture (3.D.2.2) only for indirect $N_2O$ emissions and not for $CH_4$, therefore we can't add the inventory value. For clarity, we included the following paragraph and provided references for this statement, related to eutrophication.

*"Globally, the contribution of eutrophication is estimated to lead to a further increase in lake and reservoir emission by 30 to 90% over the 21$^{st}$ century, which would be the result of a ~3 times higher nutrient loading to lakes and reservoirs (Beaulieu et al., 2019), similar to the review by Li et al. 2021 who gathered a lot of prove that eutrophication significantly increase $CH_4$ emissions. In temperate Europe, eutrophication contributes significantly to the overall increase natural emissions and Rinta et al., 2017 found that eutrophic, central European lakes show $CH_4$ emission rates which are about one order of magnitude higher than those of oligotrophic boreal lakes and this study's model results are consistent with it."*

Li et al. 2021 (https://www.sciencedirect.com/science/article/pii/S0048969720381134?via%3Dihub)

Beaulieu et al., 2019, https://www.nature.com/articles/s41467-019-09100-5)

Rinta et al., 2017 (https://doi.org/10.4081/jlimnol.2016.1475)

Line 328 – When you say "While many different inversions have been used…" do you mean "While many different inversions exist…"

Thank you, we will rephrase this sentence as suggested.

Line 372 – 373 – It is not clear why you highlight that the first GST will include 2021, given that the GST will be in 2023. Do you mean that the first GST will include data up until 2021? If so, why don't your 5-year means align with a 2017 – 2021 window?

Thank you for your comment. Indeed, the GST connection here is a bit out of its scope, however what we were trying to say is that our 5yr mean results are of importance to the five-yearly GSTs, to give an overview of emission trends, even if, for the 2023 1$^{st}$ GST, we won't be able to provide an estimate, as this should be provided by the NGHGI compilers. At this point we are not able to include 2017-2021 mean because no dataset used has provided us with results up until 2021 and most importantly we would need the UNFCCC data will only be available in 2023 for 2021. We rephrased as following:

*"Figure 1 shows the total $CH_4$ fluxes from the NGHGIs for base year 1990, as well as five-year mean values for the 2011-2015 and 2015-2019 periods.  The five-year periods are informing on emission trends and what could be achieved by the GST process.  Given that the GST is only repeated every five years, a five-year average is clearly of interest even if, in this current study 2021 estimates are not available."*

Line 383 – What contributes to the decrease in methane emissions? It would be helpful to comment at least briefly on what is driving the trend, or point the reader to where in the text you discuss this.

Thank you for your question. The decrease in $CH_4$ emissions observed everywhere in Europe is mainly due to the EU's legislation, policies and strategies aimed at reducing emissions in Europe. The EU's methane emissions dropped by a third between 1990 and 2019. Since 1990s and up to 2005, the decreasing trend was triggered by the implementation in the early 90s of European and country-specific emission reduction policies on agriculture and the environment, as well as socioeconomic changes in the sector resulting in overall lower agricultural livestock and lower emissions from managed waste disposal on land and from agricultural soils. For the consequent periods (2005-2019), the relative agricultural $CH_4$ (sector which dominates the $CH_4$ emissions) reduction is smaller but still consistent between all data sources. For the Central and Eastern Europe, reductions were abrupt just after the dissolution of the Soviet Union (1989–1991) and the consequent structural changes in their economy of the former eastern European communist centralized economy block and in 1990 $CH_4$ emissions registered very high $CH_4$ reductions which afterwards showed a constant decreasing trend. (Petrescu et al., 2020).

We will add the following paragraph:

"*The decrease in $CH_4$ emissions is mainly due to the EU legislation policies and strategies starting with the implementation in the early 90s of European and country-specific emission reduction policies on agriculture and the environment, as well as socioeconomic changes in the sector resulting in overall lower agricultural livestock and lower emissions from managed waste disposal on land and from agricultural soils. After 2005, these trends maintain their decreasing trajectory, even if, at a lower intensity. For the Central and Eastern Europe, reductions were abrupt and mainly due to the dissolution of the Soviet Union (1989–1991) and the consequent structural changes in the economy of the former eastern European communist centralized economy block (Petrescu et al., 2020).*"

---

## Author Comment (AC2)

**Referee #2**

This manuscript is an update on the previous one (Petrescu et al., 2021), which synthesized and compared the estimates of CH4 and N2O emissions over Europe from NGHGI, and BU and TD approaches. The current manuscript follows the nice structure established in the previous version, while with improvements in uncertainty estimation and spatial patterns for posterior CH4 and N2O fluxes from inversions. The manuscript is well written with highlights on the improvements and changes compared to the previous version.

We thank Referee #2 for his review comments and we answer below to it:

A few comments and remarks as follows:

As the results are mainly for 1990-2019 with few data in 2020, the title should not include 2020.

We agree with the reviewer that the majority of datasets used in this study provide data until 2019, therefore we will change the title as suggested.

The resolution of the figures are low at least in the current pdf version.

Thank you. We will make sure, if the manuscript will be accepted for publication, to provide the journal with good quality figures as we already received from the modelers.

Figure 2 and Figure 10: The bars between 2011-2015 and 2015-2019 are difficult to tell. In particular the yellow uncertainty range are almost invisible. It could be better to use two bars instead one bar for each estimates, which will give clearer information on both mean and uncertainty.

Thank you for your comment. Figures 1 and 10, indeed, are complex and try to summarize the results. We separated the previous 2011-2015 average period and plotted it next to the 2015-2019 one.

[Figure]

[Figure]

Figure 4b The colors for inland waters, peatlands and mineral soils, geological emissions and biomass burning are difficult to tell.

We agree with the reviewer that natural $CH_4$ fluxes are similar in both magnitude and color, and hard to separate. Therefore we will add to Appendix B a separate figure (Figure B1c) for natural $CH_4$ emissions and the following text on L580:

*"....from BU natural sources, represented separate in Figure B1c, Appendix B.."*

[Figure]

Figure 4 and 13: It is a little surprising that top-down estimates varied a lot between Petrescu et al., 2021a and current study. It would be necessary to explain such differences (model improvements, site observation availability, satellite data availability etc.).

We agree with the reviewer that more explanation is needed regarding differences between previous results shown in Petrescu et al., 2021a and current study. Below we summarize the reasons as following, and this paragraph will be added from L588 onwards:

*"The differences between inversion results in current study and Petrescu et al., 2021a can be summarized as following: for the version used in this study, FLExKF-CAMSv19r_EMPA, the background mole fraction was taken from a global CAMS v19r assimilation run with assimilation of surface observations of $CH_4$ only (no satellite data) where the domain was cut out following the two-step approach of Rödenbeck et al. (2009). Background concentrations from CAMS v19r are on average about 5 ppb lower than those of the TM5-4DVAR system used previously, which results in somewhat higher emission estimates over Europe compared to Petrescu et al. (2021a). The major differences to previous CTE-FMI run are the prior fluxes, except for biomass-burning which remained GFED. The new VERIFY_S5 (core) run uses fluxes as described in Thomson et al., 2021. Lake & geological emissions were not included in Petrescu et al., 2021a synthesis, but are included in the current CTE-FMI simulation, which probably also contributes to higher total emissions. On top of this, the assimilated data (i.e. observation network) contributes to the differences (enlarged observation network, more sites (five core sites for $CH_4$ located in Spain, France and UK were added). The FLEXINVERT version used in this study updated the atmospheric observation network (more sites were added) as well as the prior emissions. The background mole fraction was also coupled with that from CAMS v19r assimilation run, which, similar to FLExKF model, might imply higher emissions."*

Regarding the CTE-FMI model we also replaced in Figure 8 the VERIFY S4 runs to S5, as the run S4 included the most assimilated sites, however for comparability reasons we should have used instead the runs having in both years 2006 and 2018 the same number of observation stations, referred here as S5 "core sites".

[Figure]

We therefore rewrote the following paragraph from Lines 719-727:

*"The second inversion system, CTE-$CH_4$ (Tsuruta et al., 2017) calculated the uncertainty reduction maps from surface inversions (SURF) for 2006 and 2018, as those used in Thompson et al., (2021), referred to here as VERIFY_S5*

*("core" inversion) (Figure 8). The system included two sets of inversions with different observation sets assimilated. However, the degrees of freedom in the state of the system was low, and therefore, the uncertainty estimates may not differ much between the two. The data from CTE-CH$_4$ includes uncertainties (standard deviations) and fluxes for 2006 and 2018. The differences in the simulations are observation sets and underlying prior covariance structure. "VERIFY_S5" uses data from only those sites that have long-term measurements assimilated i.e. there is little differences in the assimilated sites between the years. From the two panels of the Fig.8, higher uncertainty reductions are seen in 2018 compared to 2006 in E Poland, N Italy and Spain".*

L586: The differences in the trends could benefit a bit more explanation or discussion.

Regarding trends, we agree that more explanation is needed, however this will also require additional sensitivity tests and investigations from the modelers, which we hope to have for this year's new synthesis, hosted by the CoCO$_2$ EU funded project. Therefore, we will add the following text (on CH$_4$) after the new L588 paragraph describing the increase in CH$_4$ emissions:

*"Regarding decreasing trends seen for the current CH$_4$ results, for FLExKF model the trend in CH$_4$ emission was slightly negative over 2005-2019, at -0.48% per year, which is lower than the decrease in the prior of -0.8% per year. For the other models, based on Thompson et al., 2021, the differences in trends might be due to regional vs. global inversion differences."*

The N$_2$O discussion will be added after the Figure 13:

*"For the N$_2$O results (FLEXINVERT in Figure 13), after updating the last runs, both the magnitude and trends of the N$_2$O emissions changed. This new decreasing trend is confirming the UNFCCC trend but shows a larger average source after correcting for the estimate of natural emissions. Future work should focus on establishing the uncertainty and variability in the natural emissions of N$_2$O so that the results of inversion could be more directly compared to emission inventories."*

L608-612: For such interesting findings, it is necessary to explain it more clearly. The current speculation is not easy to understand.

We agree with the reviewer that the short paragraph in lines 608-612 required further explanation and clarification. We thus have added one sentence regarding the main drivers if thee seasonality and corrected the second sentence which contained a typo. Indeed, the words "for rivers" were misplaced in this sentence because the comparison only holds for lakes. We believe that this updated version clarifies the statement:

*"Model results however also reveal a strong seasonal variability in CH$_4$ emissions, with much lower fluxes during winter. **This seasonality is driven by physical factors (changing ice cover and bottom-water temperature) and biogeochemical factors (autotrophic primary production) that are well established drivers of the temporal variability in lake CH$_4$ emissions (Del Sontro et al., 2018; Jansen et al., 2022).** This finding **provides a likely explanation as to** why the spatio-temporally resolved model results lead to significantly lower estimates than observation-based methods that do not capture well the temporal variability in lake CH$_4$ emissions."*

DelSontro, T., Beaulieu, J. J., and Downing, J. A. (2018) Greenhouse gas emissions from lakes and impoundments: Upscaling in the face of global change, *Limnol. Oceanogr. Lett*. 3, 64–75.

Jansen, J., Woolway, R.I., Kraemer, B.M., Albergel, C., Bastviken, D. et al. (2022) Global increase in methane production under future warming of lake bottom waters. *Global Change Biology*, 28-18, 5427-5440, 2022. 10.1111/gcb.16298.

L652: "Further" rather than "father".

Thank you, we replaced farther, with further.